# MEISSONIC: REVITALIZING MASKED GENERATIVE TRANSFORMERS FOR EFFICIENT HIGH-RESOLUTION TEXT-TO-IMAGE SYNTHESIS

**Jinbin Bai**[1,2*], **Tian Ye**[3*], **Wei Chow**[6], **Enxin Song**[6], **Qing-Guo Chen**,
**Xiangtai Li**[2], **Zhen Dong**[5], **Lei Zhu**[3,4†], **Shuicheng Yan**[2,1†]

[1]National University of Singapore [2]Skywork AI [3]HKUST(GZ) [4]HKUST [5]UC Berkeley [6]ZJU

Model: https://huggingface.co/MeissonFlow/Meissonic
Code: https://github.com/viiika/Meissonic

## ABSTRACT

We present Meissonic, which advances non-autoregressive text-to-image Masked Image Modeling (MIM) to a level comparable with state-of-the-art diffusion models like SDXL. By incorporating a series of architectural innovations, advanced positional encoding strategies, and optimized sampling conditions, Meissonic significantly improves MIM's performance and efficiency. Additionally, we leverage high-quality training data, integrate micro-conditions informed by human preference scores, and employ feature compression layers to further enhance image fidelity and resolution. Our model not only matches but often exceeds the performance of existing methods in generating high-quality, high-resolution images. Extensive experiments validate Meissonic's capabilities, demonstrating its potential as a new standard in text-to-image synthesis.

## 1 INTRODUCTION

Diffusion models, such as Stable Diffusion (Rombach et al., 2022; Podell et al., 2024; per, 2024; Art, 2023), have rapidly advanced to become the dominant paradigm in visual generation by replacing Generative Adversarial Network (GAN). Recent developments like LlamaGen (Sun et al., 2024) have ventured into autoregressive image generation using discrete image tokens derived from VQ-VAE (Yu et al., 2022a). Despite progress, the substantial number of image tokens compared to text tokens makes autoregressive generation inefficient. For example, tokenizing one $1024 \times 1024$ image using a $16\times$ downsampled VQVAE yields 4096 tokens, where a sequential generation process is prohibitively slow.

Masked generative transformers, a class of generative models, have achieved significant results in the fields of image generation, Specifically, MaskGIT (Chang et al., 2022) introduced a more efficient, non-autoregressive alternative, where all image tokens are predicted simultaneously in a parallel, iterative refinement process. Then, MUSE (Chang et al., 2023) extended this technique to higher resolutions, achieving $512 \times 512$ resolution T2I generation. These non-autoregressive methods offer around 99% reduction in decoding steps compared to autoregressive methods. However, despite their efficiency, non-autoregressive transformers remain limited in performance compared to advancing diffusion or autoregressive models, particularly in high-quality, high-resolution text-to-image synthesis.

In this work, we address these challenges and introduce two key innovations to make masked image modeling (MIM) competitive with advanced diffusion models:

**Enhanced Transformer Architecture**: Previous MIM methods (Chang et al., 2023; 2022) predominantly utilized naive transformer architectures, potentially limiting their capabilities. We dis-

---

*Equal contribution. ✉: jinbin.bai@u.nus.edu †Corresponding authors.

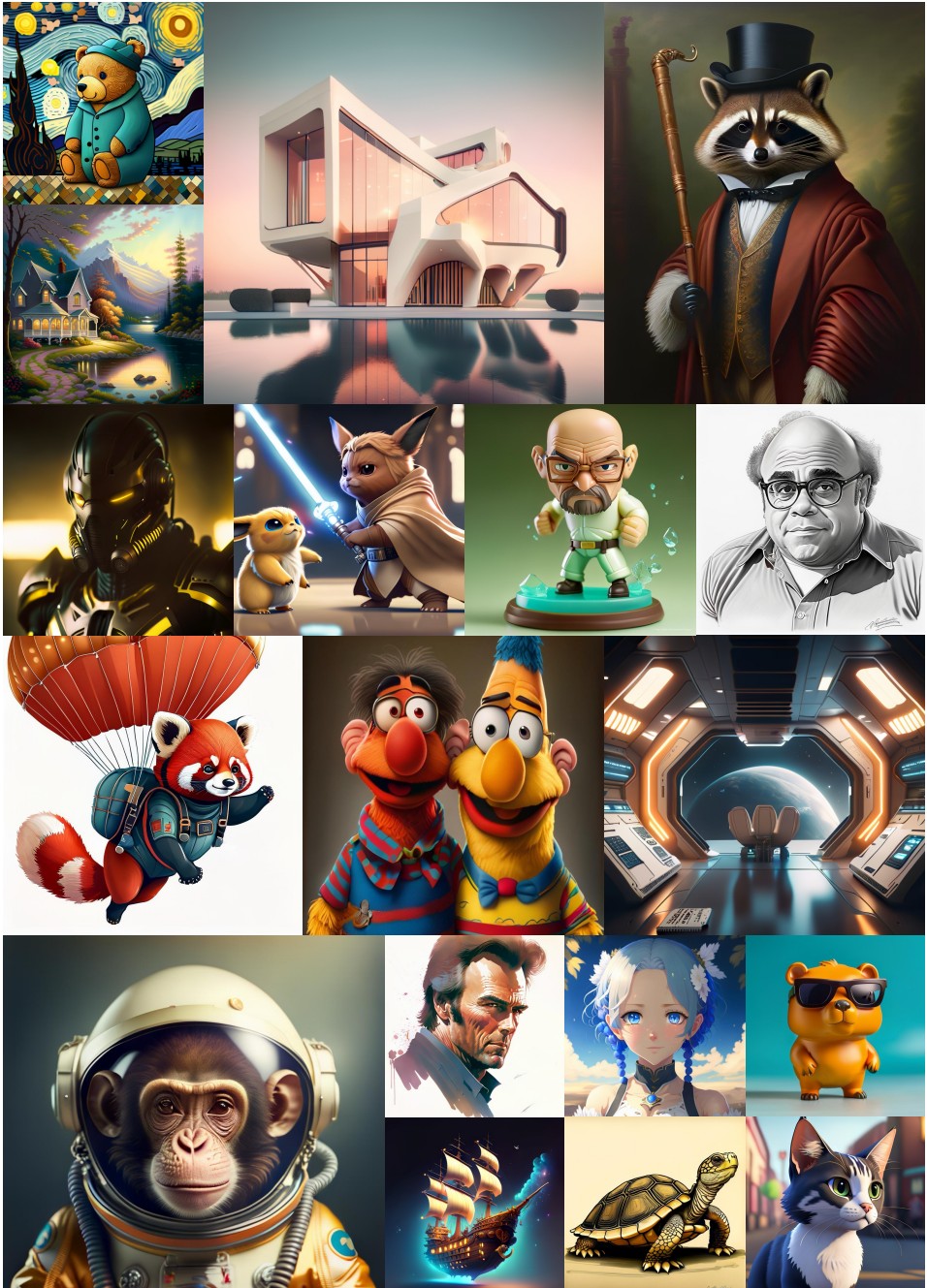

Figure 1: Images produced by Meissonic exhibit exceptional image quality. More samples can be found in Appendix M. Notably, Meissonic can effortlessly produce images with solid-color backgrounds without requiring any additional modifications.

covered that a combination of multi-modal and single-modal transformer layers can significantly boost MIM training efficiency and performance. Language and vision representations are inherently different. The multi-modal transformer can effectively capture cross-modal interactions, extracting information from unpooled text representations and effectively bridging the gap between these distinct modalities. This allows the model to harness useful signals from noisy data. Additionally, subsequent single-modal transformer layers refine the visual representation, improving performance and training stability.

**Advanced Positional Encoding & Masking Rate as Sampling Condition**: We incorporate Rotary Position Embedding (RoPE) (Su et al., 2024) for encoding positional information in queries and

keys, which helps maintain detail in high-resolution images. RoPE effectively addresses the issue of context disassociation in transformers as the number of tokens increases. Additionally, we introduce the masking rate as a dynamic sampling condition throughout the generation process. Previous MIM methods Chang et al. (2023; 2022) have overlooked this aspect, resulting in suboptimal image details. This issue arises because the number of tokens predicted by the MIM model changes dramatically throughout the sampling loop. With the masking rate condition, the model can ascertain the current stage of the sampling period by leveraging conditional information from the masking rate.

Beyond these architectural improvements, to achieve comparable performance with SDXL for high-resolution generation, we adopt effects in three additional aspects:

**High-Quality Training Data**: The quality of training data is crucial. While LAION (Schuhmann et al., 2022) offers a diverse visual dataset, its captions can be subpar (Chen et al., 2024). We curated a high-quality internal dataset with accurate captions, which, combined with our training strategy, significantly improved the generative capabilities of the base model.

**Micro-Conditioning**: We identified that incorporating original image resolution, crop coordinates, and human preference scores (Wu et al., 2023) as micro-conditions greatly enhances model stability during high-resolution aesthetic training.

**Feature Compression Layers**: To efficiently generate high-resolution images, we integrated feature compression layers, maintaining computational efficiency even at $1024 \times 1024$ resolution.

Our contributions culminate in **Meissonic**, a next-generation T2I model based on masked discrete image token modeling. Unlike larger diffusion models such as SDXL (Podell et al., 2024) and DeepFloyd-XL (Liu et al., 2024a), Meissonic, with just 1B parameters, offers comparable or superior $1024 \times 1024$ high-resolution, aesthetically pleasing images while being able to run on consumer-grade GPUs with only 8GB VRAM without the need for any additional model optimizations. Moreover, Meissonic effortlessly generates images with solid-color backgrounds, a feature that usually demands model fine-tuning or noise offset adjustments in diffusion models.

Advancement of Meissonic represents a significant stride towards high-resolution, efficient, and accessible T2I MIM models. We evaluate Meissonic using various qualitative and quantitative metrics, including HPS, MPS, GenEval benchmarks, and GPT4o assessments, demonstrating its superior performance and efficiency.

## 2 METHOD

### 2.1 MOTIVATION

Recent breakthroughs in text-to-image synthesis have been largely propelled by diffusion models, such as Stable Diffusion XL, which have set *de facto* standards for image quality, detail, and conceptual fidelity.

Another approach, non-autoregressive Masked Image Modeling (MIM) techniques, exemplified by MaskGIT and MUSE, has shown potential for **efficient** image generation to replace slow autoregressive techniques like Llamagen. Yet, despite their promise, MIM approaches face two critical limitations:

**(a) Resolution Constraint.** Current MIM methods are limited to generating images at a maximum resolution of $512 \times 512$ pixels. This limitation hinders their broader adoption and advancement, particularly as the text-to-image synthesis community increasingly adopts $1024 \times 1024$ resolution as the standard.

**(b) Performance Gap.** Existing MIM techniques have not yet achieved the level of performance exhibited by leading diffusion models like SDXL. They notably underperform in key areas such as image quality, intricate detailing, and conceptual representation, which are critical for practical applications.

These challenges necessitate the exploration of new approaches. Our objective is to empower MIM to efficiently generate high-resolution images (e.g., $1024 \times 1024$), while narrowing the gap with top-tier diffusion models, and ensuring computational efficiency suitable for consumer-grade hardware.

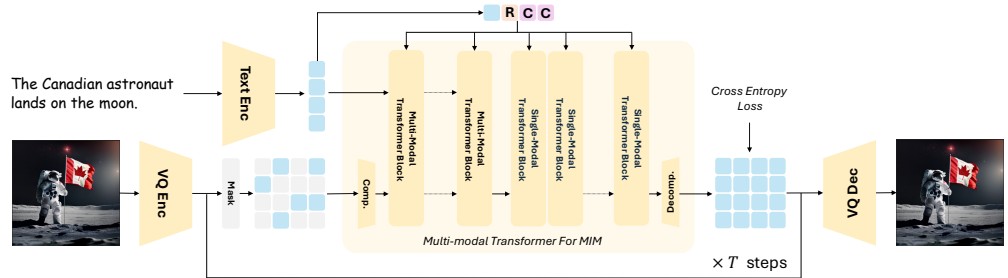

Figure 2: **The architecture of Meissonic.** During the image generation process, discrete tokens are created randomly according to a predefined schedule. Meissonic then applies masking and performs predictions over several steps to reconstruct all tokens and decode the resulting image. In the case of image editing, the original image is converted into discrete tokens, which are masked according to a specified masking strategy. After a series of processing steps, the masked tokens are reconstructed and utilized to decode the target image. Text prompts and other conditions are incorporated to control the synthesis process. $R$ represents the masking rate condition, and $C$ indicates the micro conditions. More details about Multi-modal Transformer Block can be found in Appendix I.

Through our work, Meissonic, we aim to push the boundaries of MIM methods and bring them to the forefront of text-to-image synthesis.

## 2.2 MODEL ARCHITECTURE

The Meissonic model is architected to facilitate efficient high-performance text-to-image synthesis through an integrated framework comprising a CLIP text encoder (Radford et al., 2021), a vector-quantized (VQ) image encoder and decoder (Esser et al., 2021), and a Multi-modal Transformer backbone. Figure 2 illustrates the overall structure of the model.

**Vector-quantized Image Encoder and Decoder.** We employ a VQ-VAE model (Esser et al., 2021) to convert raw image pixels into discrete semantic tokens. This model comprises an encoder, a decoder, and a quantization layer that maps input images into sequences of discrete tokens using a learned codebook. For an image of size $H \times W$, the encoded token size is $\frac{H}{f} \times \frac{W}{f}$, where $f$ represents the downsampling ratio. In our implementation, we utilize a downsampling ratio of $f = 16$ and a codebook size of 8192, allowing a $1024 \times 1024$ image to be encoded into a sequence of $64 \times 64$ discrete tokens.

**Flexible and Efficient Text Encoder.** Instead of using large language model encoders, such as T5-XXL[1] (Raffel et al., 2020) or LLaMa (Touvron et al., 2023), which are prevalent in previous works (Chen et al., 2024; Esser et al., 2024), we utilize a single text encoder from the state-of-the-art CLIP model with a latent dimension of 1024, and fine-tune for optimal T2I performance. While this decision may limit the model's capacity to fully comprehend lengthy text prompts, our observations indicate that excluding large-scale text encoders like T5 does not diminish visual quality. Moreover, this approach significantly reduces GPU memory requirements and computational cost. Notably, offline extraction of T5 features would entail approximately 11 times more processing time and 6 times more storage than employing the CLIP text encoder, underscoring the efficiency of our design.

**Multi-modal Transformer Backbone for Masked Image Modeling.** Our transformer architecture builds upon the Multi-modal Transformer framework (Sauer et al., 2024), incorporating sampling parameters $r$ to encode sampling parameters and Rotary Position Embeddings (RoPE) (Su et al., 2024) for spatial information encoding. We introduce feature compression layers to efficiently handle high-resolution generation with numerous discrete tokens. These layers compress embedding features from $64 \times 64$ to $32 \times 32$ before processing through the transformer, and followed by feature decompression layers to $64 \times 64$, thereby alleviating computational burdens. To enhance training stability and mitigate the *NaN Loss* issue, we follow the training strategy from LLaMa Touvron et al.

---

[1]Many works indicate that the T5 text encoder is the key factor in obtaining the ability to synthesize words, we still show the ability to synthesize letters in Figure 10. We leave this a future improvement.

(2023), implementing gradient clipping and checkpoint reloading during distributed training and integrating QK-Norm layers into the architecture. We elaborate on the designs of our transformer in the subsequent section.

**Diverse Micro Conditions.** To augment generation performance, we incorporate additional conditions such as original image resolution, crop coordinates, aesthetic score, and human preference score (Wu et al., 2023). These conditions are transformed into sinusoidal embeddings and concatenated as additional channels to the final pooled hidden states of the text encoder.

**Masking Strategy.** Following the approach established in Chang et al. (2023), we employ a variable masking ratio with cosine scheduling. Specifically, we randomly sample a masking ratio $r \in [0, 1]$ from a truncated $arccos$ distribution characterized by the following density function:

$$p(r) = \frac{2}{\pi}(1 - r^2)^{-\frac{1}{2}}$$

In contrast to autoregressive models that learn conditional distributions $P(x_i \mid x_{<i})$ for fixed token orders, our approach utilizes random masking with variable ratios to enable the model to learn $P(x_i \mid x_\Lambda)$ for arbitrary subsets of tokens $\Lambda$. This flexibility is pivotal for our parallel sampling strategy and facilitates various zero-shot image editing capabilities, which will be demonstrated in Section 3.

## 2.3 MULTI-MODAL TRANSFORMER FOR MASKED IMAGE MODELING

Meissonic employs the Multi-modal Transformer as its foundational architecture and innovatively customizes the modules to address the distinctive challenges inherent in high-resolution masked image modeling. We introduce several specialized designs for MIM as follows:

- *Rotary Position Embeddings.* RoPE (Su et al., 2024) has demonstrated exceptional performance within in LLMs (Su et al., 2024; Touvron et al., 2023; Ding et al., 2024; Bai et al., 2023). Some studies (Lu et al., 2024; Lin et al., 2023; Zhuo et al., 2024) have attempted to extend 1D RoPE (Su et al., 2024) to 2D or 3D for image diffusion models. We apply rotary position embeddings to each query and key features in attention layers.

- *Deeper Model with Single-modal Transformer.* Although the Multi-modal Transformer block demonstrated commendable performance, reducing the number of multi-modal blocks to a single-modal block configuration offers a more stable and computationally efficient approach for training T2I models. Therefore, we opt to employ Multi-modal Transformer blocks in the initial stages of the network, transitioning to exclusively Single-modal Transformer blocks in the latter half. Our findings suggest an optimal block ratio of about 1:2.

- *Micro Conditions with Human Preference Score.* Incorporating three micro-conditions is pivotal for achieving a stable and reliable High-resolution MIM Model: original image resolution, crop coordinates, and human preference score. The original image resolution effectively aids the model in implicitly filtering out low-quality data and learning the properties of high-quality, high-resolution data, while crop coordinates enhance training stability, likely due to improved consistency between image conditions and semantic conditions during cropped patch coordination. In the final stage, we leverage the Human Preference Score (Wu et al., 2023) to effectively enhance image quality, using signals provided by the Human Preference Model to guide the model's outputs in mimicking and approximating human preferences.

- *Feature Compression Layers.* Unlike multi-stage approaches like MUSE Chang et al. (2023) and DeepFloyd-XL DeepFloyd (2023), which employ cascading multiple subnetworks for higher-resolution image generation. We simplify the process by integrating streamlined feature compression layers during the fine-tuning stage to facilitate efficient high-resolution generation process learning. This approach functions akin to a lightweight high-resolution adapter Guo et al. (2024), a module extensively explored and integrated within Stable Diffusion. By incorporating 2D convolution-based feature compression layers into the transformer backbone, we compress the feature maps prior to the transformer layers and subsequently decompress them after the transformer layers, effectively addressing the challenges of efficiency and resolution transition.

Table 2: HPS v2.0 benchmark. Scores are collected from https://github.com/tgxs002/HPSv2. We highlight the **best**.

| Model | HPS v2.0 | | | | |
|-------|----------|---|---|---|---|
| | Animation | Concept-art | Painting | Photo | Averaged |
| DALL·E 2 (Ramesh et al., 2022) | 27.34 | 26.54 | 26.68 | 27.24 | 26.95 |
| Stable Diffusion v1.4 (Rombach et al., 2022) | 27.26 | 26.61 | 26.66 | 27.27 | 26.95 |
| Stable Diffusion v2.0 (Rombach et al., 2022) | 27.48 | 26.89 | 26.86 | 27.46 | 27.17 |
| SDXL Base 0.9 (Podell et al., 2024) | 28.42 | 27.63 | 27.60 | 27.29 | 27.73 |
| Realistic Vision (rea, 2024) | 28.22 | 27.53 | 27.56 | 27.75 | 27.77 |
| SDXL Refiner 0.9 (Podell et al., 2024) | 28.45 | 27.66 | 27.67 | 27.46 | 27.80 |
| SDXL Base 1.0 (Podell et al., 2024) | 28.88 | 27.88 | 27.92 | 28.31 | 28.25 |
| SDXL Refiner 1.0 (Podell et al., 2024) | 28.93 | 27.89 | 27.90 | 28.38 | 28.27 |
| Meissonic-512 | 28.90 | 28.15 | 28.22 | 28.04 | 28.33 |
| Meissonic | **29.57** | **28.58** | **28.72** | **28.45** | **28.83** |

## 2.4 TRAINING DETAILS

Meissonic is constructed using a CLIP-ViT-H-14[2] text encoder (Ilharco et al., 2021), a pre-trained VQ image encoder and decoder (Patil et al., 2024), and a customized Transformer-based (Esser et al., 2024) backbone. We employ classifier-free guidance (CFG) (Ho & Salimans, 2022) and cross-entropy loss to train Meissonic. Training occurs across three resolution stages, leveraging both public datasets and our curated data. First, we train Meissonic-256 with a batch size of 2,048 for 100,000 steps. Second, we

Table 1: Comparison of training data and time for various models.

| Model | Params (B) | Training Images (M) | 8×A100 GPU Days[a] |
|-------|-----------|---------------------|---------------------|
| Würstchen (Pernias et al., 2024) | 1.0 | 1420 | 128.1 |
| SD-1.5 (Rombach et al., 2022) | 0.9 | 4800 | 781.2 |
| SD-2.1 (Rombach et al., 2022) | 0.9 | 3900 | 1041.6 |
| Imagen (Saharia et al., 2022) | 3.0 | 860 | 891.5 |
| Dall-E 2 (Ramesh et al., 2022) | 6.5 | 650 | 5208.3 |
| GigaGAN (Kang et al., 2023) | 0.9 | 980 | 597.8 |
| SDXL (Podell et al., 2024) | 2.6 | unknown | unknown |
| **Meissonic** | 1.0 | **210** | **19**[b] |

[a] Data collected from Sehwag et al. (2024).
[b] FP16 Tensor Core of A100 is 312 TFLOPS and H100 is 756.5 TFLOPS. GPU hours are adjusted from 48 H100 days based on this rate.

continue training Meissonic-512 with a batch size of 512 for an additional 100,000 steps. Third, we continue training Meissonic with a batch size of 256 for 42,000 steps with a resolution of $1024 \times 1024$. The performance results of Meissonic-512 and Meissonic are reported in Table 2. All experiments are carried out with a fixed learning rate of $1 \times 10^{-4}$. Further details are elaborated in Sec. 2.5. All inferences in this paper are performed with CFG = 9 and 48 steps.

It's crucial to highlight the resource efficiency of our training process. Our training is considerably more resource-efficient compared to Stable Diffusion (Podell et al., 2024). Meissonic is trained in approximately 48 H100 GPU days, demonstrating that a production-ready image synthesis foundation model can be developed with considerably reduced computational costs. Additional details on this comparison can be found in Table 1.

## 2.5 PROGRESSIVE AND EFFICIENT TRAINING STAGE DECOMPOSITION

Our approach systematically decomposes the training process into four carefully designed stages, allowing us to progressively build and refine the model's generative capabilities. These stages, combined with precise enhancements to specific components, contribute to continual improvements in synthesis quality. Given that SDXL has not disclosed details regarding its training data, our experience is particularly valuable for guiding the community in constructing SDXL-level text-to-image models. We present images generated by Meissonic at each of the four training stages in Figure 3 to support our claims. More examples can be found in Appendix K.

**Stage 1: Understanding Fundamental Concepts from Extensive Data.** Previous studies (Chen et al., 2024; Yu et al., 2024) indicate that raw captions from LAION are insufficient for training text-to-image models, often requiring the caption refinement provided by MLLMs such as LLaVA (Liu

---

[2]We utilize "laion/CLIP-ViT-H-14-laion2B-s32B-b79K" from OpenCLIP as our initial weights.

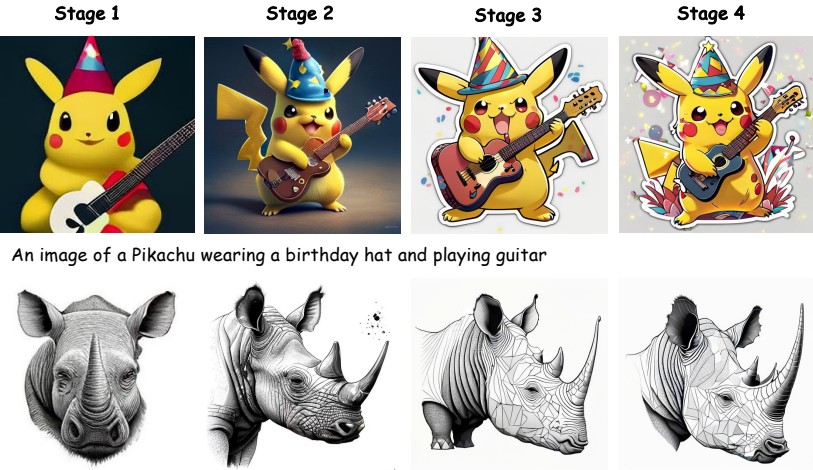

Figure 3: Images generated using the same prompt across Meissonic's four training stages. The resolutions for stages 1 and 2 are $256^2$ and $512^2$, respectively, while stages 3 and 4 are $1024^2$. For clarity and comparison, all images are displayed in a consistent layout.

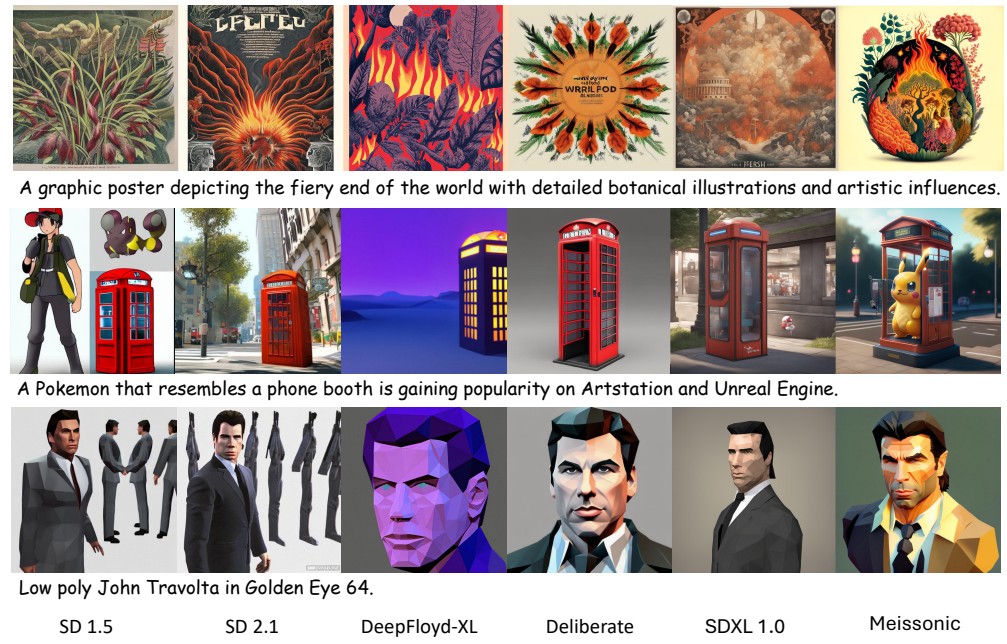

Figure 4: Qualitative Comparisons with SD 1.5, SD 2.1, DeepFloyd-XL, Deliberate, and SDXL.

et al., 2024b). However, this solution is computationally demanding and time-intensive. While some studies (Chen et al., 2024; Sehwag et al., 2024) utilize the extensively annotated SA-10M (Kirillov et al., 2023) dataset, our findings reveal that SA-10M does not comprehensively cover fundamental concepts, particularly regarding human faces. Thus, we carefully curated the deduplicated LAION-2B dataset by filtering out images with aesthetic scores below 4.5, watermark probabilities exceeding 50%, and other criteria outlined in Kolors (2024). This meticulous selection resulted in approximately 200 million images, which were employed for training at a resolution of $256 \times 256$ in this initial stage.

Table 3: GenEval benchmark. We highlight the **best** result.

| Model | Overall | Objects | | Counting | Colors | Position | Attribution |
|---|---|---|---|---|---|---|---|
| | | Single | Two | | | | |
| DALL-E mini | 0.23 | 0.73 | 0.11 | 0.12 | 0.37 | 0.02 | 0.01 |
| SD v1.5 | 0.43 | 0.97 | 0.38 | 0.35 | 0.76 | 0.04 | 0.06 |
| SD v2.1 | 0.50 | 0.98 | 0.51 | 0.44 | 0.85 | 0.07 | 0.17 |
| DALL-E 2 | 0.52 | 0.94 | 0.66 | **0.49** | 0.77 | 0.10 | 0.19 |
| SD XL | **0.55** | 0.98 | **0.74** | 0.39 | 0.85 | **0.15** | **0.23** |
| Meissonic | 0.54 | **0.99** | 0.66 | 0.42 | **0.86** | 0.10 | 0.22 |

Table 4: MPS scores on RealUser-800 Prompts. We highlight the **best** result.

| Model | MPS |
|---|---|
| Stable Diffusion v1.4 [Rombach et al. (2022)] | 13.89 |
| Stable Diffusion v2.0 [Rombach et al. (2022)] | 14.39 |
| SDXL Base 1.0 [Podell et al. (2024)] | 16.46 |
| SDXL Refiner 1.0 [Podell et al. (2024)] | 16.56 |
| Meissonic | **17.34** |

**Stage 2: Aligning Text and Images with Long Prompts.** In the second stage, we focus on improving the model's capability to interpret long, descriptive prompts. We filtered the initial LAION set more rigorously, retaining only images with aesthetic scores above 8, and other criteria outlined in Kolors (2024). Additionally, we incorporate 1.2 million synthetic image-text pairs with refined captions exceeding 50 words, primarily derived from publicly available high-quality synthetic datasets, complemented by additional high-quality images from our internal 6 million dataset. This aggregation results in around 10 million image-text pairs. Notably, we maintain the model architecture while increasing the training resolution to $512 \times 512$, enabling the model to capture more intricate image details. We observed a significant boost in the model's ability to capture abstract concepts and respond accurately to complex prompts, including diverse styles and fantasy characters.

**Stage 3: Mastering Feature Compression for Higher-resolution Generation.** High-resolution generation remains an unexplored area within MIM (Chang et al., 2023; 2022; Patil et al., 2024). Unlike methods such as MUSE(Chang et al., 2023) or DeepFloyd-XL (DeepFloyd, 2023), which rely on external super-resolution (SR) modules, we demonstrate that efficient $1024 \times 1024$ generation is feasible through feature compression for MIM. By introducing feature compression layers, we achieve a seamless transition from $512 \times 512$ to $1024 \times 1024$ generation with minimal computational cost. In this stage, we further refine the dataset by filtering based on resolution and aesthetic score, selecting approximately 100K high-quality, high-resolution image-text pairs from the LAION subset utilized in Stage 2. This, combined with the remaining high-quality data, results in approximately 6 million samples for training at 1024 resolution.

**Stage 4: Refining High-Resolution Aesthetic Image Generation.** In the final stage, we fine-tune the model using a small learning rate, without freezing the text encoder, and incorporate aesthetic score as a micro condition. This can significantly enhance the model's performance in high-resolution image generation. This targeted adjustment significantly enhances the model's performance in generating high-resolution images, while also improving diversity. The training data remains the same as in Stage 3.

## 3 RESULTS

### 3.1 QUANTATIVE COMPARISON

Classic evaluation metrics for image generation models, such as FID and CLIP Score, have limited relevance to visual aesthetics, as highlighted by Podell et al. (2024); Chen et al. (2024); Kolors (2024); Sehwag et al. (2024). Therefore, we report our model's performances using Human Preference Score v2 (HPSv2) (Wu et al., 2023), GenEval (Ghosh et al., 2024), and Multi-Dimensional Human Preference Score (MPS)[3] (Zhang et al., 2024b), as illustrated in Table 2,3,4.

In our pursuit of making Meissonic accessible to the broader community, we optimized our model to 1 billion parameters, ensuring that it runs efficiently on 8GB VRAM, making inference and fine-tuning both convenient. Figure 5 provides a comparative analysis of GPU memory con-

Table 5: Comparison of 1 step (50 steps) inference time (s) for Different Models and Batch Sizes.

| Model | Batch Size | | | |
|---|---|---|---|---|
| | 1 | 2 | 4 | 8 |
| SDXL Base 1.0 | 0.36 (5.38) | 0.75 (10.06) | 1.41 (19.69) | 2.79 (38.58) |
| Meissonic-256 | 0.09 (3.11) | 0.10 (3.14) | 0.11 (3.22) | 0.16 (4.70) |
| Meissonic-512 | 0.13 (3.24) | 0.17 (4.24) | 0.28 (7.74) | 0.51 (14.51) |
| Meissonic-1024 | 0.24 (3.48) | 0.35 (4.62) | 0.62 (8.52) | 1.17( 16.46) |

---

[3]Given that the KolorsPrompts benchmark was unavailable, we curated a diverse prompt dataset consisting of 800 real user-generated prompts spanning various concepts and themes for the MPS evaluation.

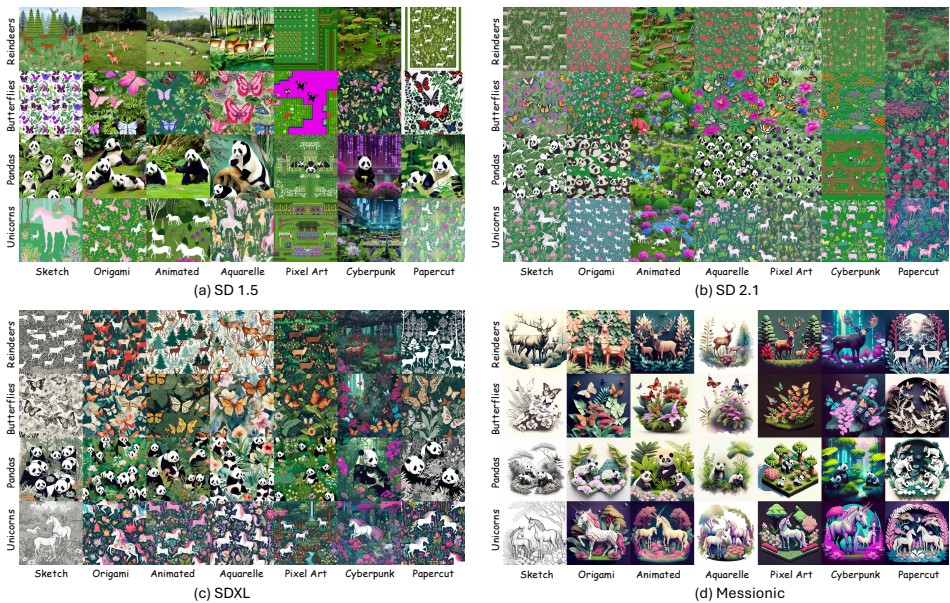

Figure 6: Evaluating the ability to generate diverse styles. *Prompt*: A garden full of [Y] illustrated in [X] style.

sumption[4] across different inference batch sizes against SDXL. Additionally, Table 5 details the inference time per step[5].

## 3.2 QUALITATIVE COMPARISON

We also present qualitative comparisons of image quality and text-image alignment in Figure 4, with additional comparisons provided in the Appendix. Furthermore, Figure 6 illustrates Meissonic's proficiency in generating text-driven style art image. To complement these analyses, we conduct human evaluation by K-Sort Arena (Li et al., 2024), and we conduct GPT-4o to evaluate the performance between Meissonic and other models in Figure 9.

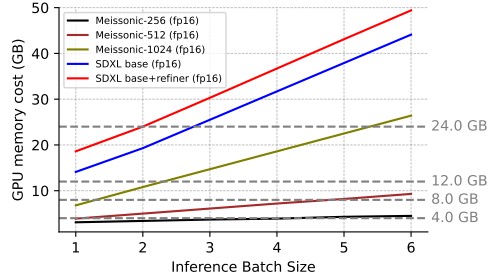

Figure 5: GPU Memory Cost vs Inference Batch Size for Different Models.

All Figures and Tables demonstrate that Meissonic achieves competitive performance in human performance and text alignment compared to DALL-E 2 and SDXL, as well as showcasing its efficiency.

## 3.3 ZERO-SHOT IMAGE-TO-IMAGE EDITING

For image editing tasks, we benchmarked Meissonic against state-of-the-art models using the EMU-Edit dataset (Sheynin et al., 2024), with results presented in Table 6. Additionally, examples from HumanEdit Bai et al. (2024), including mask-guided editing in Figure 7 and mask-free editing in Figure 8, further showcase Meissonic's versatility. Remarkably, Meissonic

| Model | CLIP-I↑ | CLIP-T↑ | DINO↑ |
|---|---|---|---|
| InstructPix2Pix (Brooks et al., 2023) | 0.834 | 0.219 | 0.762 |
| MagicBrush (Zhang et al., 2024a) | 0.838 | 0.222 | 0.776 |
| PnP (Tumanyan et al., 2023) | 0.521 | 0.089 | 0.153 |
| Null-Text Inv. (Mokady et al., 2023) | 0.761 | 0.236 | 0.678 |
| EMU-Edit (Sheynin et al., 2024) | 0.859 | 0.231 | **0.819** |
| Meissonic | **0.871** | **0.266** | 0.760 |

Table 6: Results on the EMU-Edit test set.

achieved this performance without any training or fine-tuning on image editing-specific data or instruction dataset. More comparisons for zero-shot image editing ability can be found in Appendix F.

---

[4]GPU memory usage was gauged using `torch.cuda.memory_reserved()`. While this method might yield higher values, all models are measured under identical settings to maintain fairness.

[5]Inference time is assessed using an A100 GPU with fp16 models. Notably, the reported times contributions from the VAE and text encoder, meaning that multi-step inferences do not scale linearly.

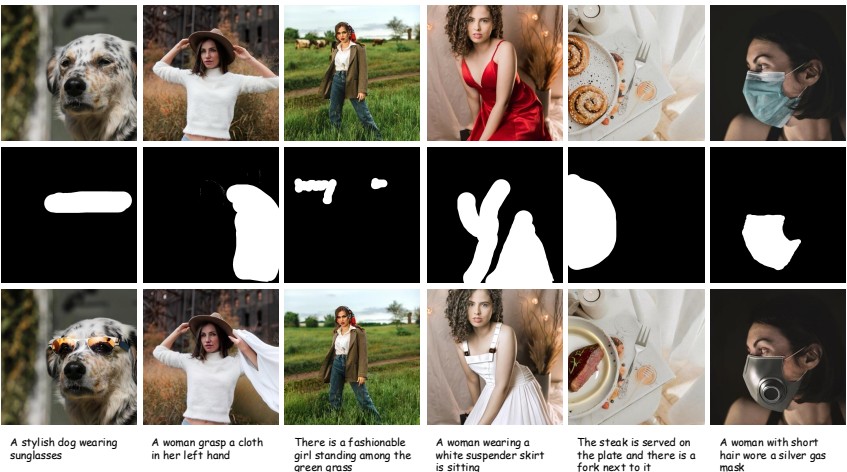

Figure 7: Examples of image editing with mask on internal Image Editing Dataset

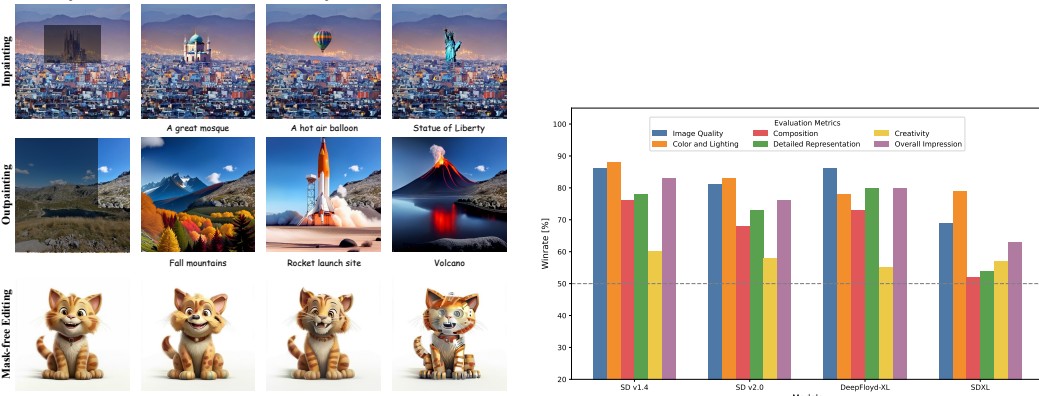

Figure 8: Examples of image inpainting, out-painting, and mask-free image editing on our internal Image Editing Dataset

Figure 9: GPT4o Preference Evaluation of Meissonic against current open Text-to-image Models.

## 4 CONCLUSION AND IMPACT

In this work, we have significantly advanced masked image modeling (MIM) for text-to-image (T2I) synthesis by introducing several key innovations: a transformer architecture blends multi-modal and single-modal layers, advanced positional encoding strategies, and an adaptive masking rate as the sampling condition. These innovations, coupled with high-quality curated training data, progressive and efficient training stage decomposition, micro-conditions, and feature compression layers, have culminated in Meissonic, a 1B parameter model that outperforms larger diffusion models in high-resolution, aesthetically pleasing image generation while remaining accessible on consumer-grade GPUs. Our evaluations demonstrate Meissonic's superior performance and efficiency, marking a significant step towards accessible and efficient high-resolution non-autoregressive T2I MIM models.

**Broader Impact.** Recently, offline text-to-image applications on mobile devices have emerged, such as Pixel Studio from Google Pixel 9 and Image Playground from Apple iPhone. These innovations reflect a growing trend toward enhancing user experience and privacy. As a pioneering resource-efficient foundation model, Meissonic represents a significant advancement in this field, delivering state-of-the-art image synthesis capabilities with a strong emphasis on user privacy and offline functionality.

**Acknowledgements.** This work was supported in part by NUS Start-up Grant A-0010106-00-00, the Guangdong Science and Technology Department (No. 2024ZDZX2004), the Nansha Key Area Science and Technology Project (No. 2023ZD003) and the InnoHK funding launched by Innovation and Technology Commission, Hong Kong SAR.

We would like to express our gratitude to all those who contributed their time, expertise, and insights during the development of Meissonic. Listed in no particular order: Jingjing Ren, Sixiang Chen from HKUST(GZ), Wenhao Chai from University of Washington, Donghao Zhou from CUHK, and other anonymous friends. We are profoundly grateful for their commitment and the unique perspectives they brought to this project.

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

## A    MODEL NAME ORIGIN

The name "Meissonic" is derived from a combination of the renowned French painter Ernest Meissonier and the term "sonic". Ernest Meissonier is celebrated for his meticulous attention to detail and his ability to capture dynamic moments in art. The addition of "sonic" evokes a sense of speed and modernity, highlighting the model's capabilities in efficient image synthesis and transformation.

## B    RELATED WORK

**Diffusion-based Image Generation.** Diffusion models have achieved remarkable advances in image generation, with notable contributions like Stable Diffusion (Rombach et al., 2022), and the more recent SDXL (Podell et al., 2024), often driven by large-scale datasets. These models move beyond pixel-level operations by working within compressed latent spaces, forming what we now recognize as latent diffusion models (Luo et al., 2023; Podell et al., 2024; Wu et al., 2024a; Shi et al., 2024; Zhou et al., 2024; Yi et al., 2024; Wu et al., 2024b). SDXL represents a significant leap in this domain, introducing micro-conditions and multi-aspect training to gain greater control over image generation, which has inspired a wide range of derivative models in the community, such as Deliberate (per, 2024) and RealVisXL (rea, 2024).

The integration of transformer architectures has also become more prevalent, with models like DiT (Peebles & Xie, 2023) and U-ViT (Bao et al., 2023) demonstrating the potential of diffusion transformers in this field. SD3 (Esser et al., 2024), which combines diffusion transformers with flow matching at an impressive scale of 8B parameters, underscores the scalability and potential of the multimodal transformer-based diffusion backbone. Despite these advances, diffusion models still face challenges, particularly their reliance on acceleration techniques (Sauer et al., 2023; Luo et al., 2023; Yin et al., 2024) to speed up inference, making them cumbersome for real-time applications. Additionally, the quantization of diffusion transformers has proven less straightforward than with large language models (Li et al., 2023). The research community continues to explore better paradigms for image generation. Addressing these limitations, our work aims to contribute an efficient, high-quality alternative in the form of Meissonic.

**Token-based Image Generation.** Token-based autoregressive transformers (Lee et al., 2022; Chen et al., 2018; Yu et al., 2022b), first validated by VQ-GAN (Esser et al., 2021), have shown considerable promise for image generation. However, these methods are inherently computationally demanding, requiring the prediction of hundreds to thousands of tokens to form a single image. As a pioneering work, MaskGIT (Chang et al., 2022) challenged this paradigm by introducing a masked image modeling (MIM) approach, achieving competitive fidelity and diversity in class-conditional image generation. Building on this, MUSE (Chang et al., 2023) extended MIM to text-to-image synthesis, scaling up to 3B parameters and achieving remarkable performance.

MUSE demonstrates the viability of non-autoregressive token-based models, but it encountered limitations in generating high-resolution images, capping at $512 \times 512$, and lagging behind SDXL (Podell et al., 2024) in terms of fidelity and text-image alignment. Meissonic advances the performance of token-based models beyond what latent diffusion methods have achieved, effectively pushing the envelope in terms of both quality and resolution in the text-to-image synthesis landscape with the MIM method.

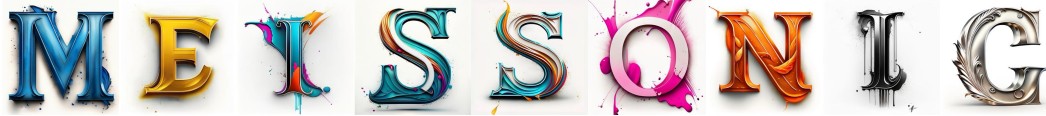

Figure 10: Zero-shot generation of stylized letters. Meissonic can synthesize individual letters to form the word "MEISSONIC." *Prompt*: A post featuring a [COLOR] '[LETTER]' painted on top.

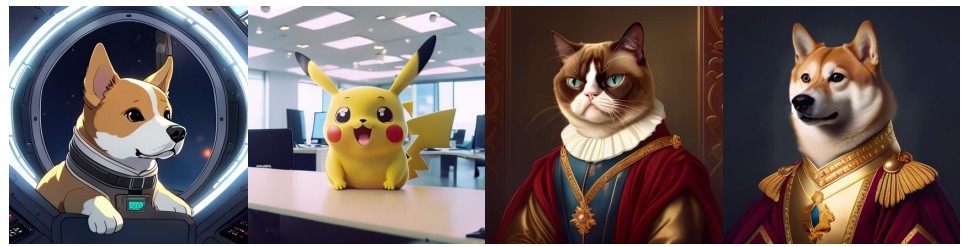

Figure 11: Memes generated by Meissonic.

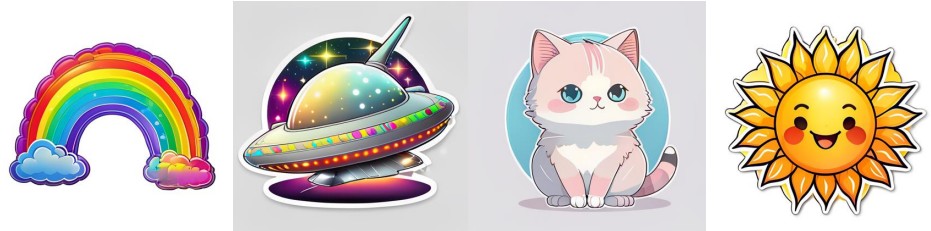

Figure 12: Cartoon Stickers generated by Meissonic.

## C    APPLICATIONS

We present the letter synthesis capability of Meissonic in Figure 10.

We present the combination capability of complex concepts of Meissonic in Figure 1.

We present meme generation in Figure 11.

We present cartoon sticker generation in Figure 12.

## D    PERFORMANCE COMPARISONS FOR COMPLEX VERSUS SIMPLE PROMPTS

We present performance comparisons for complex prompts versus simple prompts in Figure 13.

## E    PERFORMANCE COMPARISONS WITH DIFFERENT NUMBERS OF INFERENCE STEPS AND CLASSIFIER FREE GUIDANCE (CFG)

We present performance comparisons with different numbers of inference steps and Classifier Free Guidance (CFG) in Figure 14,15,16,17,18,19.

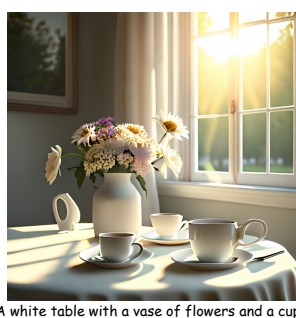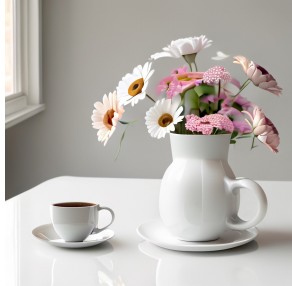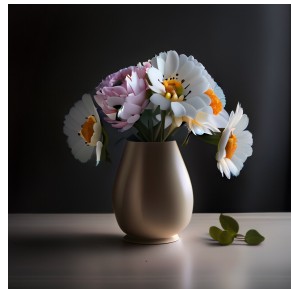

A white table with a vase of flowers and a cup of coffee on top of it, accompanied by a plate of buttery croissants, a folded linen napkin, and a faint ray of sunlight streaming through a nearby window in a cozy dining room.

A white table with a vase of flowers and a cup of coffee on top of it.

Table flowers.

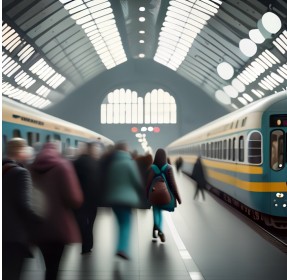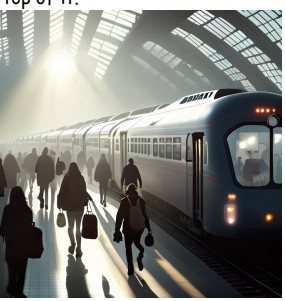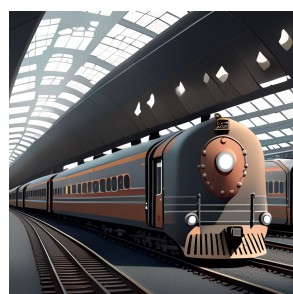

A busy train station with people hurrying along the platforms, some carrying luggage, while a sleek modern train is arriving, its headlights cutting through the slight morning haze, under a vast glass roof with beams of sunlight streaming in.

A busy train station with people hurrying along the platforms.

Train station.

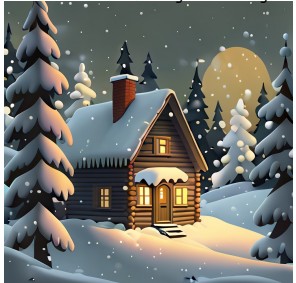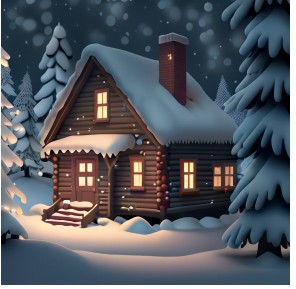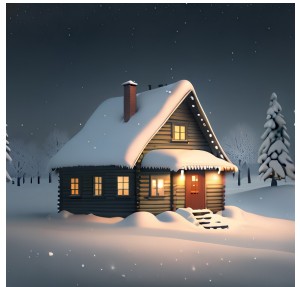

A cozy wooden cabin covered in a blanket of snow, with smoke rising from its chimney, surrounded by tall pine trees, as soft snowflakes fall from the gray sky, and a warm yellow glow from the windows invites you in.

A cozy wooden cabin covered in a blanket of snow.

Snow cabin.

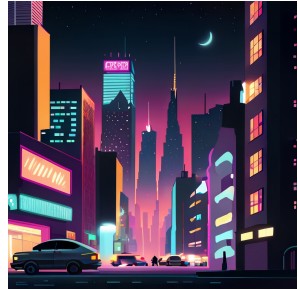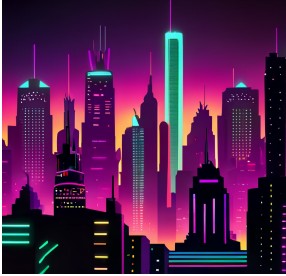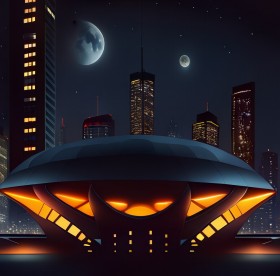

A vibrant city at night with skyscrapers illuminated by neon lights, busy streets filled with cars and people, and a towering billboard flashing colorful advertisements, while a clear night sky reveals the faint twinkle of distant stars.

A vibrant city at night with skyscrapers illuminated by neon lights.

Night city.

Figure 13: Performance Comparisons for Complex versus Simple Prompts

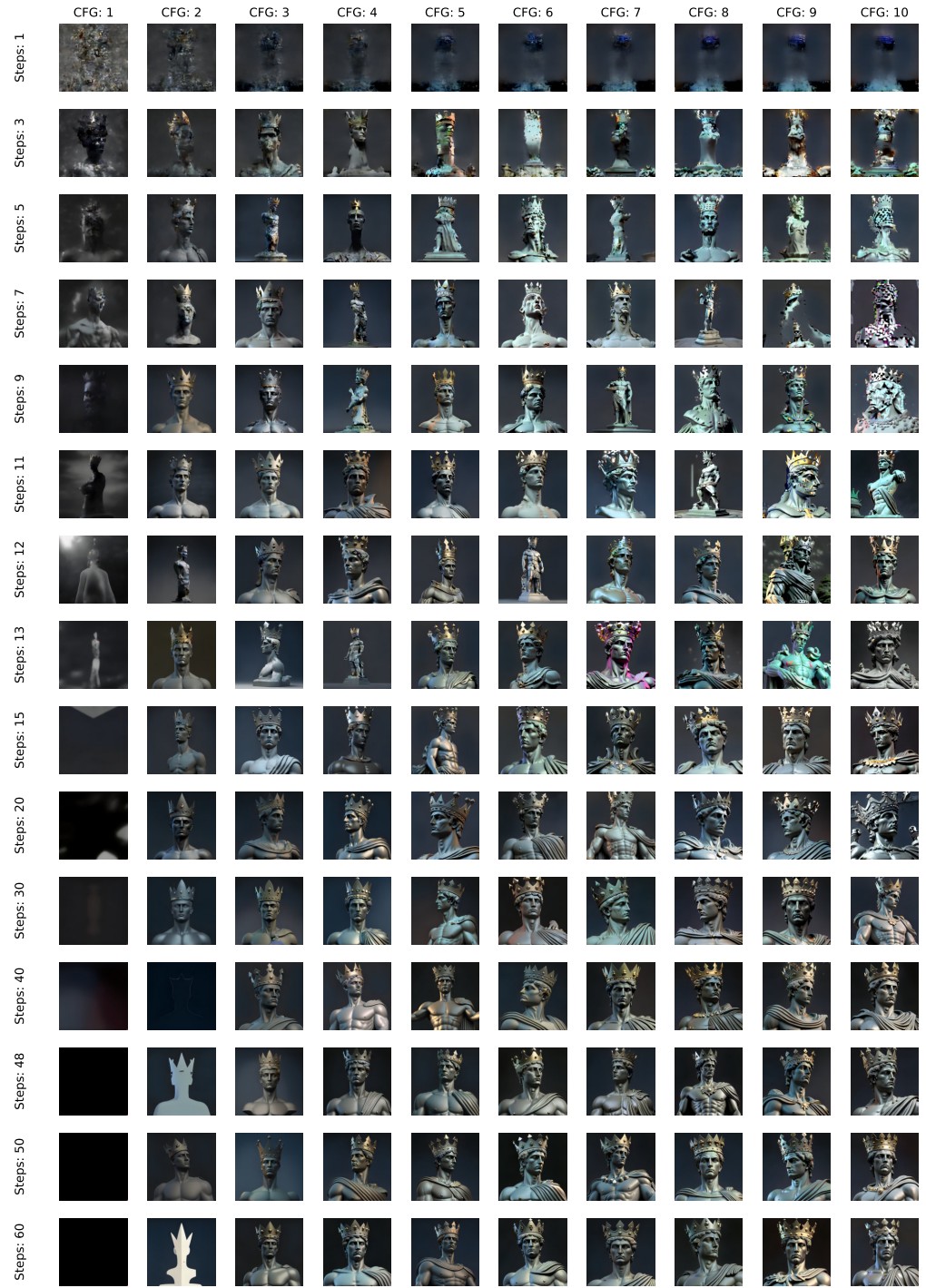

Figure 14: Performance Comparisons with Different Numbers of Inference Steps and Classifier Free Guidance (CFG). *Prompt*: A statue of a man with a crown on his head.

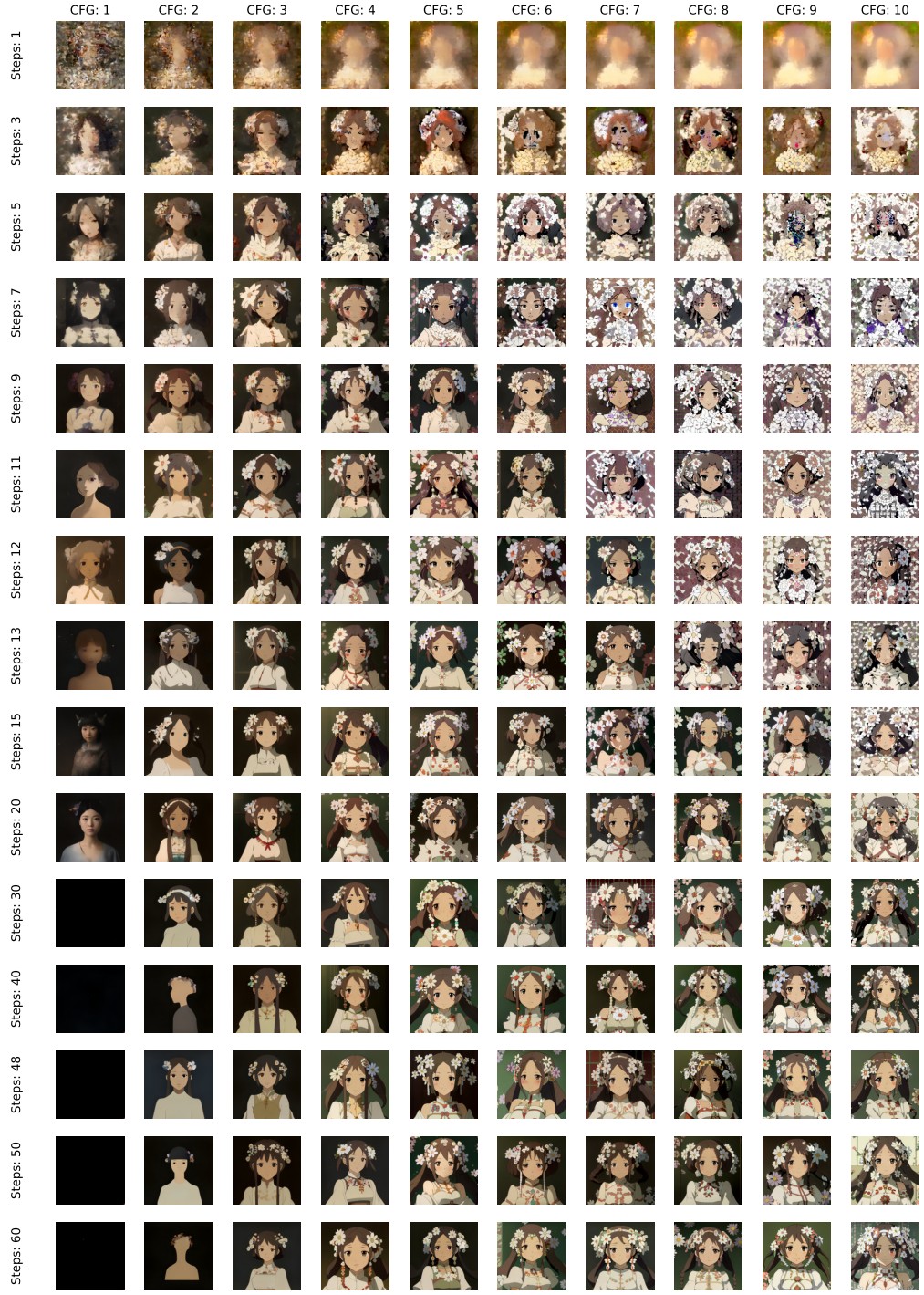

Figure 15: Performance Comparisons with Different Numbers of Inference Steps and Classifier Free Guidance (CFG). *Prompt*: Studio photo portrait of Lain Iwakura from Serial Experiments Lain wearing floral garlands over her traditional dress.

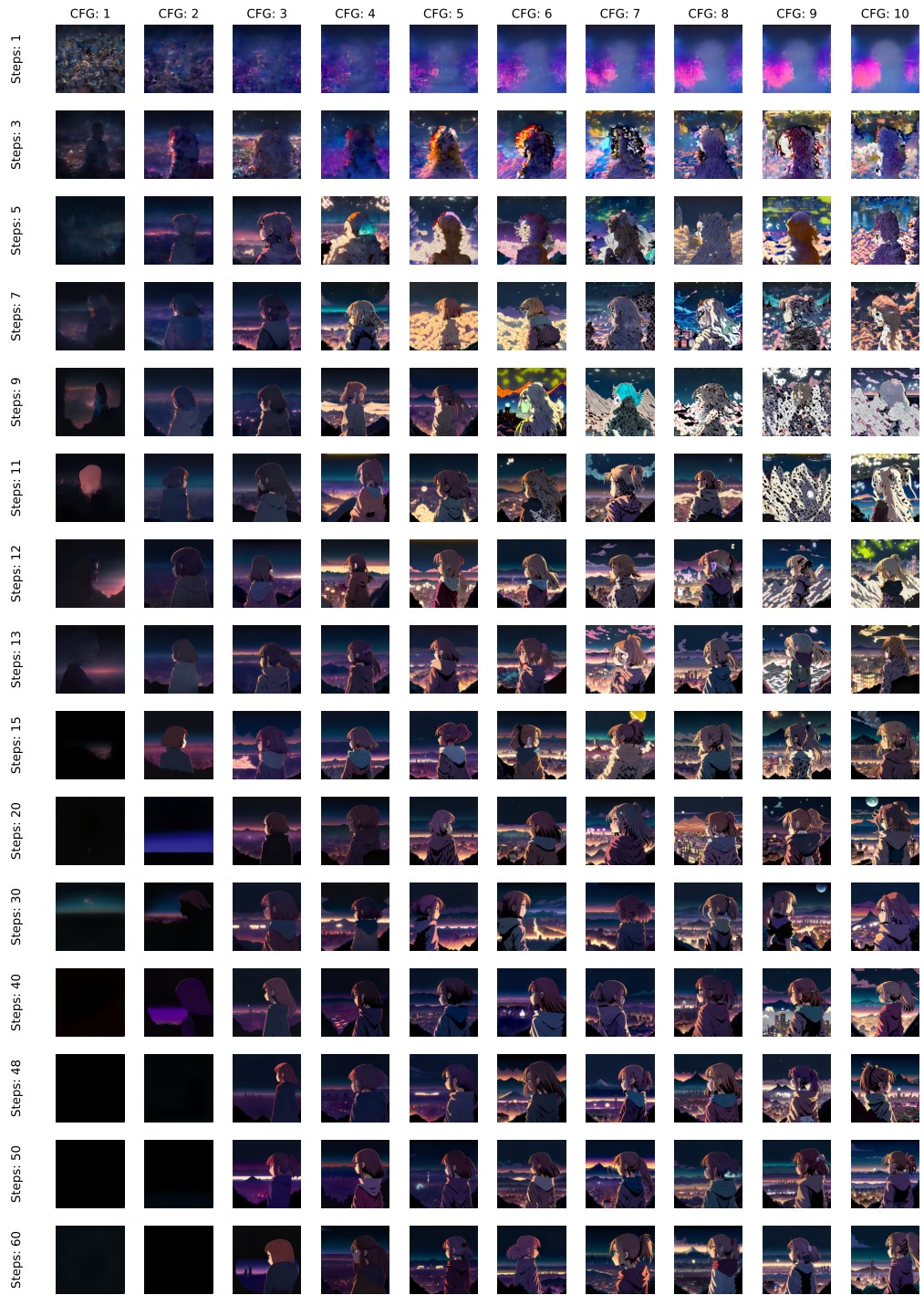

Figure 16: Performance Comparisons with Different Numbers of Inference Steps and Classifier Free Guidance (CFG). *Prompt*: A girl gazes at a city from a mountain at night in a colored manga illustration by Diego Facio.

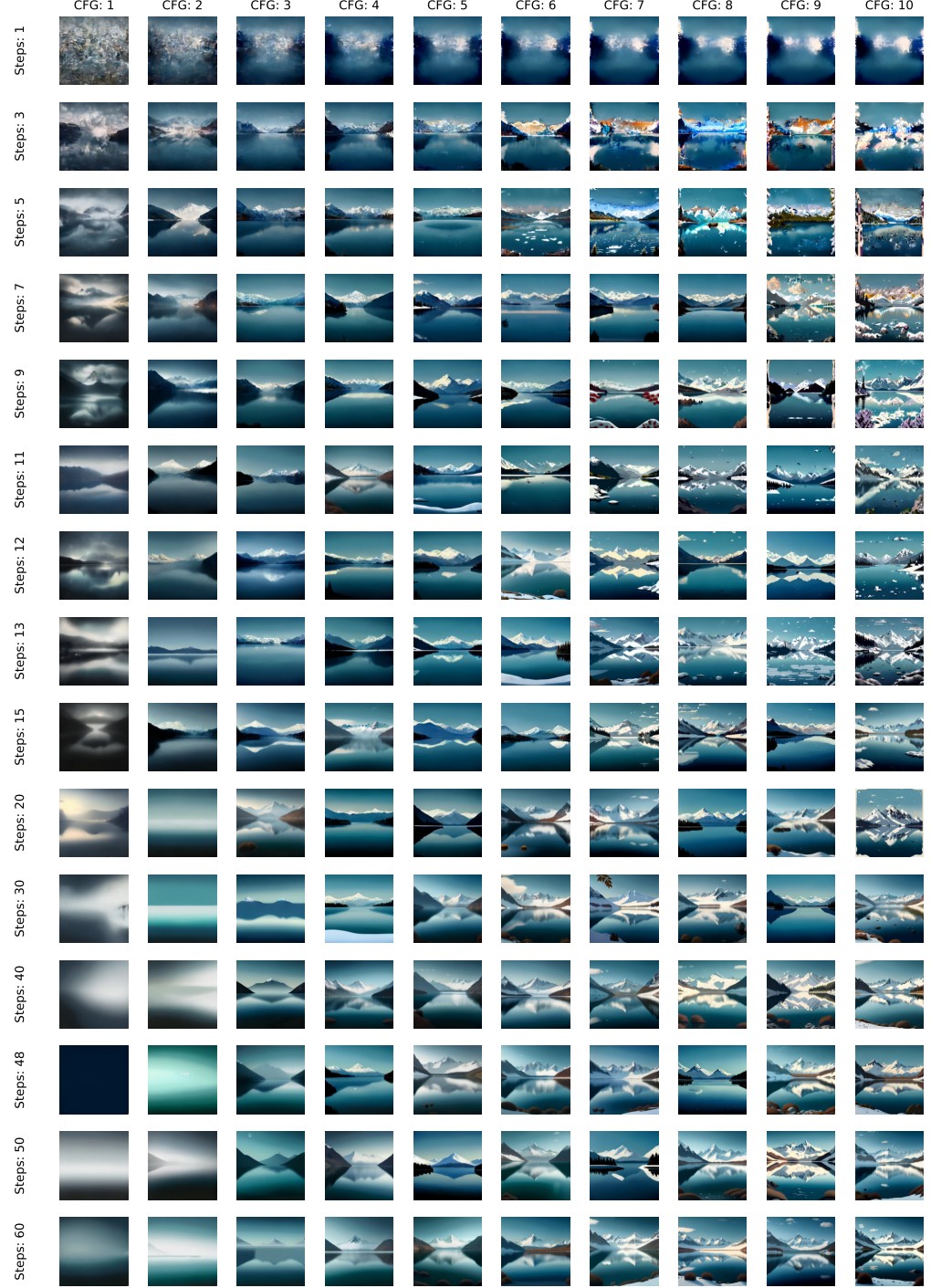

Figure 17: Performance Comparisons with Different Numbers of Inference Steps and Classifier Free Guidance (CFG). *Prompt*: A tranquil lake surrounded by snow-capped mountains under a clear sky.

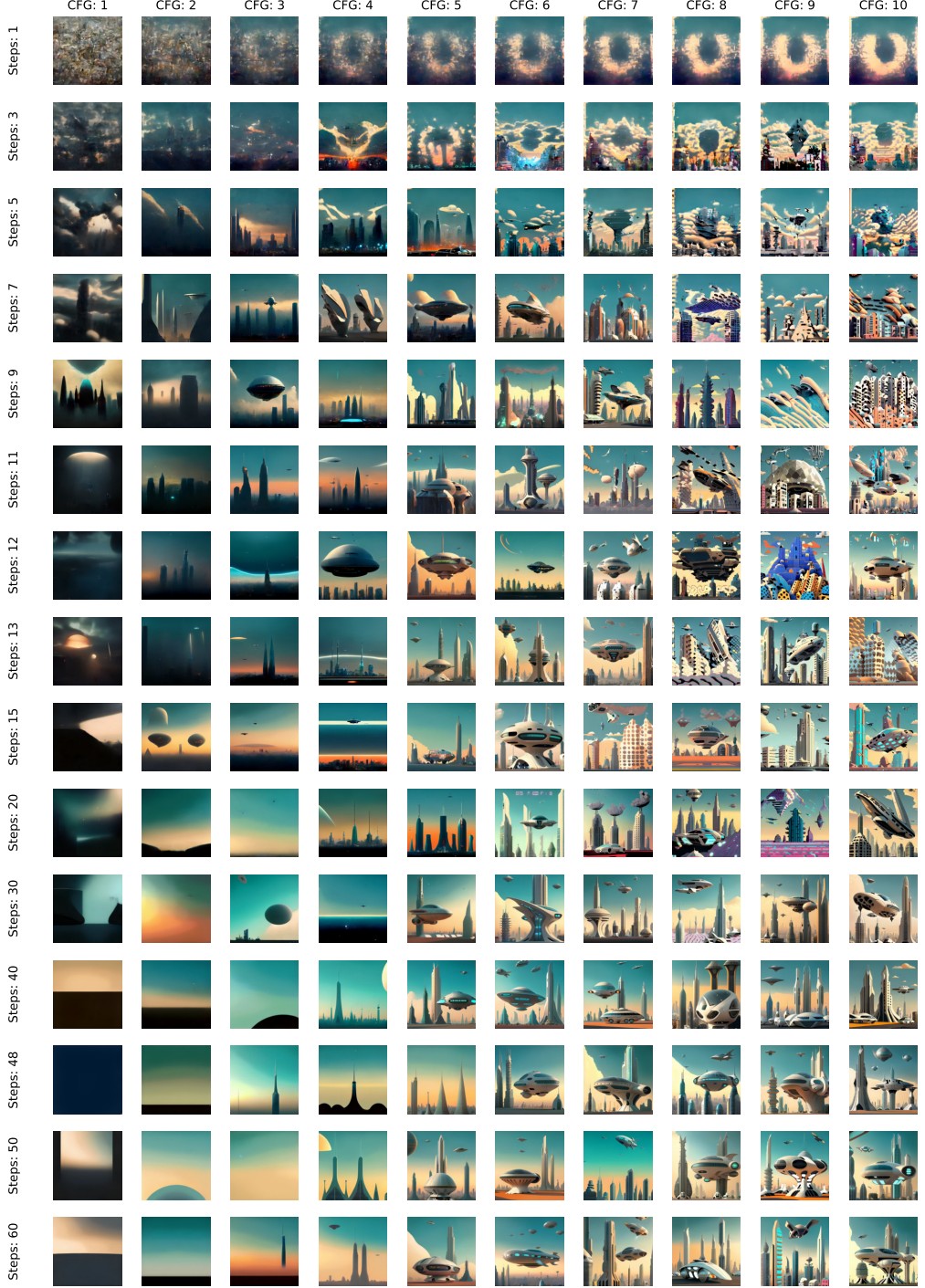

Figure 18: Performance Comparisons with Different Numbers of Inference Steps and Classifier Free Guidance (CFG). *Prompt*: A futuristic cityscape with hovering vehicles and towering structures.

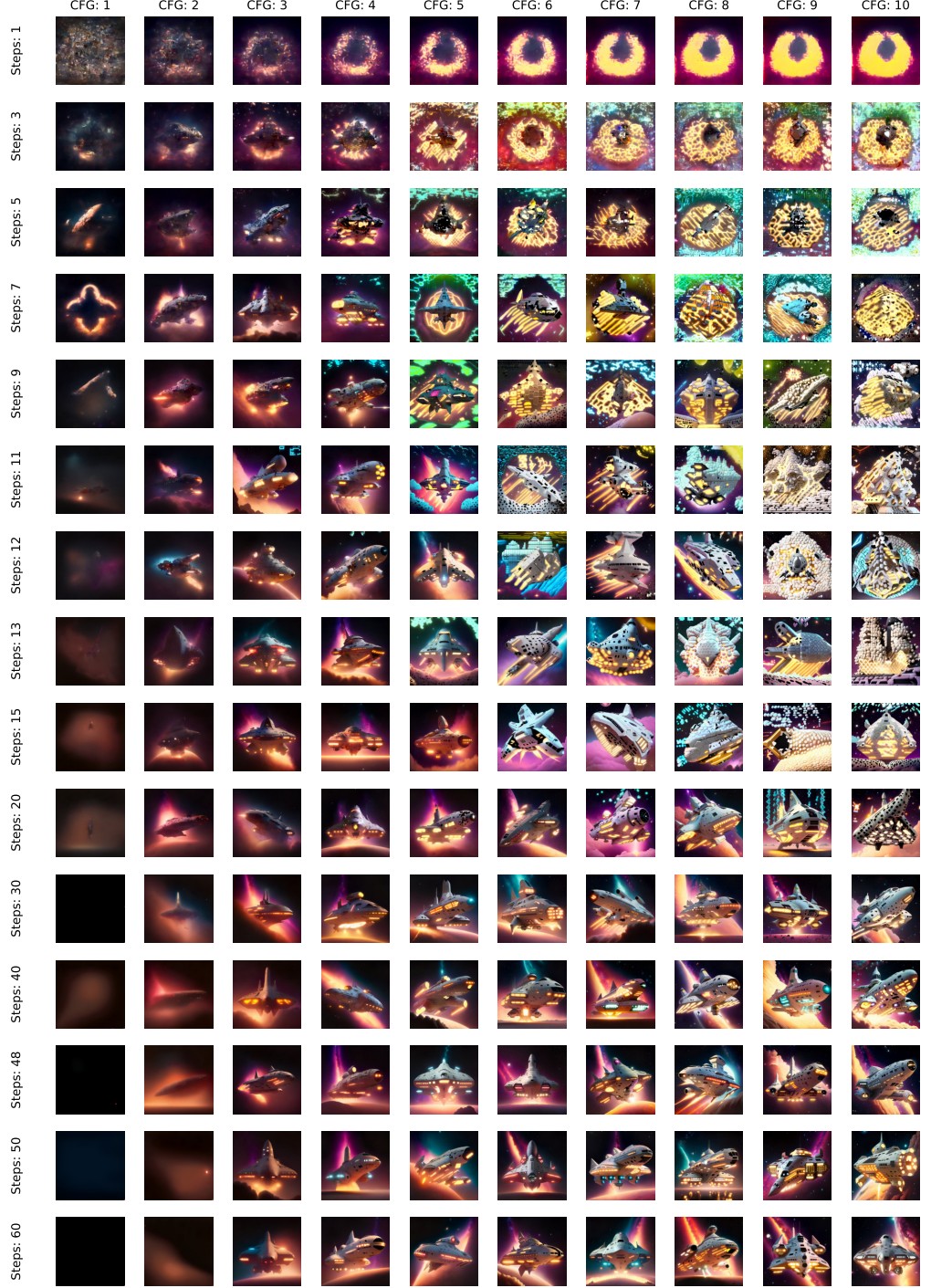

Figure 19: Performance Comparisons with Different Numbers of Inference Steps and Classifier Free Guidance (CFG). *Prompt*: A massive starship docked in a glowing nebula.

## F   MORE COMPARISONS FOR ZERO-SHOT IMAGE EDITING ABILITY

To ensure fair evaluations of zero-shot capabilities with SD1.5 and SDXL, we utilize Null-Text Inversion (Mokady et al., 2023) for zero-shot editing with our method, taking into account that other methods have been extensively trained on editing datasets. The configurations used for Null-Text Inversion, along with any undocumented parameters, align with those provided in the official code repository. The primary parameters are outlined as follows:

- `cross_replace_steps.default = 0.8`
- `self_replace_steps = 0.5`
- `blend_words = None`
- `equilizer_params = None`

For consistency, we used the recommended $512 \times 512$ resolution for editing and ran tests using `torch.float32`, which is the official setting for Null-Text Inversion.On A6000 GPUs (48 GB), the execution of MagicBrush (Zhang et al., 2024a) took approximately 36 hours for SD1.5 and 60 hours for SDXL. The runtime for Emu-Edit was significantly longer. Given the extensive computation, we randomly sampled 500 examples per benchmark for testing.

We present more comparisons for zero-shot image editing ability on EMU-Edit in Table 7.

|                       | CLIP-I↑ | CLIP-T↑ | DINO↑ | L1↓   | CLIPdir↑ |
|-----------------------|---------|---------|-------|-------|----------|
| SD 1.5 + Null-Text Inv. | 0.780   | 0.240   | 0.637 | 0.159 | 0.096    |
| SDXL + Null-Text Inv.   | 0.787   | 0.238   | 0.653 | 0.146 | 0.085    |
| Meissonic-512 (Ours)    | 0.791   | 0.244   | 0.689 | 0.128 | 0.102    |

Table 7: EMU-Edit Results

We present more comparisons for zero-shot image editing ability on MagicBrush in Table 8.

|                       | CLIP-I↑ | CLIP-T↑ | DINO↑ | L1↓   | CLIPdir↑ |
|-----------------------|---------|---------|-------|-------|----------|
| SD 1.5 + Null-Text Inv. | 0.824   | 0.228   | 0.647 | 0.121 | 0.106    |
| SDXL + Null-Text Inv.   | 0.840   | 0.241   | 0.665 | 0.122 | 0.111    |
| Meissonic-512 (Ours)    | 0.835   | 0.248   | 0.689 | 0.115 | 0.120    |

Table 8: MagicBrush Results

Our findings indicate that due to the inherent characteristics of MIM, Meissonic exhibits faster zero-shot editing capabilities. Performance was evaluated with `batch size = 1` and `inference step = 50` (compared to Null-Text Inv., which requires 500 backpropagation steps). Tests were conducted on an A6000 GPU with 48 GB VRAM.

Besides, we present inference time comparision in Table 9.

|                  | SD 1.5 + Null-Text Inv. | SDXL + Null-Text Inv. | Meissonic-512 (Ours) |
|------------------|-------------------------|-----------------------|----------------------|
| Time (s/10 pairs) | 1040 + 100              | 1850 + 120            | 108                  |
| GPU (GB)          | 13.4                    | 26.8                  | 5.9                  |

Table 9: Inference Time Comparison

These results demonstrate the substantial potential for reduced processing time with Meissonic.

We also present qualitative comparisons on zero-shot image editing ability in Figure 20.

## G   MORE COMPARISONS WITH SDXL FOR IMAGE GENERATION ABILITY

We present more comparisons with SDXL for image generation ability in Figure 21,22,23.

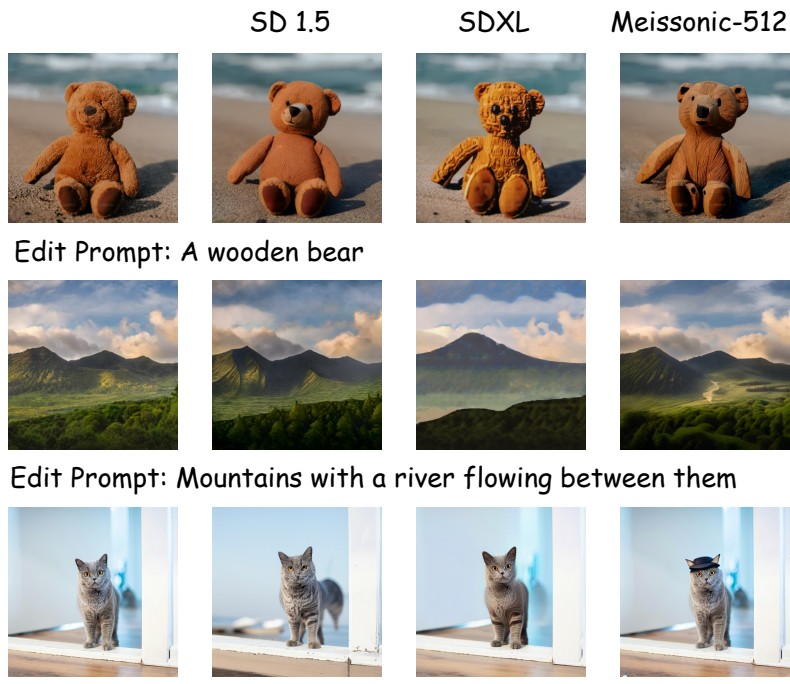

Figure 20: Qualitative comparisons on zero-shot image editing ability.

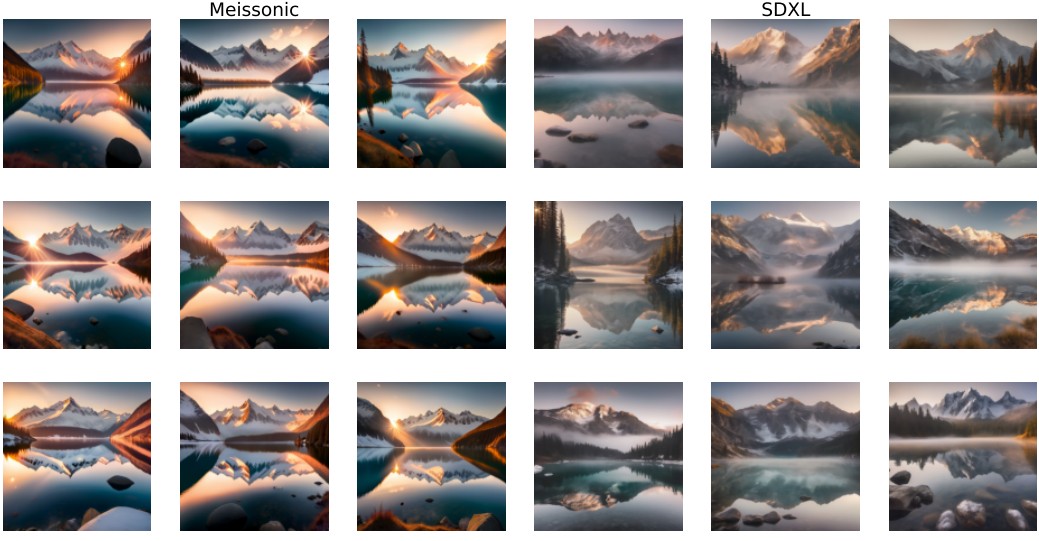

Figure 21: Qualitative comparisons with SDXL for image generation ability. *Prompt*: A breathtaking photo of a serene mountain lake at sunrise, crystal-clear water reflecting the surrounding snow-capped peaks, with a soft mist floating above the surface.

## H    ABLATION STUDY

**Detailed roadmap to build Meissonic.** We present ablation studies during training Meissonic-512 in Table. 24. The HPS v2.1 (Wu et al., 2023) scores are calculated for verifying the effectiveness of

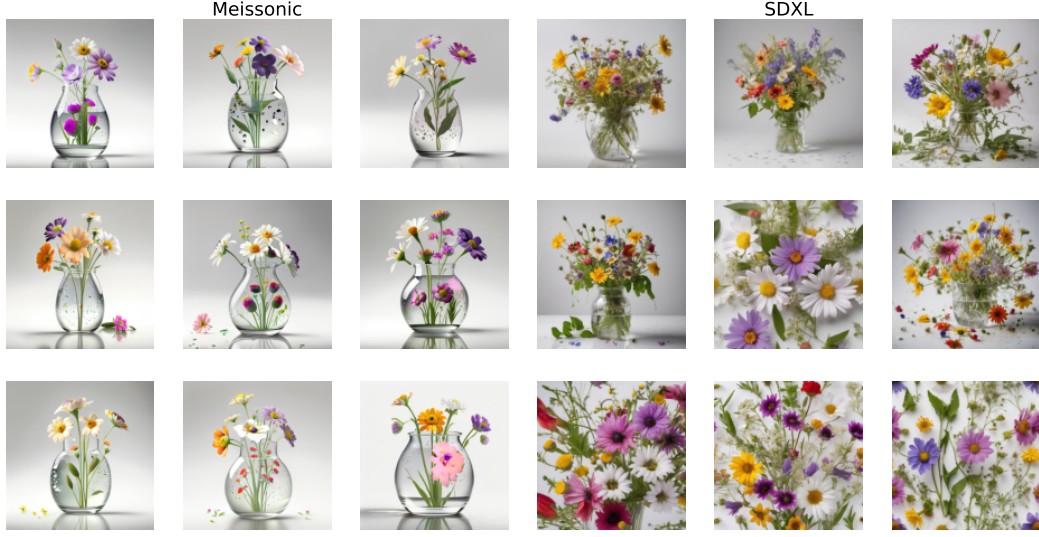

Figure 22: Qualitative comparisons with SDXL for image generation ability. *Prompt*: A professional studio photograph of a fresh bouquet of wildflowers in a glass vase, water droplets visible on the petals and leaves, placed on a clean white background.

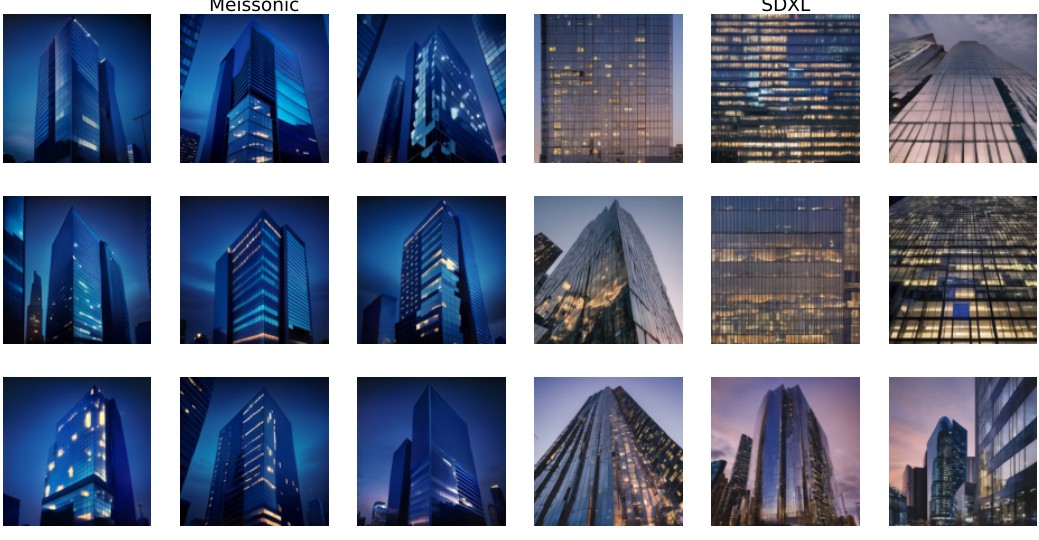

Figure 23: Qualitative comparisons with SDXL for image generation ability. *Prompt*: A sharp photo of a modern skyscraper during blue hour, its glass facade reflecting the city lights and the deep indigo sky in the background.

each compoment. Our ablations are based on training stage 2, ensuring consistency with the training dataset scale, model scale, and other training configurations.

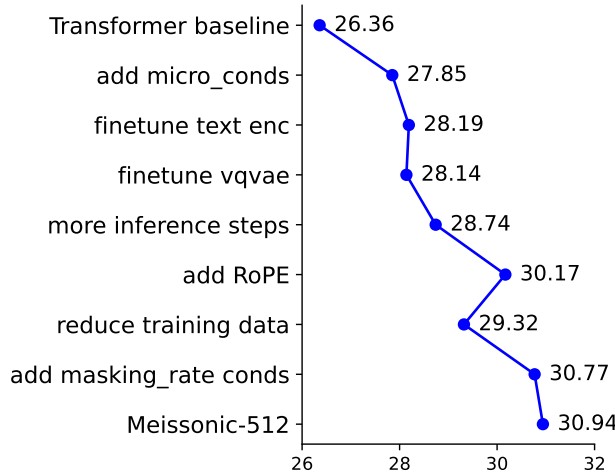

Figure 24: HPS v2.1 Score on internal 1000 prompts

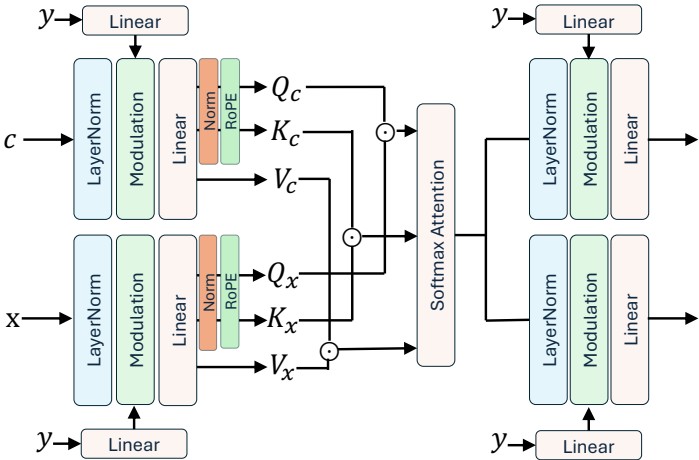

Figure 25: Multi-modal Transformer For MIM.

## I OUR MULTIMODAL TRANSFORMER BLOCK

We present a detailed structure of our Multi-modal Transformer Block for MIM in Figure 25. Specifically, $x$ denotes image embedding inputs, $c$ denotes text embedding inputs, and $y$ denotes conditions inputs.

## J WORD CLOUD OF OUR REALUSER800 BENCHMARK

We present a word cloud image that illustrates the diverse concepts, styles, and themes encompassed within our RealUser-800 prompts benchmark in Figure 26.

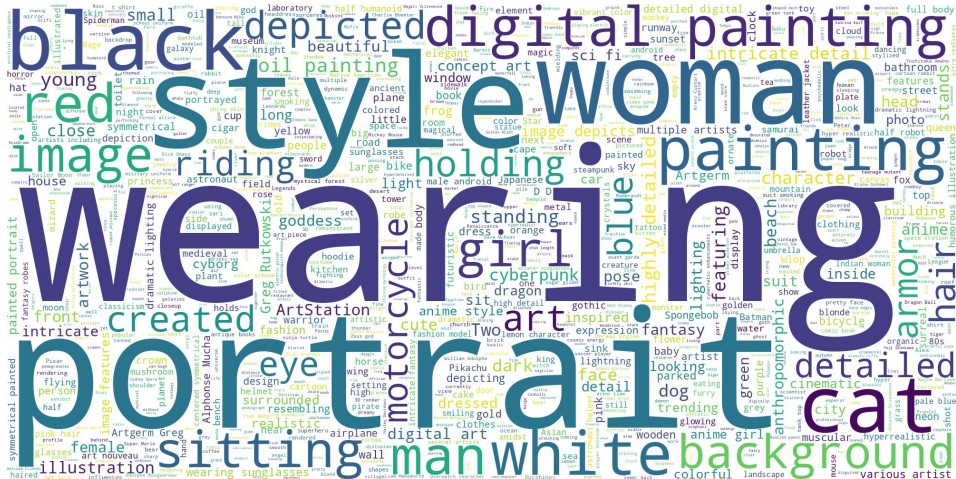

Figure 26: Word cloud image of our RealUser-800 prompts benchmark.

## K  IMAGES GENERATED DURING DIFFERENT TRAINING STAGES

We present images generated using the same prompt across Meissonic's four training stages in Figure 27.

## L  MORE EXAMPLES OF QUALITATIVE COMPARISONS

We present more examples of qualitative comparisons in Figure 28.

## M  MORE IMAGES PRODUCED BY MEISSONIC

We present additional images generated by Meissonic using CC3M (Sharma et al., 2018) items, with detailed captions provided by VILA-1.5 (Lin et al., 2023) and Morph (Pan et al., 2024). These images can be found in Figure 29,30,31,32,33,34,35,36,37,38,39,40,41,42,43,44,45,46,47.

We present additional images generated by Meissonic using HPS (Wu et al., 2023) benchmark prompts. These images can be found in Figure 48,49,50,51,52,53.

## N  MORE IMAGES PRODUCED BY MEISSONIC AT DIVERSE RESOLUTIONS

We present additional images generated by Meissonic at diverse resolutions. These images can be found in Figure 54,55.

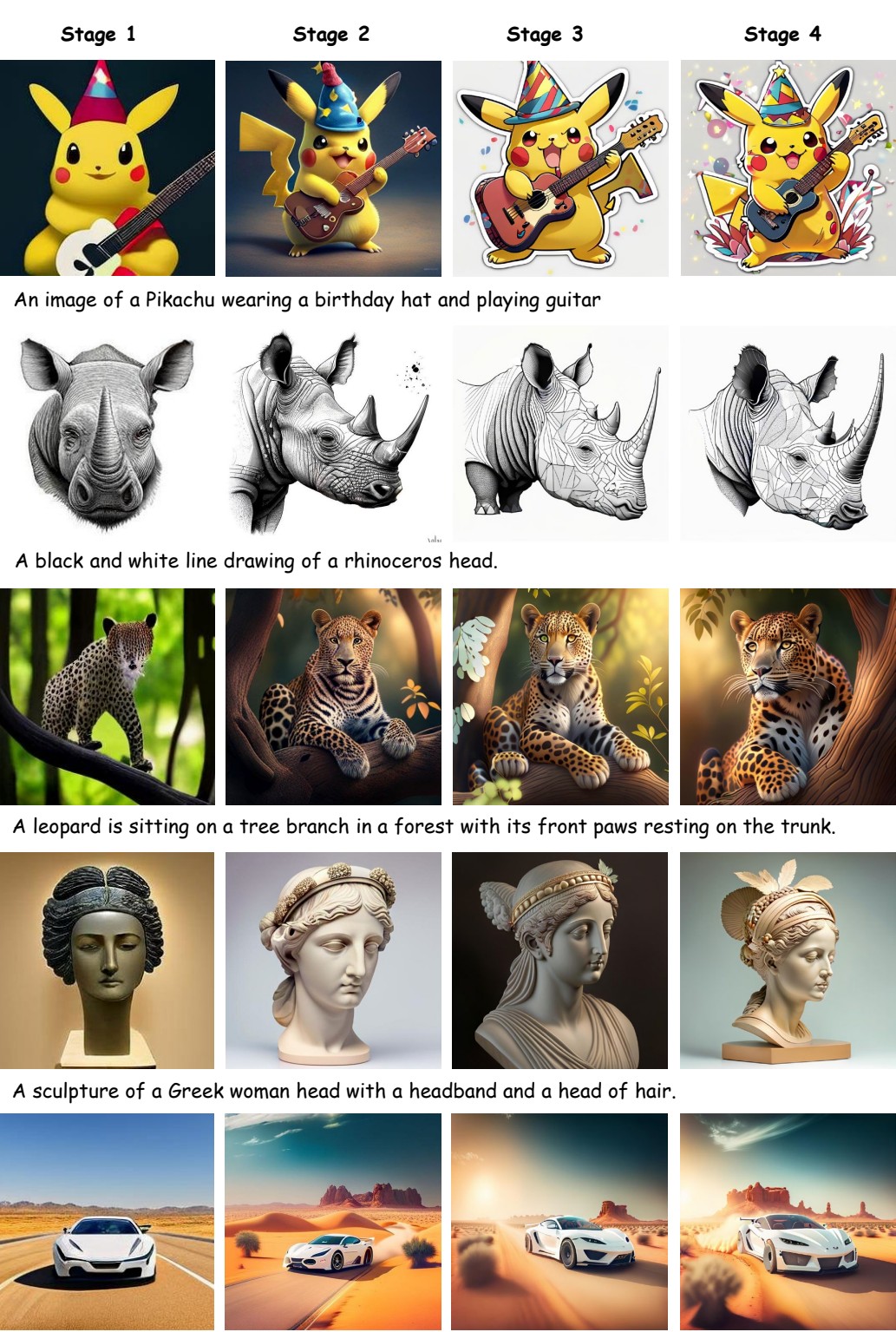

Figure 27: Images generated using the same prompt across Meissonic's four training stages. The resolutions for stages 1 and 2 are $256^2$ and $512^2$, respectively, while stages 3 and 4 are $1024^2$. For clarity and comparison, all images are displayed in a consistent layout.

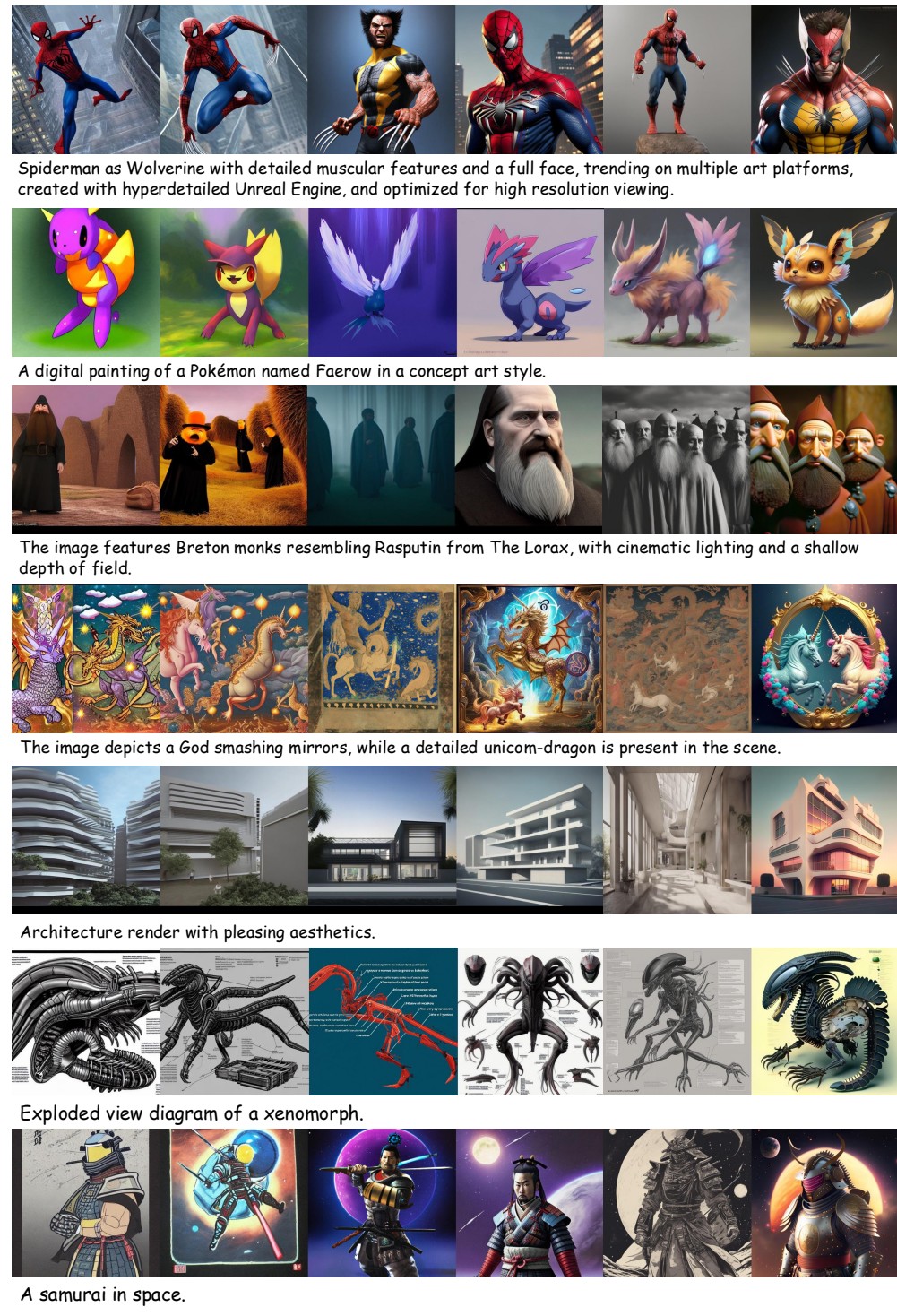

Figure 28: Qualitative Comparisons with SD 1.5, SD 2.1, DeepFloyd-XL, Deliberate, and SDXL.

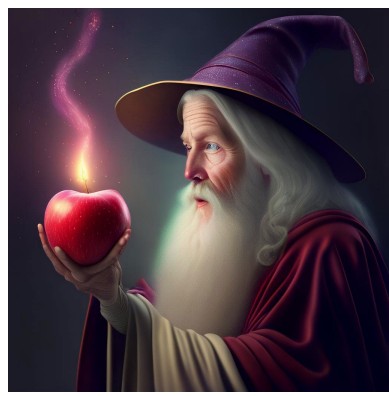

The wizard chants a spell over the apple

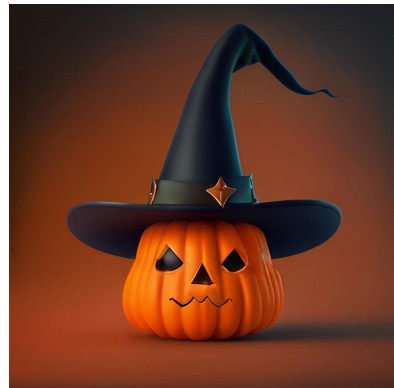

Pumpkin head wearing black wizard hat

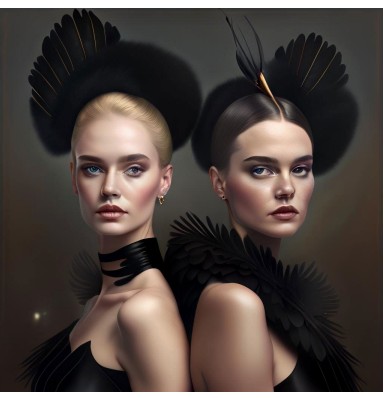

Two women in black dresses with feathers on their heads.

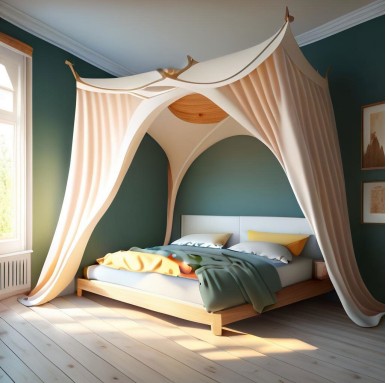

A bedroom with a canopy bed and a wooden floor

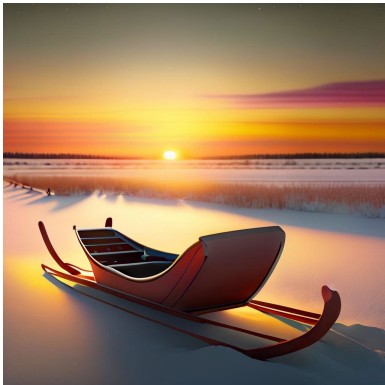

A sled sits in a field with a sunset in the background.

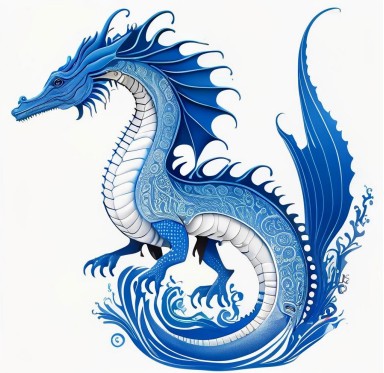

A blue and white drawing of a sea dragon.

Figure 29: High Quality Samples Produced by Meissonic.

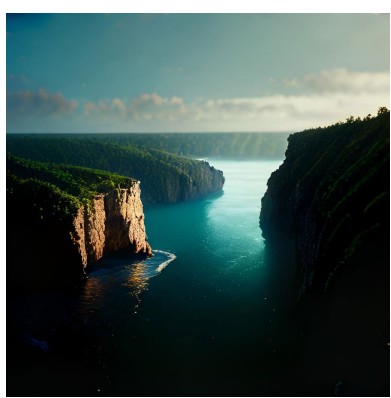

A body of water with a cliff in the background

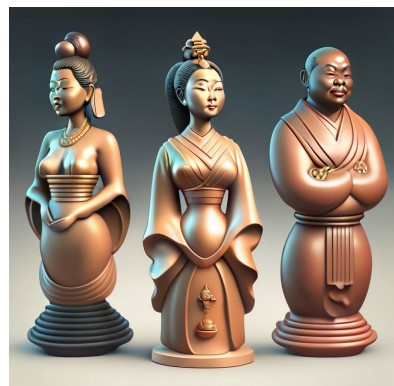

A collection of statues of Asian men and women.

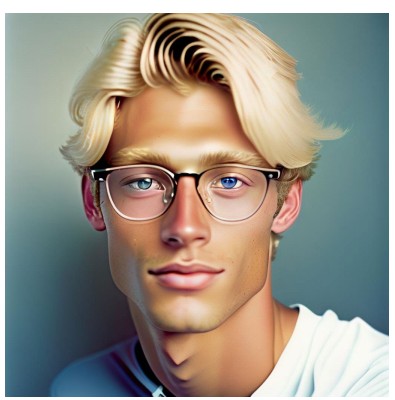

A man with blonde hair and glasses is looking at the camera.

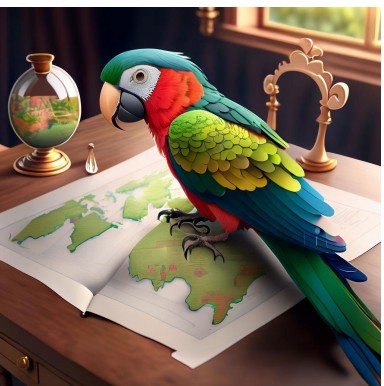

A table with a parrot on it and a map on it.

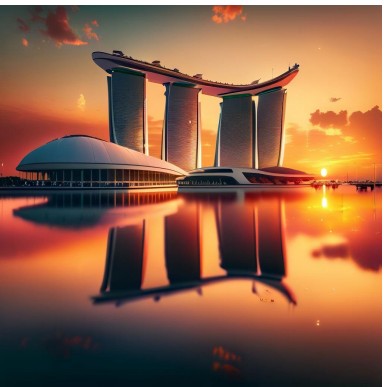

A beautiful sunset with a reflection of the Marina Bay Sands hotel.

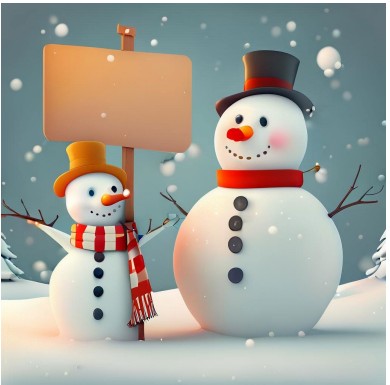

Two snowmen are standing next to a snowman with a blank sign.

Figure 30: High Quality Samples Produced by Meissonic.

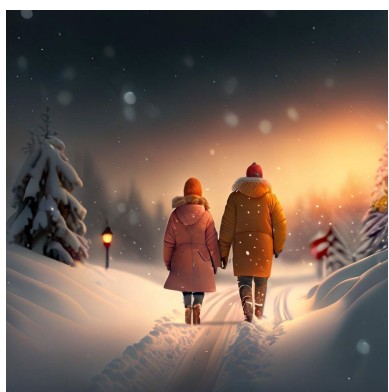

Two people walking in the snow with a sled.

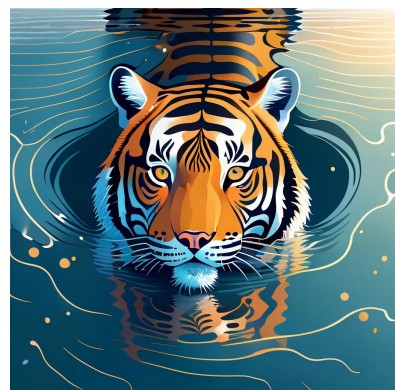

A tiger is swimming in a body of water.

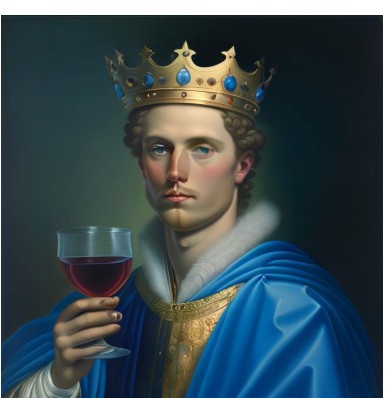

A man with a crown and a blue robe is holding a glass.

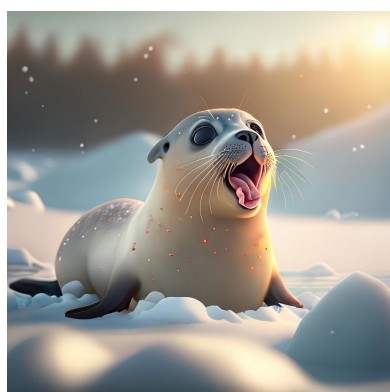

A seal is sitting in the snow with its mouth open.

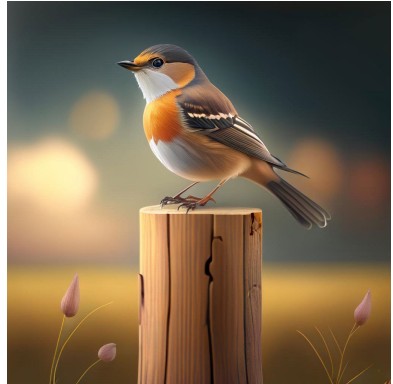

A small bird is perched on a wooden post.

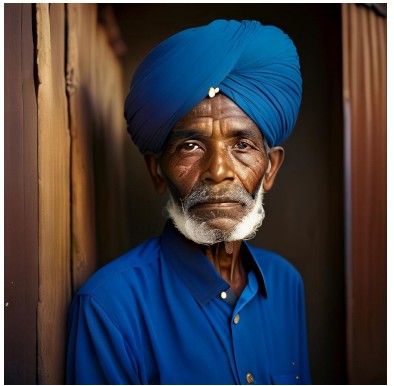

An old man with a blue turban and a blue shirt is standing in front of a wooden wall.

Figure 31: High Quality Samples Produced by Meissonic.

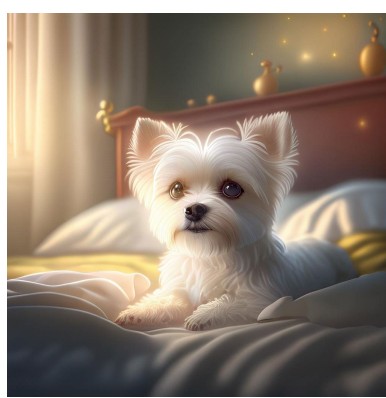

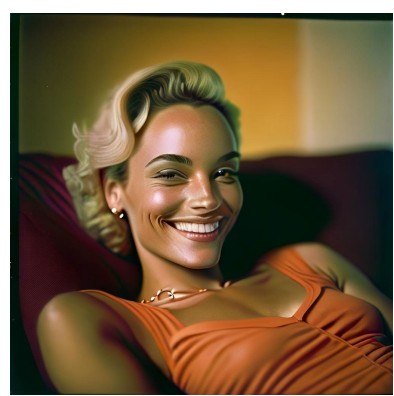

A small cute white dog is sitting on a bed.

A woman is laying on a couch and smiling.

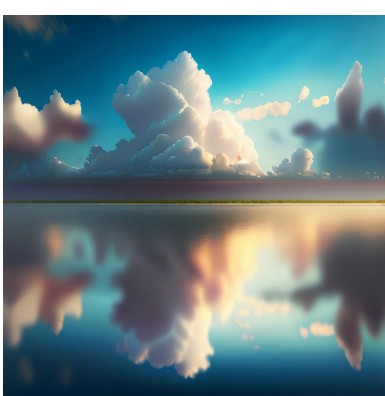

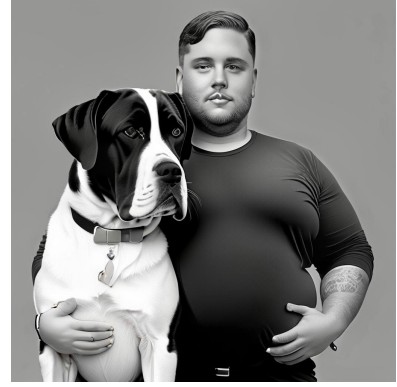

A cloudy sky over a body of water.

A fat man is holding a large black and white dog in a black-white figure style.

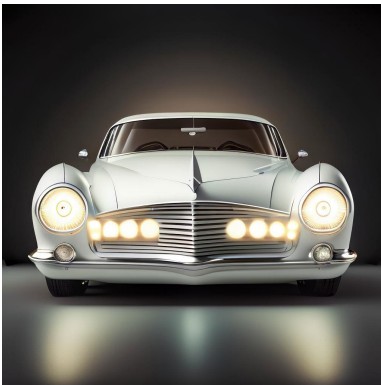

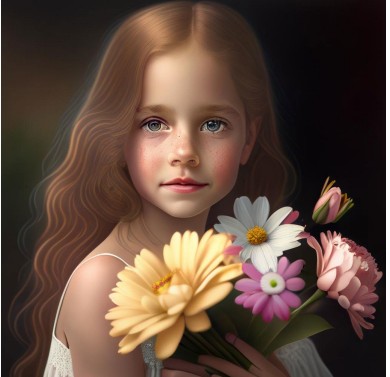

A white car with a silver rim and a headlight.

A young girl is holding a bouquet of flowers.

Figure 32: High Quality Samples Produced by Meissonic.

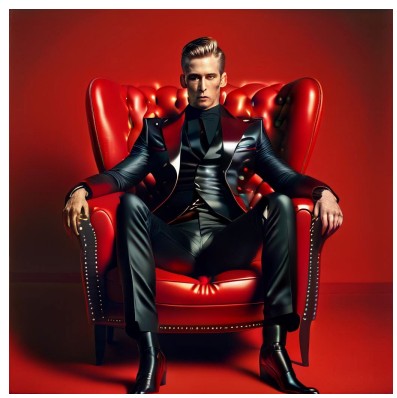

A man in a black leather suit sits in a red chair.

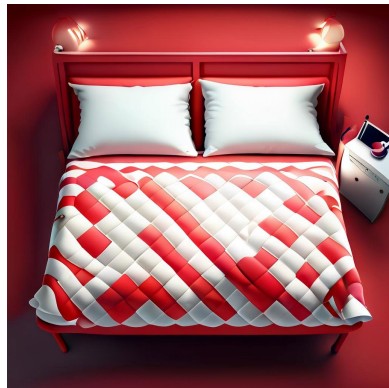

A bed with a red and white quilt on it.

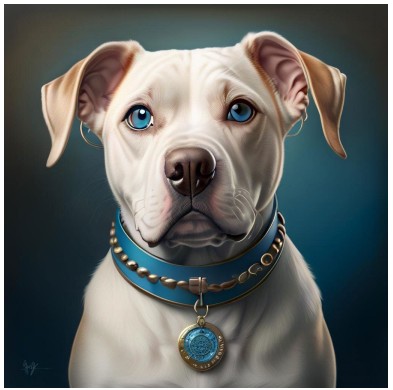

A dog with a blue collar is looking at the camera.

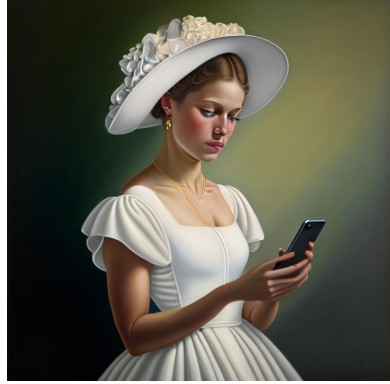

A woman in a white dress is looking at her phone.

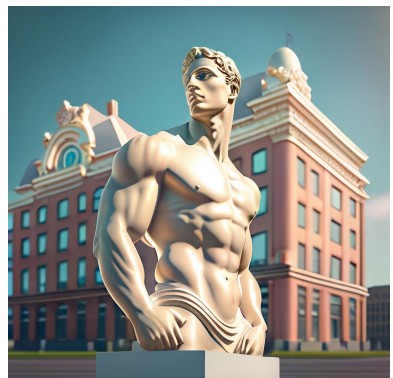

A statue of a man in front of a building.

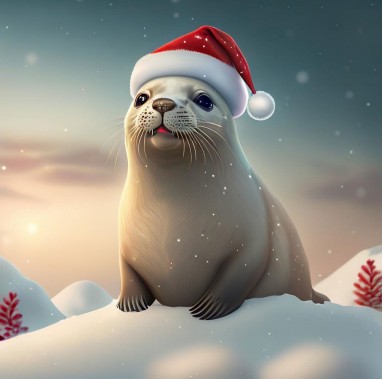

A seal is wearing a Santa hat and is on a snowy hill with the words Happy New Year written below it.

Figure 33: High Quality Samples Produced by Meissonic.

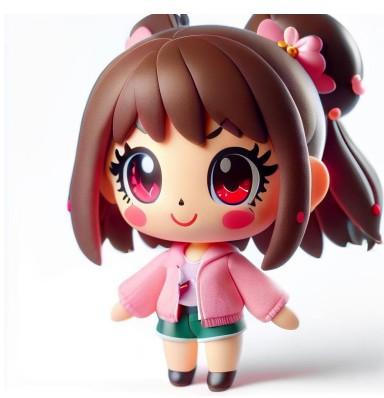

A plush toy of a girl with red eyes and a pink shirt.

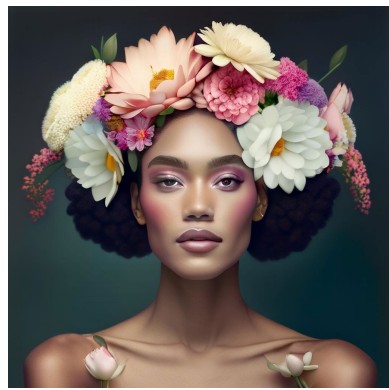

A woman with a flower crown on her head.

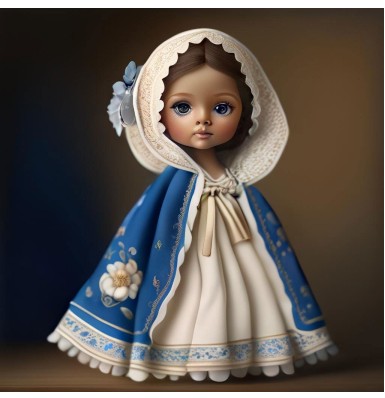

A doll wearing a blue and white dress and a tan shawl.

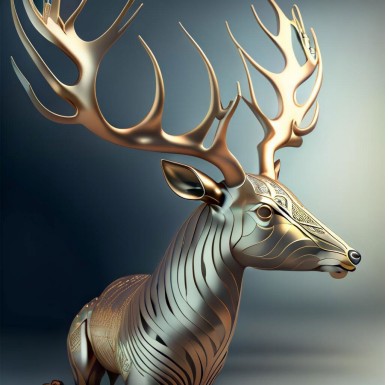

A metal sculpture of a deer with antlers.

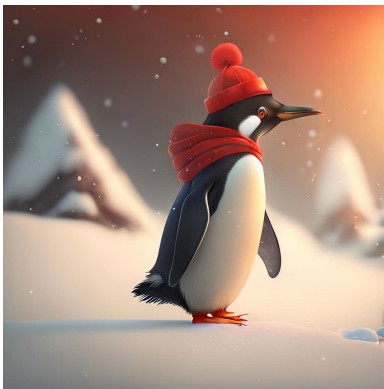

A penguin walks in the snow with a red hat on.

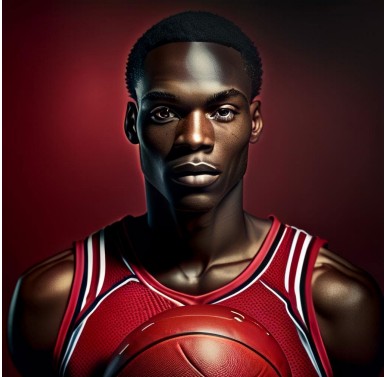

A man in a red jersey holding a basketball.

Figure 34: High Quality Samples Produced by Meissonic.

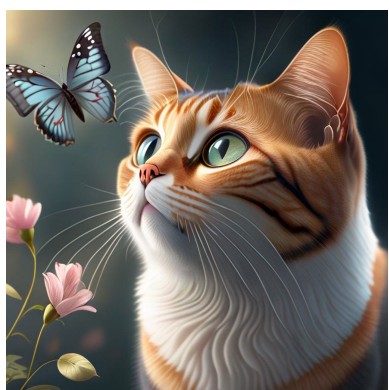

A cat is looking at a butterfly

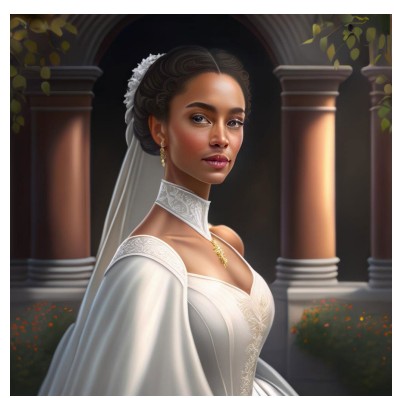

A woman in a white wedding dress stands in a courtyard.

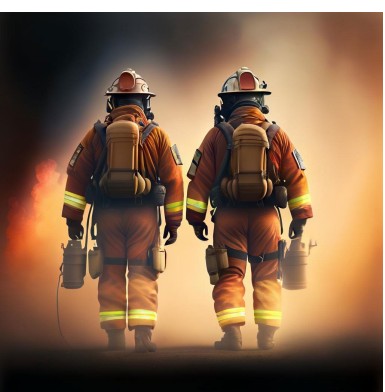

Two firefighters standing in front of a smoky background.

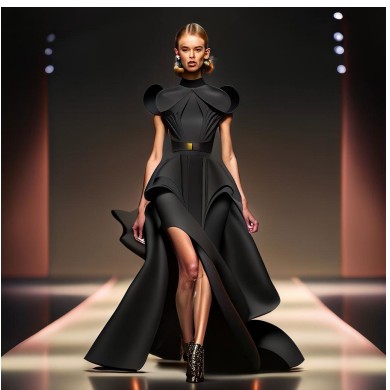

A model walks down a runway in a black dress.

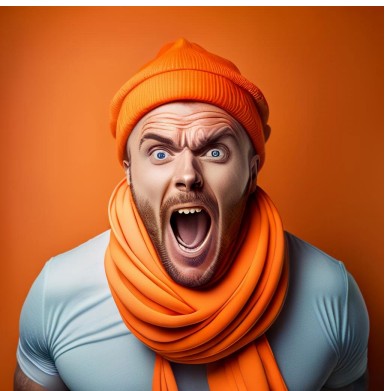

A man wearing an orange hat and scarf is screaming

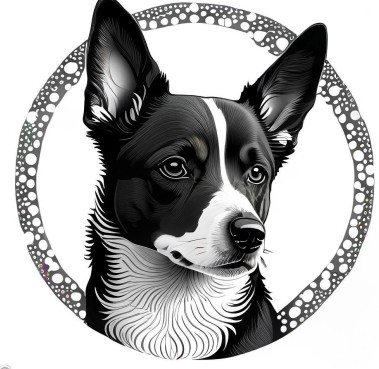

A black and white drawing of a dog's head in a circle.

Figure 35: High Quality Samples Produced by Meissonic.

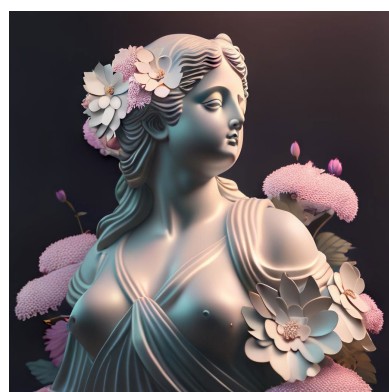

A statue of a woman surrounded by flowers

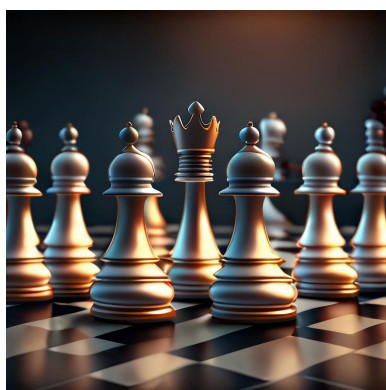

A chess board with a row of chess pieces.

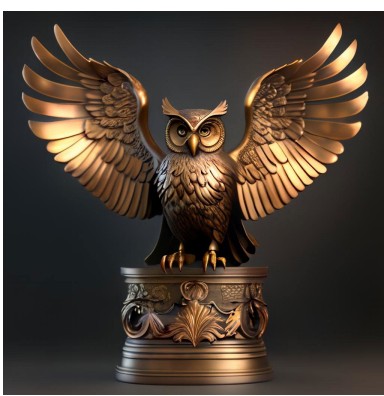

A bronze statue of an owl with its wings spread.

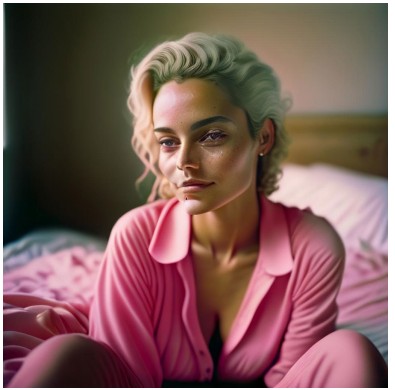

A woman in a pink shirt is sitting on a bed.

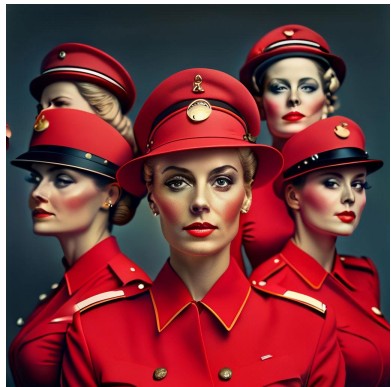

A group of women in red uniforms pose for a picture.

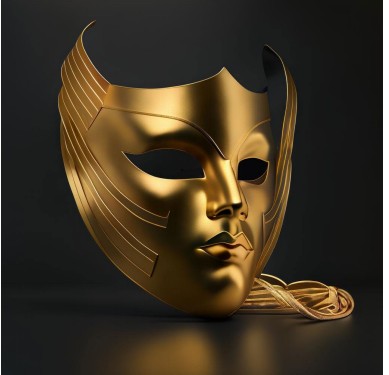

A gold mask with a gold strap is on a black surface.

Figure 36: High Quality Samples Produced by Meissonic.

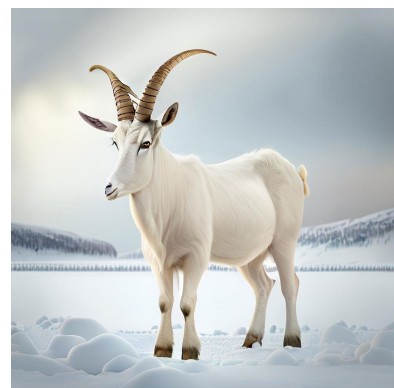

A white goat with horns is standing in the snow.

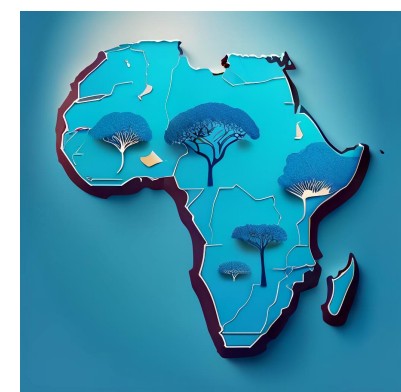

A map of Africa with a blue background.

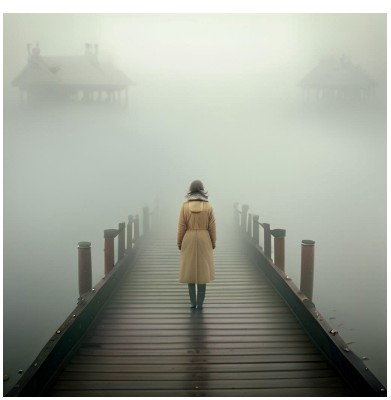

A woman stands on a dock in the fog.

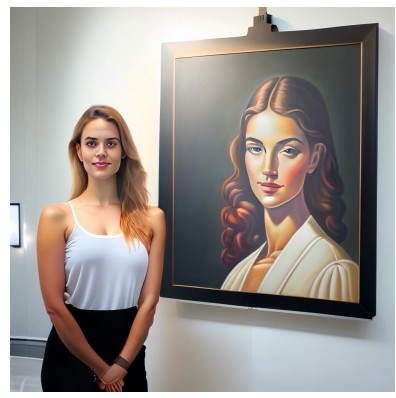

A woman is standing next to a picture of another woman.

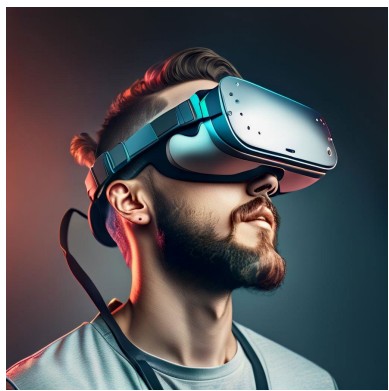

A man wearing a virtual reality headset.

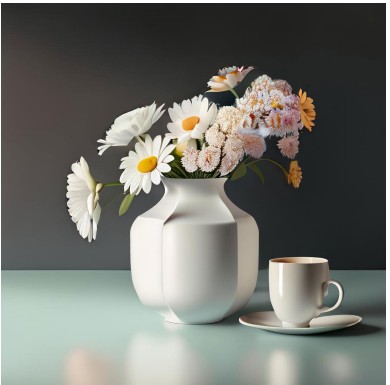

A white table with a vase of flowers and a cup of coffee on top of it.

Figure 37: High Quality Samples Produced by Meissonic.

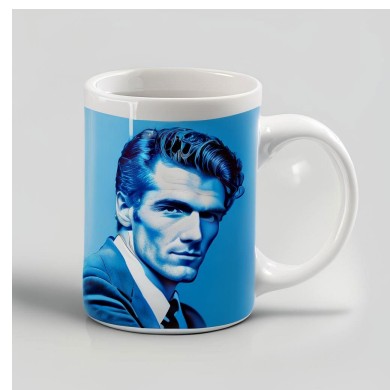

A white and blue coffee mug with a picture of a man on it.

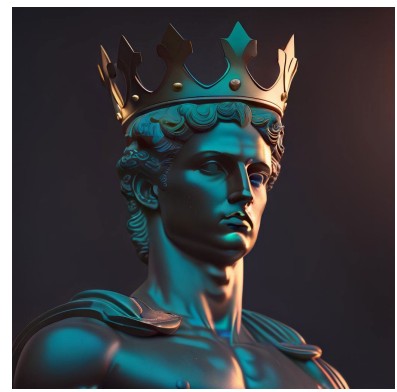

A statue of a man with a crown on his head.

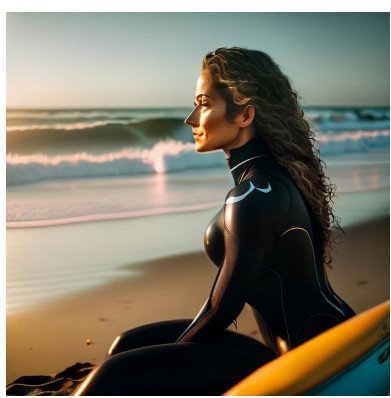

A woman in a black wetsuit sits on a bench gazing at the sea on the beach.

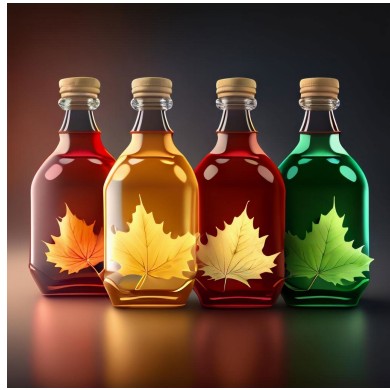

Four bottles of maple syrup in different colors.

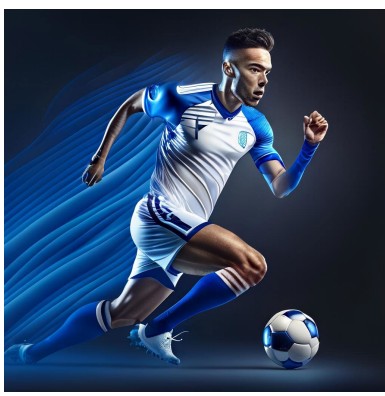

A soccer player in a blue and white uniform runs with the ball.

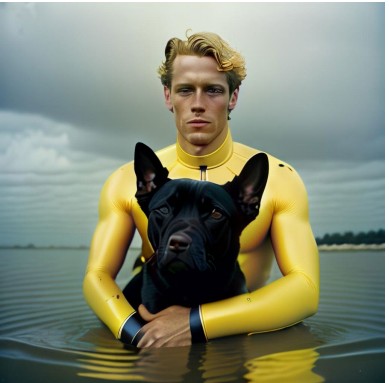

A man in a yellow wet suit is holding a big black dog in the water.

Figure 38: High Quality Samples Produced by Meissonic.

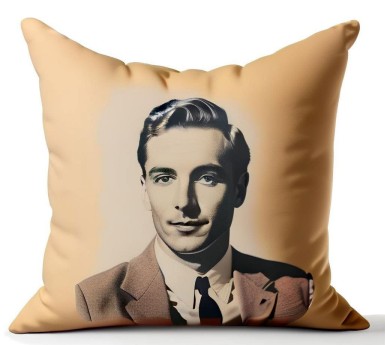

A pillow with a picture of a man on it.

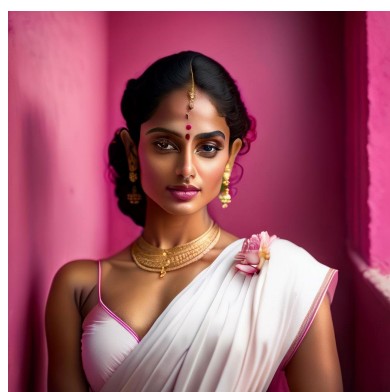

An Indian woman is wearing a white saree and standing in front of a pink wall.

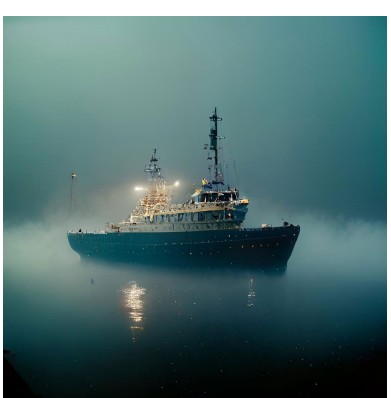

A large ship is in the water with a foggy background.

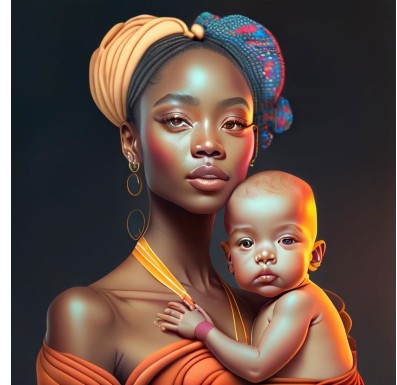

A woman holding a baby.

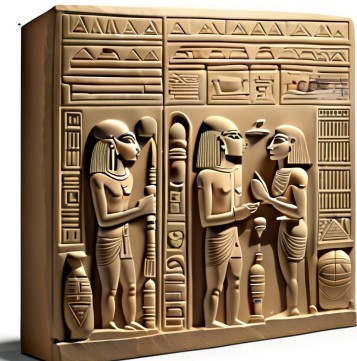

An ancient Egyptian carved stone wall with three figures and hieroglyphics.

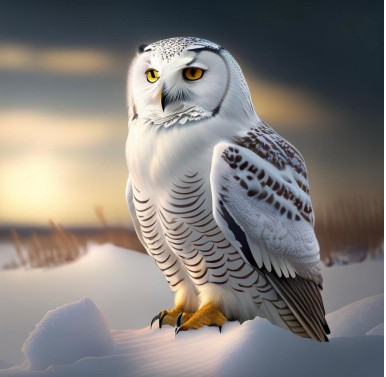

A snowy owl is sitting in the snow.

Figure 39: High Quality Samples Produced by Meissonic.

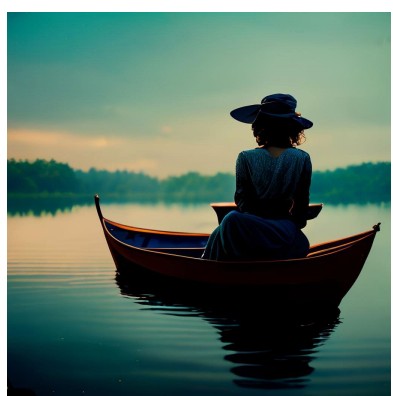

A woman is sitting on a boat and looking at a boat in the water.

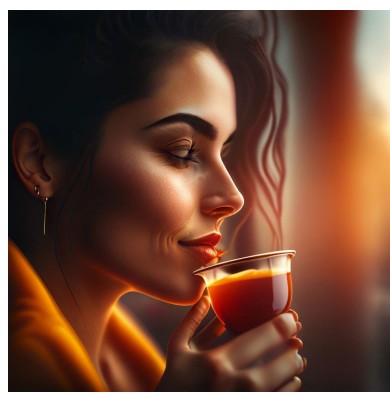

A woman drinking from a cup with a blurry background.

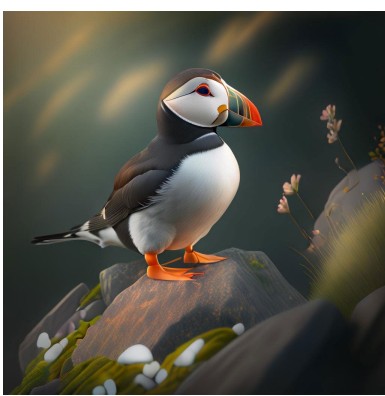

A puffin is sitting on a rock and looking off into the distance.

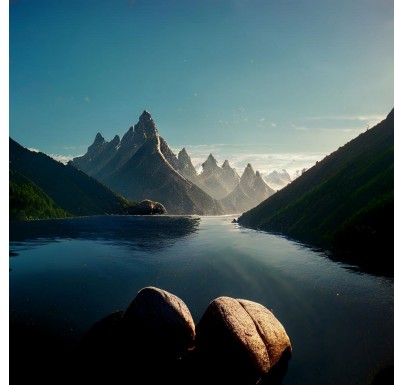

A large body of water with a rock in the middle and mountains in the background.

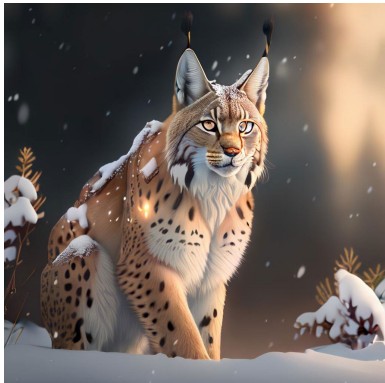

A lynx is standing in the snow.

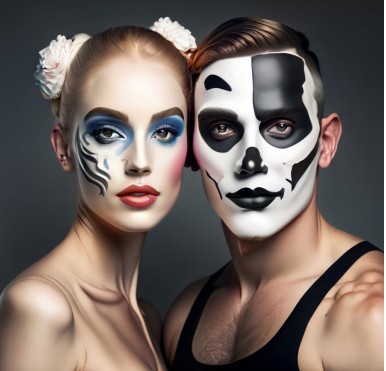

Two actors are posing for a picture with one wearing a black and white face paint.

Figure 40: High Quality Samples Produced by Meissonic.

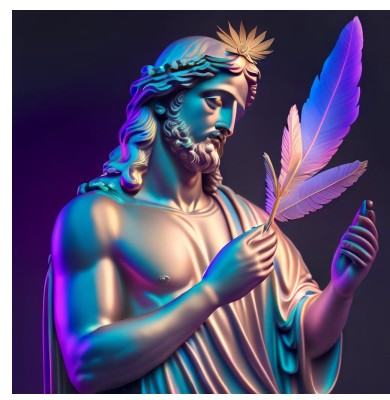

A statue of Jesus Christ is holding a feather in his hand in a purple style.

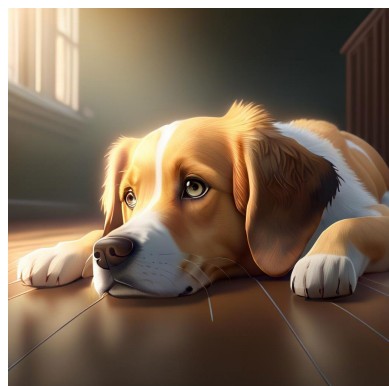

A dog is laying on the floor.

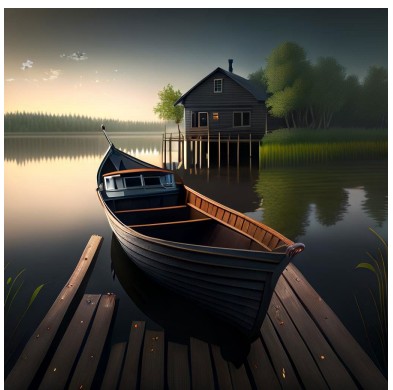

A black boat is tied to a dock on a calm lake.

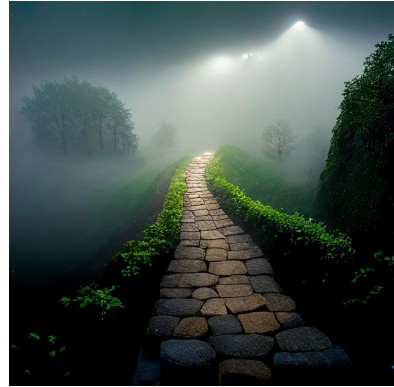

A narrow stone pathway is enveloped by lush greenery and a veil of mist.

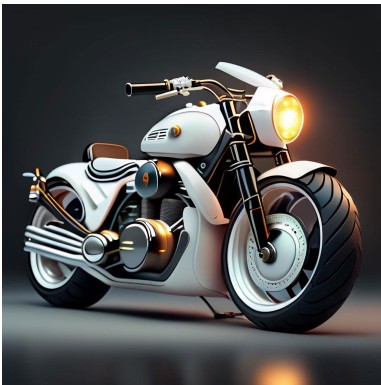

A white and black motorcycle with a headlight on it.

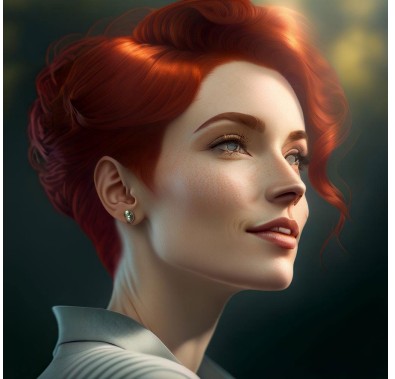

A woman with short red hair is looking off into the distance.

Figure 41: High Quality Samples Produced by Meissonic.

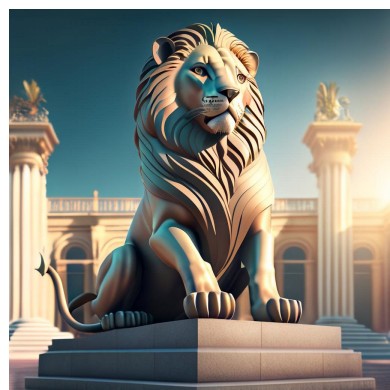

A statue of a lion stands in front of a building.

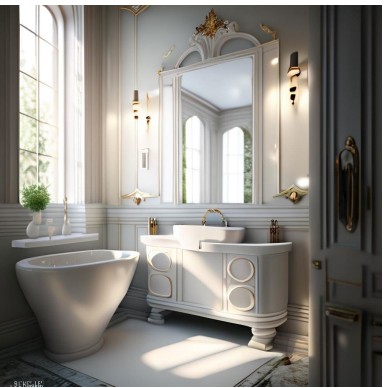

A bathroom with a modern design and a classic design.

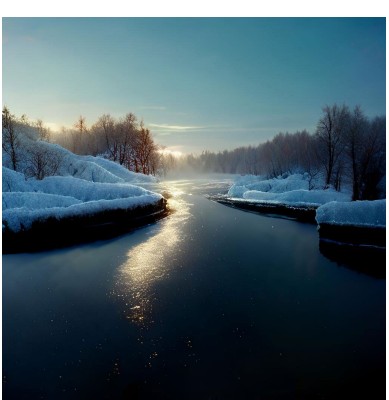

A frozen river with ice on the surface.

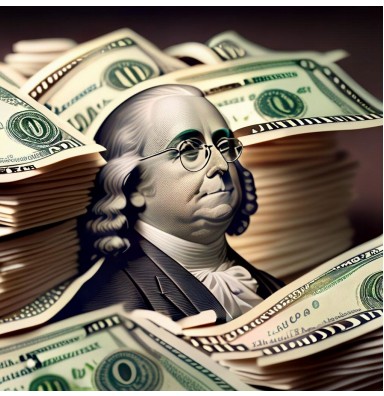

Benjamin Franklin appears among a pile of US dollars

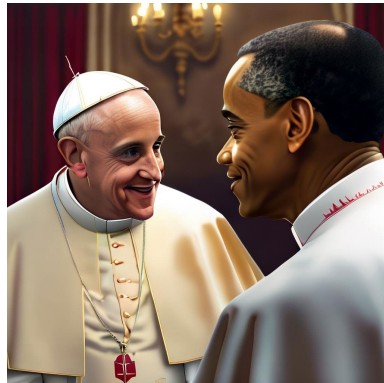

Pope Francis is talking to black priests.

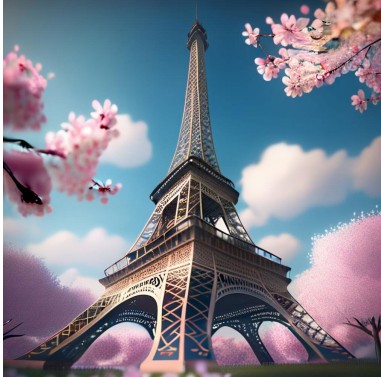

Cherry blossoms bloom under the Eiffel Tower.

Figure 42: High Quality Samples Produced by Meissonic.

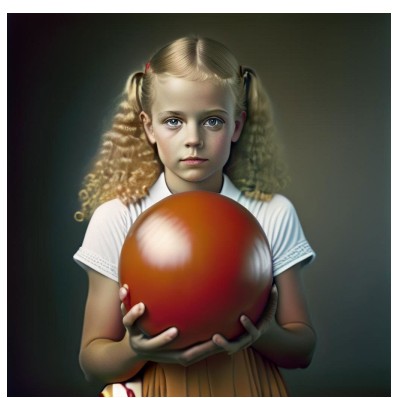

A young girl is holding a bowling ball.

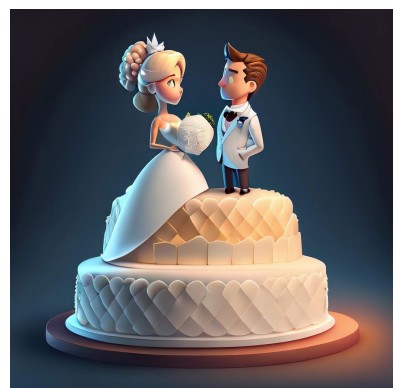

A pair of bride and groom figurines are positioned atop a white, two-tiered cake.

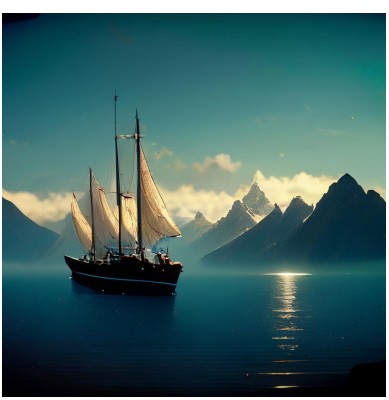

A ship is sailing in the ocean with mountains in the background.

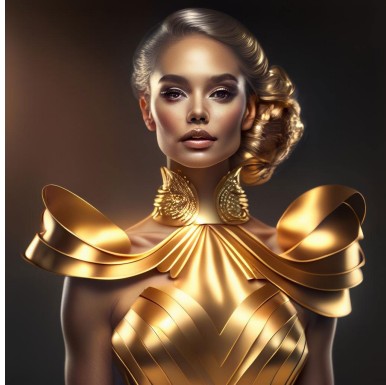

A woman in a gold dress poses for a photo.

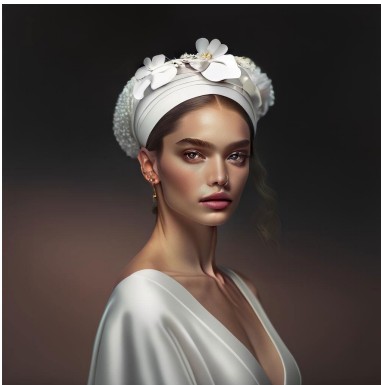

A woman wearing a headband hat and a white dress is walking down a runway.

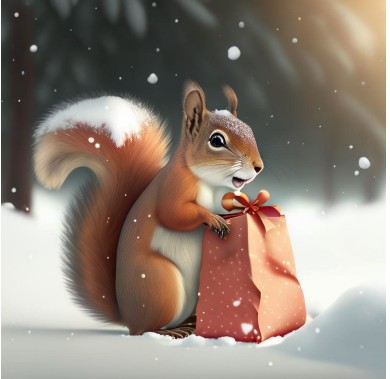

A squirrel is holding a gift bag with mouse open in the snow.

Figure 43: High Quality Samples Produced by Meissonic.

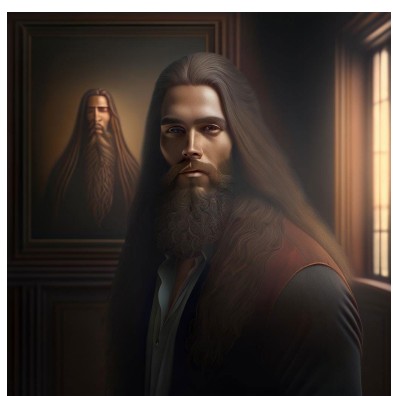

A man with long hair and a beard stands in a room, with a portrait of himself positioned behind him.

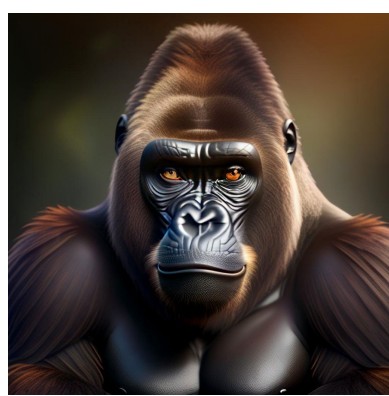

A gorilla is looking at the camera with a serious expression.

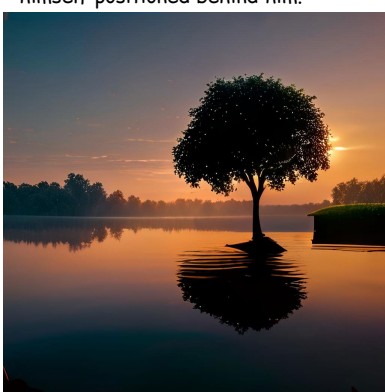

A sunset over a body of water with a tree in a small island.

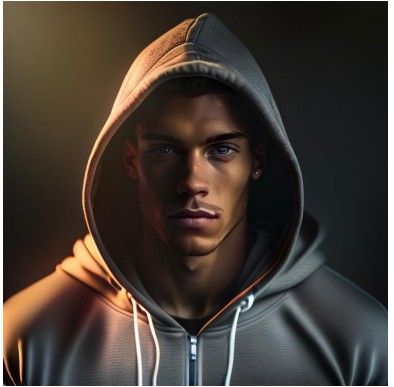

A man with a hoodie on is looking at the camera.

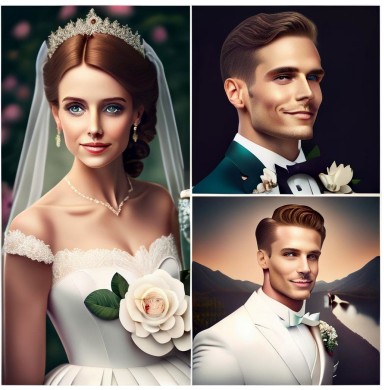

The collage consists of photos featuring the bride and groom. The bride occupies half of the collage. The groom appears in two photos, one in a white suit and the other in a black suit.

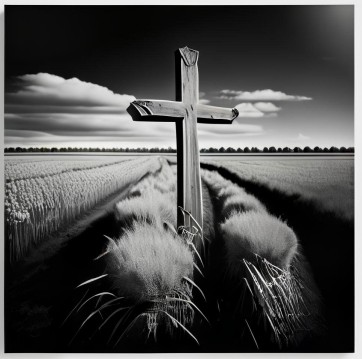

A black and white photo of a cross in a field.

Figure 44: High Quality Samples Produced by Meissonic.

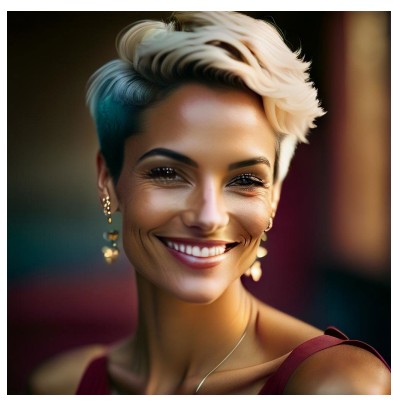

A woman with short hair and earrings is smiling.

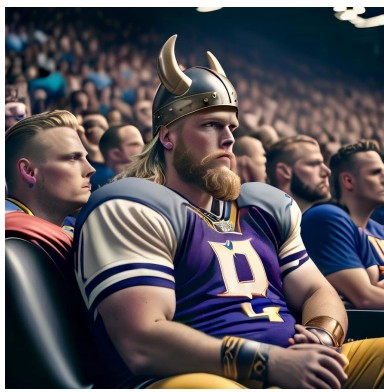

A man dressed in Viking attire is seated among the crowd.

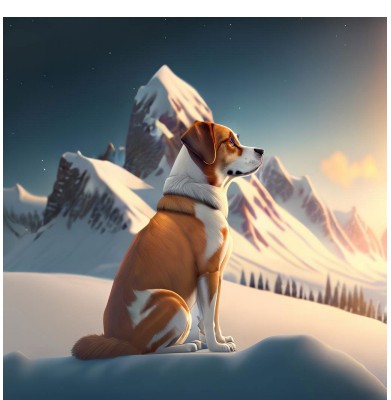

A dog is sitting in the snow in front of a mountain.

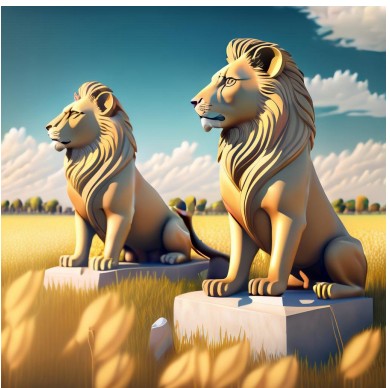

Two gloden statues of lions standing in a field.

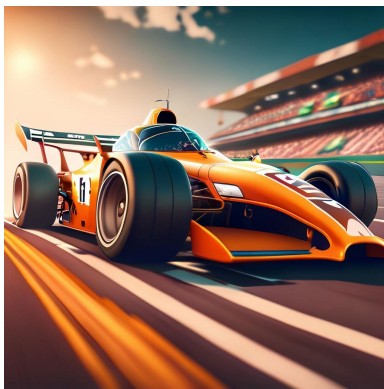

A race car is driving on a track.

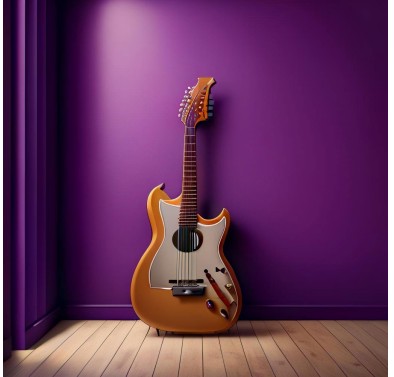

A guitar is sitting on a wooden floor in front of a purple wall.

Figure 45: High Quality Samples Produced by Meissonic.

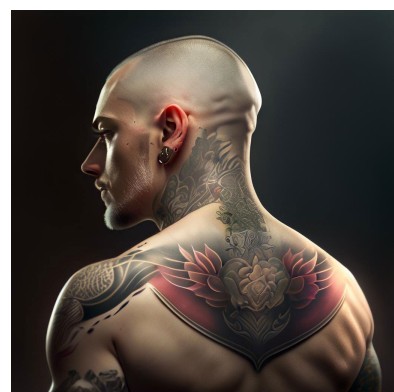

A man with a shaved head and a tattoo on his back.

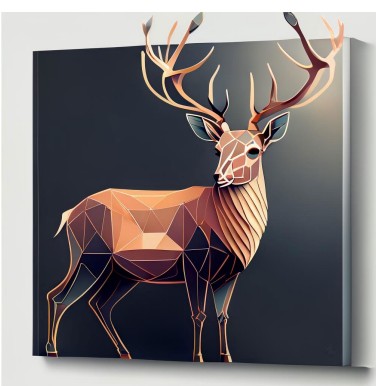

A deer is drawn in a geometric style.

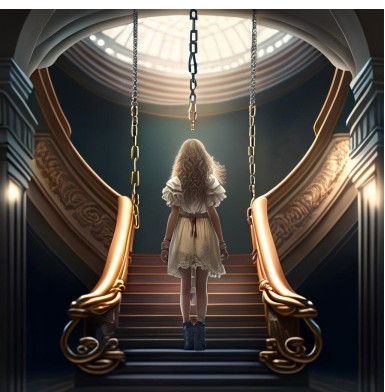

A woman is standing on a staircase, back to the camera with three chains hanging from the ceiling.

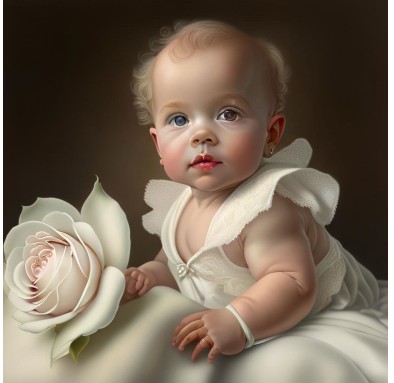

A baby is sitting on a white blanket holding a white rose.

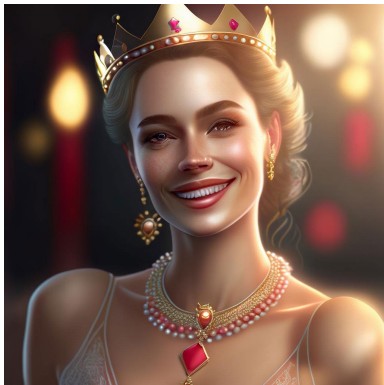

A woman wearing a crown and a necklace is smiling.

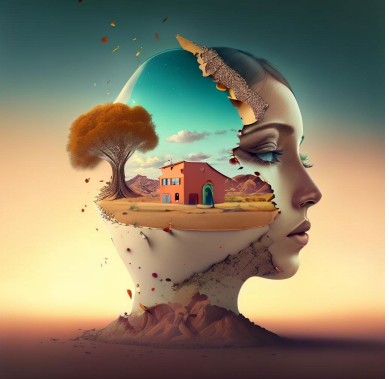

A surreal mental landscape, in which elements of nature and a house emerge from the back of a woman's head.

Figure 46: High Quality Samples Produced by Meissonic.

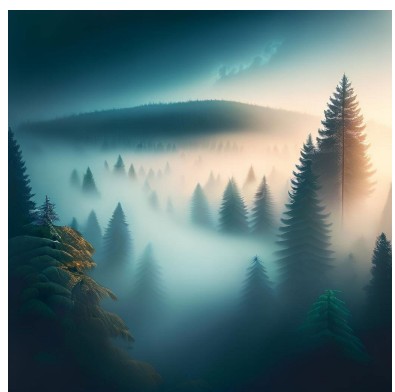

A forest with trees and fog.

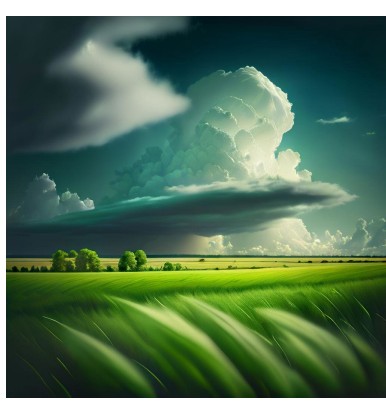

A cloudy sky over a green field.

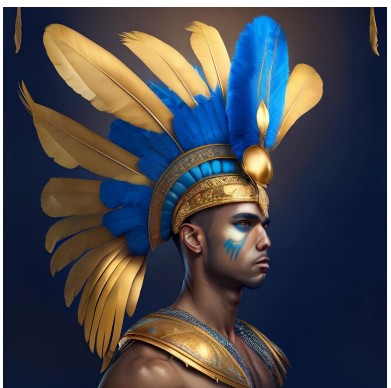

A man wearing a large blue and gold feathered headdress.

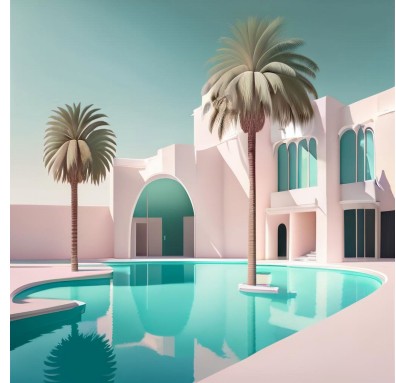

An image depicting a minimalist design featuring a pool situated in front of a white building with palm

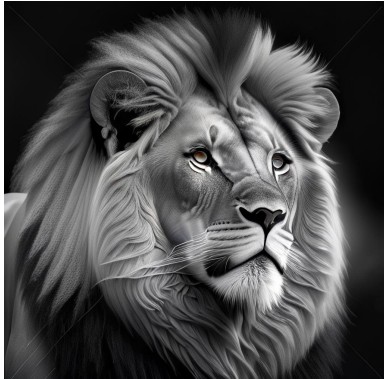

A lion's head is shown in a grayscale image.

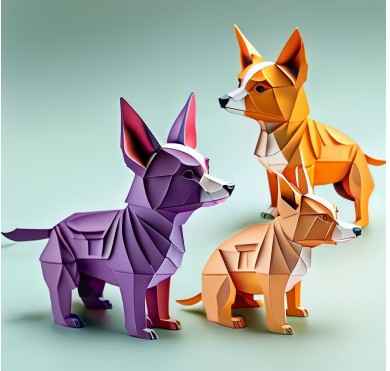

Three origami dogs, one of which is purple, while the others are yellow.

Figure 47: High Quality Samples Produced by Meissonic.

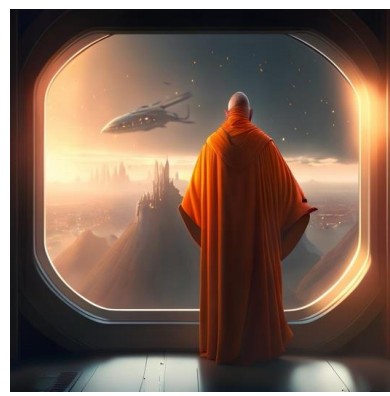

A blind monk wearing an orange robe stares out the window of a spaceship in a dramatic lighting as depicted in a matte painting.

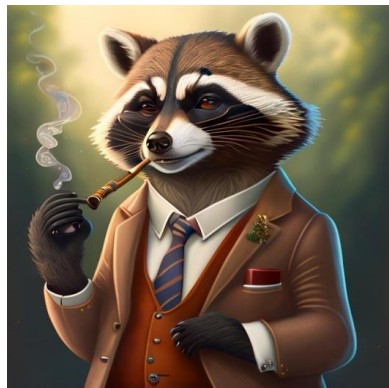

A racoon wearing a suit smoking a cigar in the style of James Gurney.

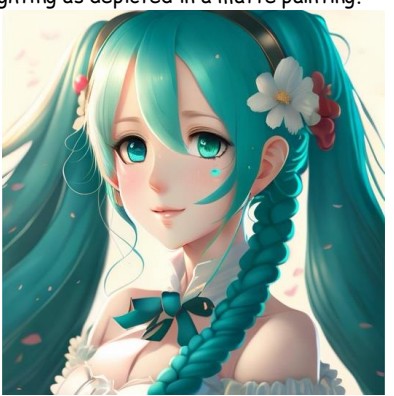

Classical romantic painting of Hatsune Miku with blue hair.

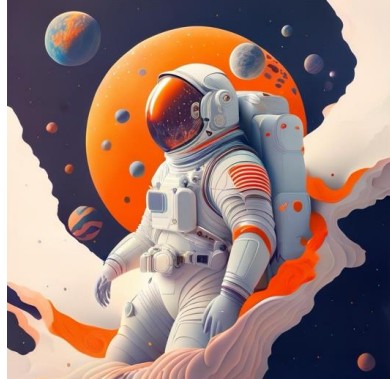

An astronaut floats amidst planets against a cosmic backdrop in a highly detailed, refreshing digital painting by James Jean.

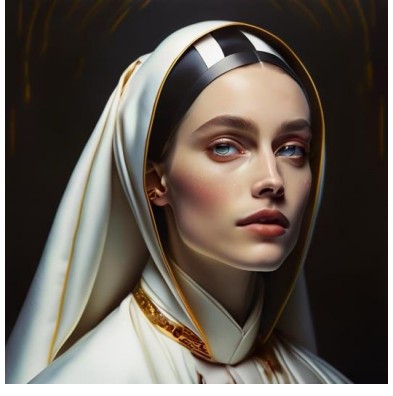

Close-up hyperrealistic oil painting portrait of a nun fashion model looking up against a black background, with classicism and 80s sci-fi Japanese book art influences.

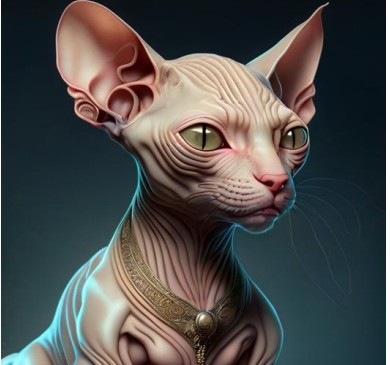

A digital painting of a hairless, inside-out cat with intricate details and a horror theme.

Figure 48: High Quality Samples Produced by Meissonic.

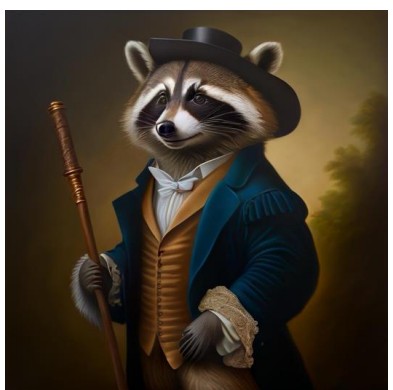

A raccoon in formal attire, carrying a bag and cane, depicted in a Rembrandt-style oil painting.

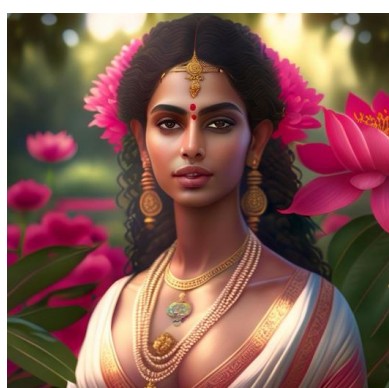

A cinematic fashion portrait of a Hindu goddess standing in a beautiful garden.

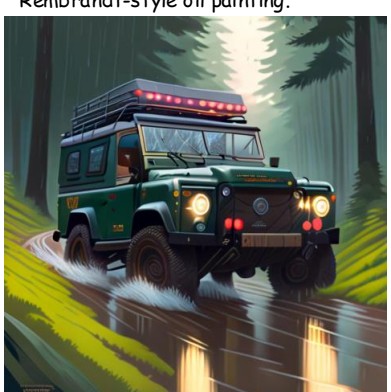

A Landrover crosses a forest path in the rain in a highly-detailed digital painting by artists Greg Rutkowski and Artgerm.

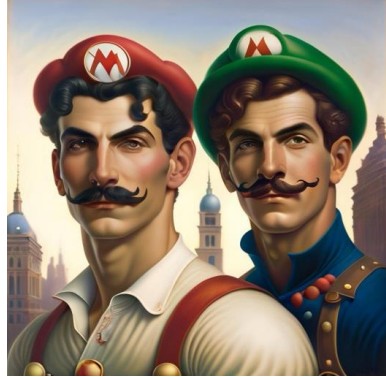

A portrait of Mario and Luigi from Mario Bros with a detailed face and a city background, painted by Bouguereau.

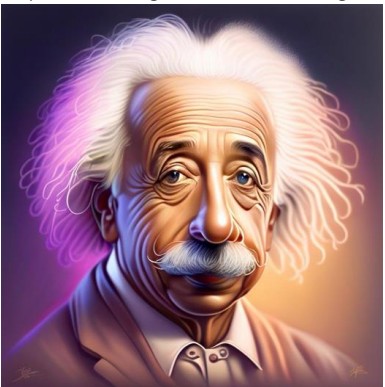

Image of Albert Einstein created by Park Jun Seong.

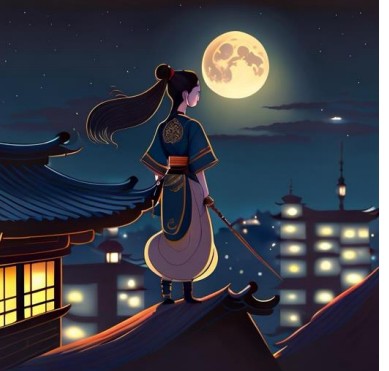

A painting depicting a wuxia character standing on a roof under a moonlit night.

Figure 49: High Quality Samples Produced by Meissonic.

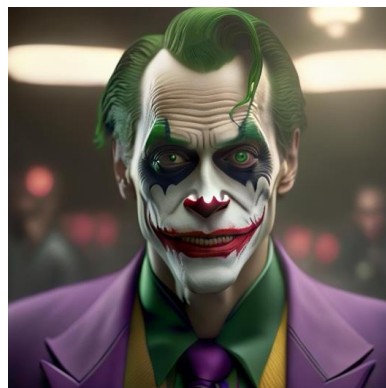

Steve Buscemi portrays the Joker.

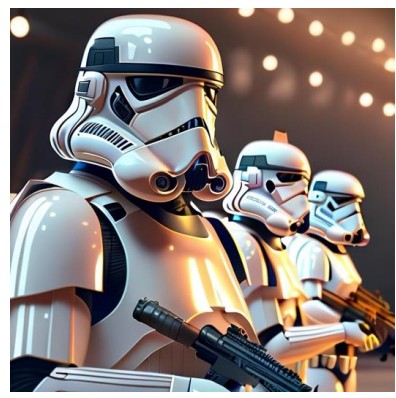

The image depicts stormtroopers in a hyper realistic style, with intricate and hyper detailed design, characterized by ambient and volumetric lighting, reminiscent of Star Wars concept art by George Lucas and Ralph McQuarrie, with a style similar to GTA V.

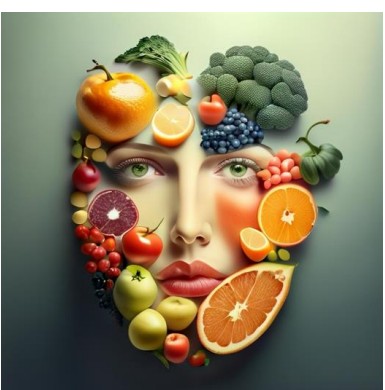

Image depicting a person's face composed entirely of fruits and vegetables.

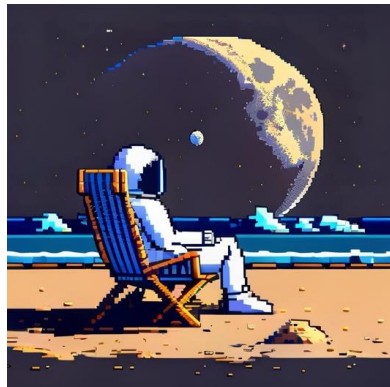

A space man sat on a beach chair on the moon, pixel art.

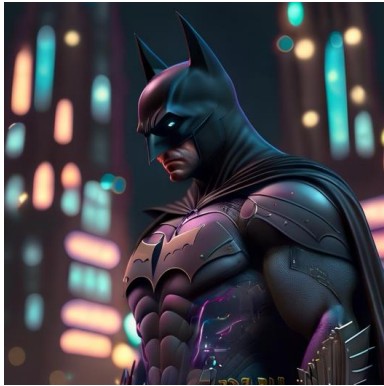

A cyberpunk-style Batman in a dark city, depicted in an extremely detailed piece of artwork by Chris Labrooy.

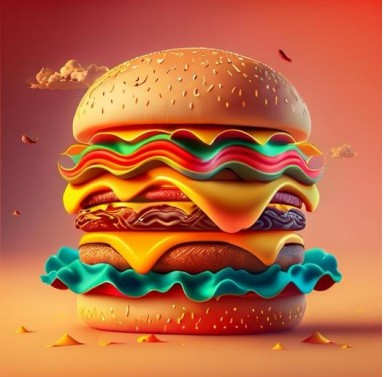

The image is a trippy cheeseburger with warm colors, depicted in highly detailed illustration and rendered in octane, created by the award winning studio 4.

Figure 50: High Quality Samples Produced by Meissonic.

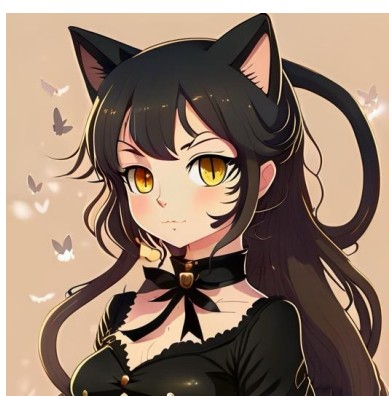

A Salem black cat girl in anime style with a simple background.

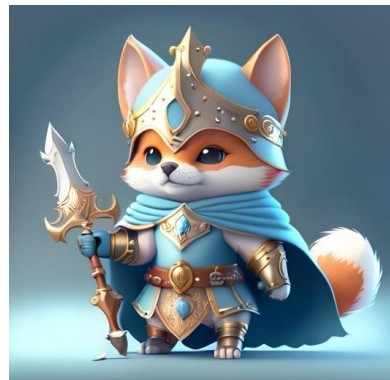

A cute anthropomorphic fox knight wearing a cape and crown in pale blue armor.

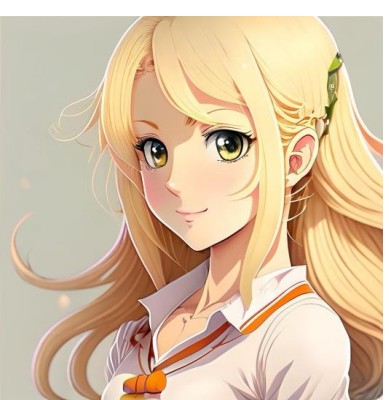

Blond-haired girl depicted in anime style.

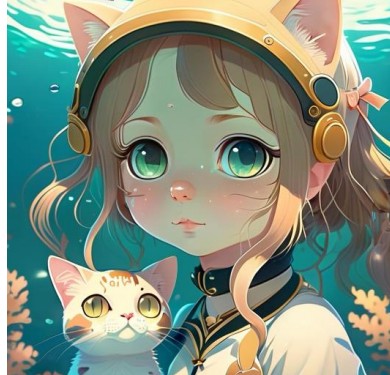

A cute anime-style female cat girl with large eyes is pictured underwater with a simple background.

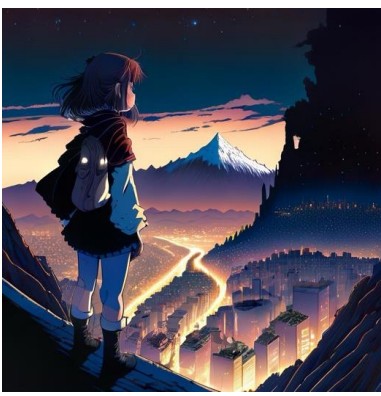

A girl peers over the edge of a mountain at a giant city in the dark of night, depicted in a manga illustration by Kentaro Miura and Hiromu Arakawa.

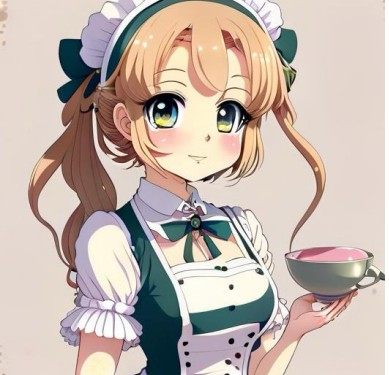

Illustration of an anime maid with a pretty face and eyes, shown in a full-body upper shot.

Figure 51: High Quality Samples Produced by Meissonic.

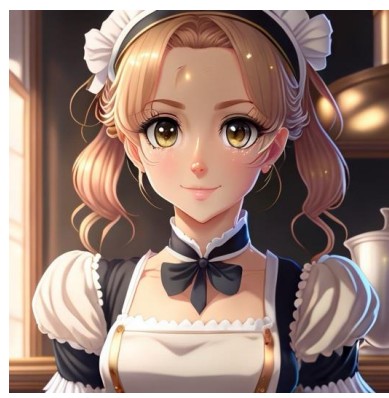

A full-body shot of an anime maid with rich detail, featuring a pretty face and eyes.

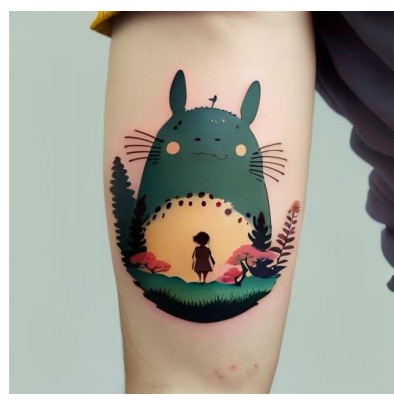

A minimalist tattoo inspired by the Studio Ghibli films

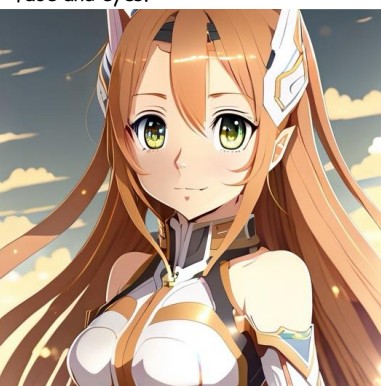

A puppy driving a car in a film still.

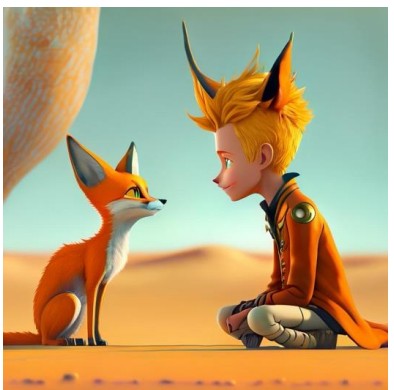

The Little Prince talking to the fox in an animation shot by Tim Burton's art.

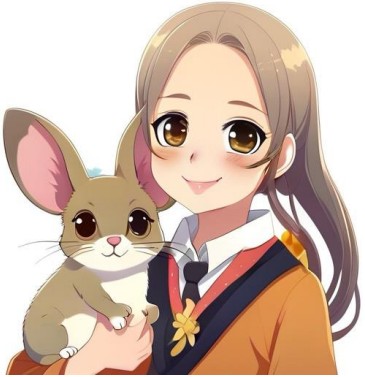

Anime portrait of an Asian schoolgirl with her pet sugar glider.

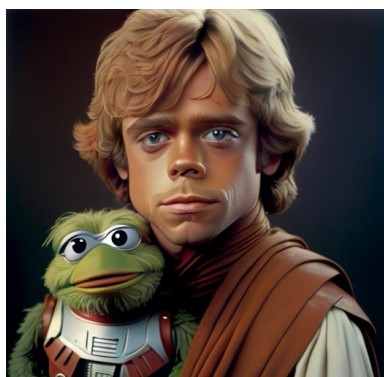

Luke Skywalker with Muppets.

Figure 52: High Quality Samples Produced by Meissonic.

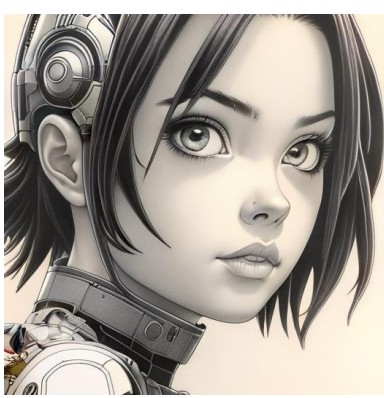

Medium shot black and white manga pencil drawing with a highly detailed face of Alita by Yukito Kishiro.

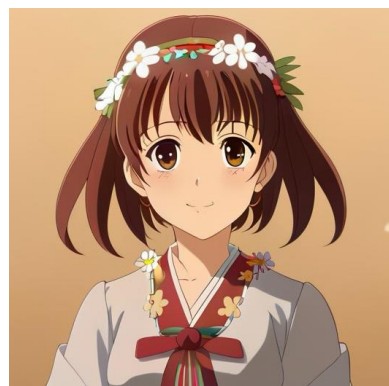

Studio photo portrait of Lain Iwakura from Serial Experiments Lain wearing floral garlands over her traditional dress.

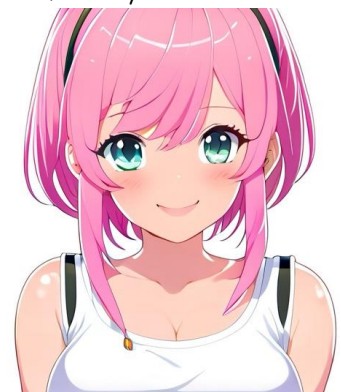

Frontal portrait of anime girl with pink hair wearing white t-shirt and smiling.

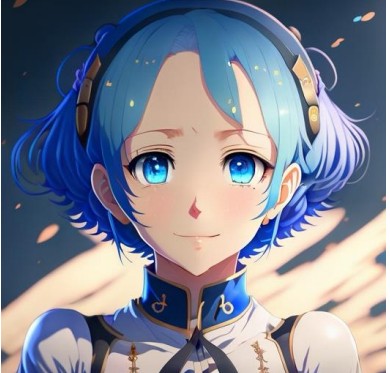

Anime oil painting of Rem from Re Zero.

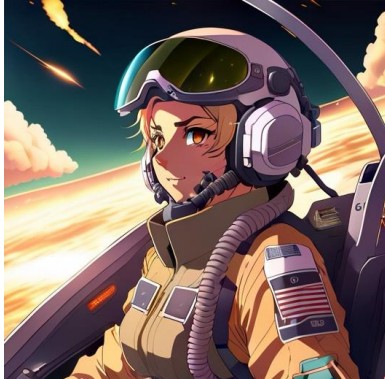

Anime-style fighter pilot in cockpit engaged in a night air battle with explosions.

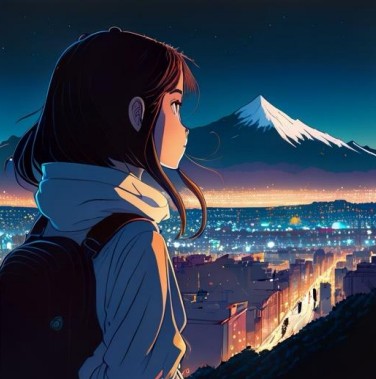

A girl gazes at a city from a mountain at night in a colored manga illustration by Diego Facio.

Figure 53: High Quality Samples Produced by Meissonic.

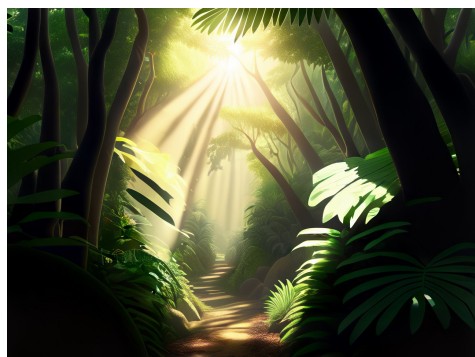

960 x 1280, A dense jungle with sunlight filtering through the canopy.

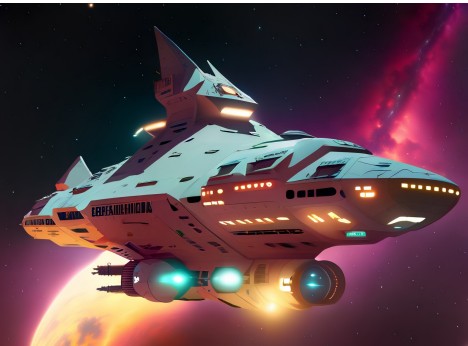

960 x 1280, A massive starship docked in a glowing nebula.

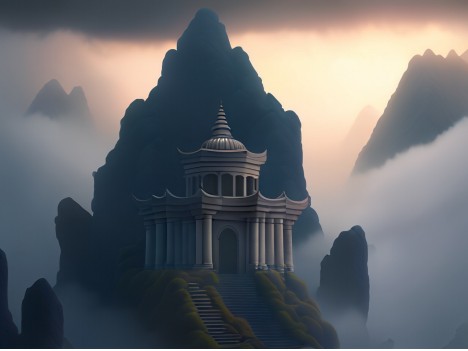

960 x 1280, A mystical temple hidden deep within a cloud-covered mountain.

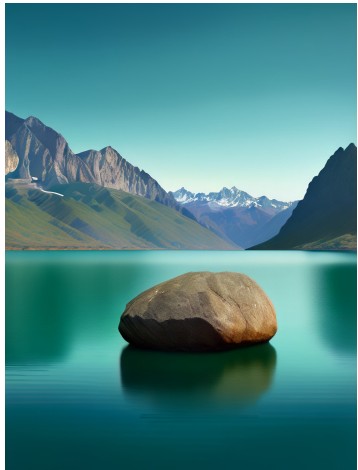

1280 x 960, A large body of water with a rock in the middle and mountains in the background.

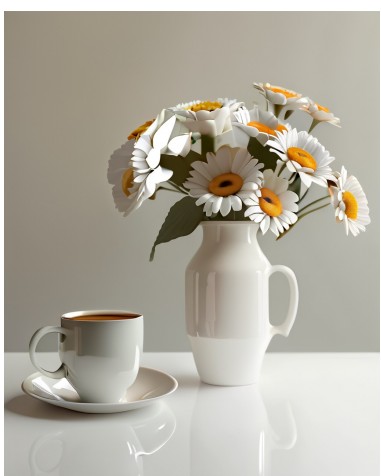

1280 x 1024, A white table with a vase of flowers and a cup of coffee on top of it.

Figure 54: More Images Produced by Meissonic at Diverse Resolutions.

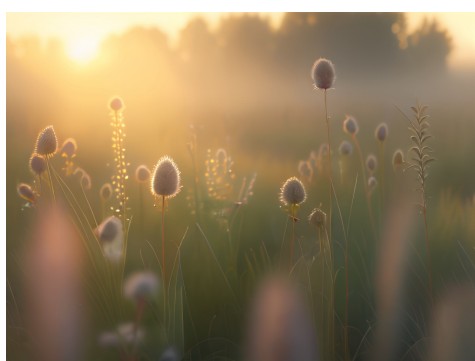

960 x 1280, A quiet meadow bathed in soft morning dew.

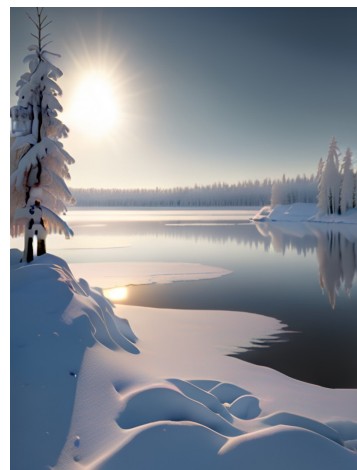

1280 x 960, A frozen lake surrounded by snow-covered trees under a pale winter sun.

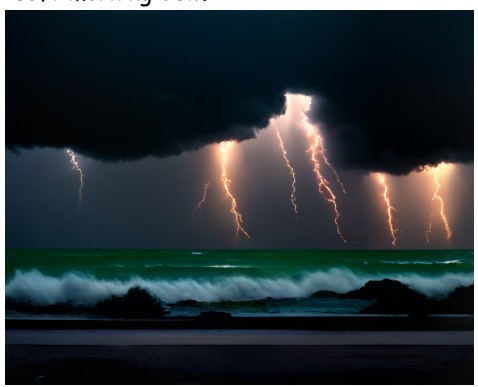

1024 x 1280, A stormy sea with crashing waves and lightning illuminating the clouds.

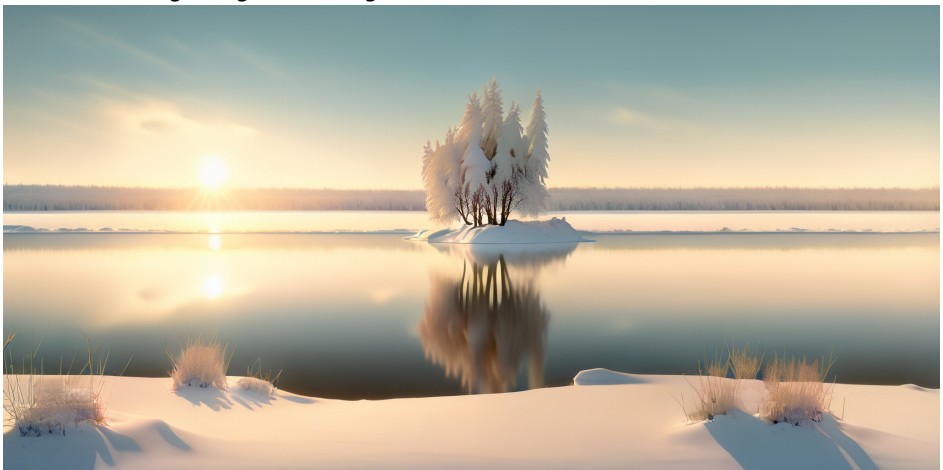

1024 x 2048, A frozen lake surrounded by snow-covered trees under a pale winter sun.

Figure 55: More Images Produced by Meissonic at Diverse Resolutions.

