# OpenReview forum: "Meissonic: Revitalizing Masked Generative Transformers for Efficient High-Resolution Text-to-Image Synthesis"
_ICLR.cc/2025/Conference — ICLR 2025 Poster_

### Official Review · Reviewer_qRU5 · 2024-10-27

**Soundness:** 3
**Presentation:** 4
**Contribution:** 2
**Rating:** 6
**Confidence:** 4

**Summary:**

The paper explores Masked Image Modeling (MIM), an non-autoregressive technique, and propose a new model named Messonic. Messonic poses a promising enhancement on MIM compared to its auto-regressive and diffusion-based counterparts. The keys to the improvement of Messonic are enhanced transformer architecture, advanced positional encoding and adding masking rate as a condition. The authors conduct extensive evaluations on Messonic to demonstrate its superiority.

**Strengths:**

### Motivation
- The paper shows a good motivation - MIM shows the potential capability to unify the language and vision modalities, and the paper attempts to bridge the gap between MIM and other popular image generation fashions.
- The paper shares several interesting observations on the current MIMs, including architecture design, positional encoding, and masking rate as a condition. It then starts from those insights and implement effective components to gain improvement.

### Method
- The proposed pipeline is straight-forward and seems effective.
- It is an interesting finding that embedding original image resolution, crop coordinates, etc would be helpful for the stability of the training, as mentioned in Line 229 - Line 232.
- The decomposition of training stages (Section 2.5) makes sense to me. It could work as a new procedure for other or future model training.

### Experiments/Results
- The authors conduct extensive experiments and evaluations to demonstrate the performance of Messonic.
- Messonic shows superiority on HPS benchmark, GenEval benchmark, and MPS scores. It shows competitive results with SDXL, while has a 2x smaller model size and transparent training costs (Table 1).
- Messonic also shows an economic expense of VRAM (i.e., 8GB). This demonstrates a promising value for customer level usage.
- Messonic shows a good performance in Zero-shot I2I task, as shown in Table 6 and Figure 5.

### Writing/Delivery
The paper is well-written. The technical parts are clear and easy to understand.

**Weaknesses:**

I do not see a critical weakness from this paper. Some minor points:
- Messonic seems to be designed to match the auto-regressive manner in language models. The authors indicates it aims to unify language-vision models (Line 014). However, I do not see a strong evidence to show the current Messonic achieves this goal in the paper.
- The proposed approach is more driven from an "add-on" manner, where most of the components in Messonic are "borrowed" from other works. For example, multi-modal transformer backbone from the ADD paper, RoPE from Roformer, tokenization from VQVAE. The core contribution of the paper, therefore, seems to be a practice of choosing these components and empirical trials.

**Questions:**

1. It is interesting that the diverse micro conditions (Line 229 - Line 232) are helpful for the training stability (Line 134). From a high level perspective, why would these conditions help with the stability?
2. It is interesting to see that you choose to use a "lightweight" text encoder, CLIP (Line 210), instead of using LLM encoders or a mixed way like in SD3. Is there a reason or a hypothesis behind this choice, not just a "trial-and-error" experience?

---

> ### Author Response · Authors · 2024-11-21
> **Official Response to Reviewer qRU5 (1/2)**
>
> Q1: How can Meissonic unify language-vision models?
>
> > Messonic seems to be designed to match the auto-regressive manner in language models. The authors indicates it aims to unify language-vision models (Line 014). However, I do not see a strong evidence to show the current Messonic achieves this goal in the paper.
>
> A1: We sincerely apologize if our initial phrasing led to any misunderstanding. To clarify, image generation models can generally be categorized into three types: diffusion models, autoregressive models, and non-autoregressive models (e.g., MIM). Meissonic belongs to the third category.
>
> It is worth noting that current unified language-vision models are usually built on autoregressive architectures. However, our focus in this work is on addressing the limitations of current non-autoregressive models, particularly their suboptimal T2I performance and resolution limitations. By leveraging the foundation of Masked Image Modeling (MIM), we propose Meissonic to achieve efficient text-to-image synthesis with SDXL-level performance.
>
>
>
> Q2: Clarify the contributions of the proposed approach.
>
> > The proposed approach is more driven from an "add-on" manner, where most of the components in Messonic are "borrowed" from other works. For example, multi-modal transformer backbone from the ADD paper, RoPE from Roformer, tokenization from VQVAE. The core contribution of the paper, therefore, seems to be a practice of choosing these components and empirical trials.
>
> A2: Meissonic is the first to elevate MIM to diffusion-level performance, verifying the feasibility and potential of MIM in high-quality image generation. This work sets the stage for future unified language-vision models and demonstrates a foundation model that can run on consumer GPUs with SDXL-level quality.
>
> We want to emphasize that our work provides a system-level contribution that integrates advancements in architecture, data utilization, and training strategies. By demonstrating that such a MIM foundation model can deliver SDXL-level results while remaining accessible for deployment on consumer GPUs.
>
> The broader impact of this research lies in inspiring the community to explore pathways toward MIM-based text-to-image foundation models. Meissonic's success underscores the potential of MIM as a cornerstone for building next-generation models that are both high-performing and efficient.

---

> > ### Comment · Reviewer_qRU5 · 2024-11-21
> >
> > RE: A1
> > Thanks for your clarification! Three categories help me understand the position of Meissonic better. Please consider revising the related part in the paper.
> >
> > RE: A2
> > Thanks for highlighting the contributions. It is notable that Meissonic works with an 8GB VRAM GPU with competitive performance.

---

> > > ### Comment · Reviewer_qRU5 · 2024-11-21
> > >
> > > RE: A3.
> > >
> > > Thanks for providing your intuitions behind micro-conditions. A follow-up question: what are the dimensions of these conditions?
> > >
> > > RE: A4
> > >
> > > Thank you for your analysis between different text encoders. It is interesting to see that CLIP performs in a more efficient way.

---

> > ### Comment · Reviewer_qRU5 · 2024-11-21
> >
> > After I read the revised paper and the responses from the authors, I think the main contribution of the paper is more about an empirical exploration on the performance and efficiency trade-off for MIM model. All the proposed designs are discussing how to achieve a balance between the performance and the efficiency.

---

> ### Author Response · Authors · 2024-11-21
> **Official Response to Reviewer qRU5 (2/2)**
>
> Q3: Explain how micro conditions affect training.
>
> > It is interesting that the diverse micro conditions (Line 229 - Line 232) are helpful for the training stability (Line 134). From a high level perspective, why would these conditions help with the stability?
>
> A3:
>
> Micro conditions significantly enhance the training process through three key mechanisms:
> 1. Image Size Conditioning: By conditioning on original image dimensions (height and width), the model learns size-related features, enabling better generalization across different image sizes without performance degradation.
> 2. Crop Parameter Conditioning: This helps the model understand cropping-related features, reducing cropping artifacts and enabling center-focused object generation during inference.
> 3. Human Performance Score Conditioning: By explicitly matching training data quality with image quality scores, the model can prioritize learning from high-quality samples while reducing the influence of lower-quality data.
>
> SD XL has demonstrated that image size and crop parameter conditioning are highly effective for training stability and model performance. We have also found that human performance score conditioning proves to be particularly valuable in our approach. **These micro conditions effectively serve as guiding labels during training, preventing unexpected performance degradation from low-quality samples and ensuring more controlled and targeted learning outcomes.**
>
>
>
> Q4: How do you balance the text encoder size with image generation involving visual texts?
>
> > It is interesting to see that you choose to use a "lightweight" text encoder, CLIP (Line 210), instead of using LLM encoders or a mixed way like in SD3. Is there a reason or a hypothesis behind this choice, not just a "trial-and-error" experience?
>
> A4: We appreciate the reviewer's question regarding the text encoder selection. Our choice of CLIP (ViT-H-14) over T5-XXL was driven by careful consideration of both training and inference efficiency while maintaining strong text understanding capabilities. As shown in our comparative analysis (see table below), CLIP offers significant advantages in both aspects:
> | Text Encoder | Sequence Length | Embedding Size | 1M Captions Inference Time (min:sec) | 1M Captions Embedding Storage (GB) | VRAM Usage (MB) | Parameter Count (M) |
> |----------------------|------------------|----------------|---------------------------|--------------|-----------------|---------------------|
> | CLIP (ViT-H-14) | 77 Tokens | 1024 | 3:20 | 157 | 1,370 | 354 |
> | T5 (T5-xxl) | 120 Tokens | 4096 | 33:04 | 983 | 18,194 | 4,762 |
>
> The training efficiency heavily favors CLIP, as T5-XXL would require massive storage resources for caption embeddings (approximately 983GB per million captions). For large-scale training with 200M image-text pairs, this would demand nearly 200TB of storage, creating substantial I/O bottlenecks and significantly reducing training throughput. In contrast, CLIP's more compact embeddings (157GB per million captions) enable much more efficient data handling while maintaining robust text understanding.
> Regarding inference efficiency, CLIP demonstrates clear advantages with only 1.37GB VRAM usage compared to T5's 18GB requirement, while delivering 10x faster inference speed (3:20 vs 33:04). This efficiency doesn't come at the cost of performance - CLIP's proven capability in understanding visual-semantic relationships makes it particularly well-suited for image generation tasks. The CLIP text encoder's 354M parameters (versus T5-xxl's 4.7B) provide an excellent balance between computational efficiency and text interpretation capabilities, making it the optimal choice for our system's requirements.
>
> Additionally, we found that the capability for word rendering primarily stems from larger models like T5 XXL, so **we present Meissonic’s ability to synthesize letters in Figure 10** (as described in line 215 of the original manuscript). Our first goal is to develop an efficient text-to-image foundation model, which guided our choice of text encoder.
>
>
> Furthermore, **we provide performance comparisons for complex and simple prompts in Appendix D in the updated manuscript**. Our findings indicate that with 1–3 word prompts, the model tends to generate simpler images. With 7–10 word prompts, it produces images of sufficient quality, and for prompts up to 70 words, it incorporates more detailed descriptions, although image quality does not significantly improve. As a result, our current model effectively handles prompts of fewer than 77 tokens, generating visually appealing images with satisfactory quality.
>
>
>
> We hope this clarifies your concerns. We are committed to thoroughly incorporating your suggestions in the updated manuscript. Thank you once again for your excellent feedback.

---

> ### Author Response · Authors · 2024-11-21
> **Official Response to Reviewer qRU5**
>
> Thanks for your valuable feedback. We have revised the abstract, introduction and method sections, with changes highlighted in blue (line 12 - 14, line 26 - 38, line 145 -147). We hope these updates improve the clarity and help readers better understand our work.
>
>
> Regarding micro-conditions, our approach begins with a five-dimensional tensor ([batch_size, 5]) that includes `original_height, original_width, crop_top, crop_left, human_preference_score`. Specifically, crop_top and crop_left are obtained using `transforms.RandomCrop.get_params`, while the `human_preference_score` is annoated using the HPS v2.1 evaluation model [1], given an image and its caption. These raw values are then embedded ([batch_size, 5, 256]), reshaped ([batch_size, 1, 1280]) and concatenated with the pooled text embedding ([batch_size, 1, 1024]) to serve as conditions, as illustrated in Figure 2. We will release all the codes to enhance clarity and reproducibility.
>
> Moreover, we want to emphasize that previous MIM models have shown quite limited performance, as discussed in section 2.1. With five aspects of architectural innovations, Meissonic is the first to elevate MIM to SDXL-level performance, verifying the feasibility and potential of MIM in high-quality image generation. This work **sets the base model for future masked image modeling research** and **presents a foundation model that can run on consumer GPUs with SDXL-level quality**.
>
> Thanks again for your prompt reply. If you have further questions or suggestions, please let us know. We are eager to address any concerns and continue improving the manuscript.
>
>
> [1] Human Preference Score v2: A Solid Benchmark for Evaluating Human Preferences of Text-to-Image Synthesis, arXiv:2306.09341

---

> > ### Comment · Reviewer_qRU5 · 2024-11-23
> >
> > Thanks for your responses. My concerns are resolved.

---

> > > ### Author Response · Authors · 2024-11-23
> > > **Official Response to Reviewer qRU5**
> > >
> > > Thank you once again for your prompt reply and the time you dedicated to reviewing this work.

---

### Official Review · Reviewer_BVBS · 2024-11-04

**Soundness:** 3
**Presentation:** 3
**Contribution:** 2
**Rating:** 6
**Confidence:** 5

**Summary:**

The paper scales up a masked image modeling (MIM) framework for professional-quality image generative model comparable to the diffusion-based SDXL. The key problems the authors have find in MIM framework is its low resolution constraint and persisting performance gap. The newly proposed model, Meissonic model incorporates efficient text encoder, mixed single-modal and multi-modal transformer backbone using rotary positional encoding, and curated training datasets. The final model performs comparable to SDXL in quality and exceeds efficiency in inference time and memory requirement.

**Strengths:**

- The presented method is lightweight, in terms of the number of parameters and the training time, lowering production cost of large scale image generative models.
- The model also have reduced inference time with lower GPU memory cost compared to SDXL.
- The training procedure is opened with transparency as shown in Figure 15.
- The model supports masked image editing and inpainting.

Overall, the proposed method in this manuscript can be regarded as a good industrial image generative model with production-ready quality.

**Weaknesses:**

The main concerns are in the presentation of the material. Although the model at this reported performance can contribute to the generative AI community, the current form of manuscript seems more like an industrial technical report than an academic article.

- Since the authors have claimed for supporting inpainting, outpainting, mask-free editing, and masked editing, there should be a comparison in both quantitative and qualitative way for each of the task in the manuscript. For now, the article only contains EMU-Edit scores with reduced number of columns--the original EMU-Edit contains five algorithmic ratings (CLIP_dir, CLIP_im, CLIP_out, L1, DINO) and two human ratings (Text Alignment, Image Faithfulness), measured in two different test sets (Emu Edit / MagicBrush). Also, the qualitative comparison is not shown in the paper, showing only the proposed method in Figure 6 and 7. Overall, this writing style makes the article more of an well-written industrial technical report than an academic paper. As a recommendation, there are many well-established detailed quantitative metrics such as masked image preservation and text alignment (e.g., PSNR/LPIPS/MSE as in BrushNet, Table 2), and overall generation quality (e.g., FID as in Table 2 of PQDiff), which can be applied to compare with, for example, SDXL.
- Maybe it is due to the intrinsic characteristics of MIM framework, but the inpainting in Figure 7 seems to change the non-masked regions of the image. This deformation should be measured in quantitative metrics such as in BrushNet.
- It would be better to have plots depicting how the generated quality changes by changing the number of iterations in the inference, or at least a recommendation of the number of inference steps of using the pre-trained model.
- I believe image demonstration from page 20 to the end is unnecessary and carry little information about the model. It is not bad to have a lot of demonstration images in a technical report, but there should be at least qualitative comparison between other models in order to have persuasive narrative. For example, generating random 5-16 images from the same text prompt without curation and show the diversity and the point of breaking (e.g., broken hands or limbs) can be visually compared with other large scale generative models such as SDXL-base, which is quoted multiple time throughout the manuscript.
- I would like to recommend adding details about training and inference algorithms. For example, the explanation of “micro controls” and architectural detail of “feature compression layers” is not described in appropriate level of details (what is chosen, how's the size of this controls, and how each component benefits the training, etc.). Moreover, even if the original idea is borrowed from previous papers, such information should be contained in the paper for completeness, e.g., how is 1D RoPE encoding used in this specific 2D MIM Transformer block?

Regarding the mentioned issues, I believe that the current version of the manuscript is not ready for publication.

**Questions:**

These are questions that are minor enough that do not count in the final scores.

- Does this model architecture supports DreamBooth-type fine-tuning for practical application?
- The success of U-Net-based architecture (e.g., SD, SDXL) partially relies on the strong controllabiliy gained from ControlNet. However, such I do not want to require the authors to provide the experimental results or detailed idea, but I wonder if they are planning to develop in this direction.
- I believe it would be great to have Figure 11 in the main manuscript, since this shows justification of changes made during the development of the proposed model.

---

> ### Author Response · Authors · 2024-11-21
> **Official Response to Reviewer BVBS (1/3)**
>
> We appreciate your positive remarks on the “production-ready quality” of our work. We also value your suggestion to include Figure 15 (Figure 27 in the updated manuscript) in the main manuscript. Originally, due to space constraints, we placed it in the appendix.
>
> Q1:  Provide more comparisons to demonstrate the zero-shot image editing ability of Meissonic.
>
> > Since the authors have claimed for supporting inpainting, outpainting, mask-free editing, and masked editing, there should be a comparison in both quantitative and qualitative way for each of the task in the manuscript. For now, the article only contains EMU-Edit scores with reduced number of columns--the original EMU-Edit contains five algorithmic ratings (CLIP_dir, CLIP_im, CLIP_out, L1, DINO) and two human ratings (Text Alignment, Image Faithfulness), measured in two different test sets (Emu Edit / MagicBrush). Also, the qualitative comparison is not shown in the paper, showing only the proposed method in Figure 6 and 7. Overall, this writing style makes the article more of an well-written industrial technical report than an academic paper. As a recommendation, there are many well-established detailed quantitative metrics such as masked image preservation and text alignment (e.g., PSNR/LPIPS/MSE as in BrushNet, Table 2), and overall generation quality (e.g., FID as in Table 2 of PQDiff), which can be applied to compare with, for example, SDXL.
>
> A: **The primary focus of our paper is to present an innovative MIM-based text-to-image (T2I) foundation model**, as emphasized in the title. Our goal is to birdge the gaps between MIM-based T2I and diffusion-based model. Thus, showcasing zero-shot editing is an auxiliary highlight, rather than our main focus.
>
> **Experimental results and details are provided in Appendix F of the updated manuscript.** We include both quantative comparisons (Table 7, 8, 9) and qualitative comparisons (Figure 20) with SD1.5 and SDXL.
>
> To ensure fair evaluations of zero-shot capabilities with SD1.5 and SDXL, we utilize Null-Text Inversion for zero-shot editing with our method, taking into account that other methods have been extensively trained on editing datasets.
>
> For consistency, we use the recommended 512 resolution for editing and run the tests using torch.float32, which is the official setting for Null-Text Inversion. On A6000 GPUs (48 GB), the execution of MagicBrush takes approximately 36 hours for SD1.5 and 60 hours for SDXL. The runtime for Emu-Edit is significantly longer. Given the extensive computation, we randomly sample 500 examples per benchmark for testing.
>
> **EMU-Edit Results:**
>
> |                         | CLIP-I↑ | CLIP-T↑ | DINO↑ | L1↓   | CLIPdir↑ |
> | ----------------------- | ------- | ------- | ----- | ----- | ------- |
> | SD 1.5 + Null-Text Inv. | 0.780   | 0.240   | 0.637 | 0.159 | 0.096   |
> | SDXL + Null-Text Inv.   | 0.787   | 0.238   | 0.653 | 0.146 | 0.085   |
> | Meissonic (Ours)        | 0.791   | 0.244   | 0.689 | 0.128 | 0.102   |
>
> **MagicBrush Results:**
>
> |                         | CLIP-I↑ | CLIP-T↑ | DINO↑ | L1↓   | CLIPdir↑ |
> | ----------------------- | ------- | ------- | ----- | ----- | ------- |
> | SD 1.5 + Null-Text Inv. | 0.824   | 0.228   | 0.647 | 0.121 | 0.106   |
> | SDXL + Null-Text Inv.   | 0.840   | 0.241   | 0.665 | 0.122 | 0.111   |
> | Meissonic (Ours)        | 0.835   | 0.248   | 0.689 | 0.115 | 0.120   |
>
> Our findings indicate that due to the inherent characteristics of MIM, Meissonic exhibits faster zero-shot editing capabilities. Performances are evaluated with `batch size = 1` and `inference step = 50` (compared to Null-Text Inv., which requires 500 backpropagation steps). Tests are conducted on an A6000 GPU with 48 GB VRAM.
>
> |                   | SD 1.5 + Null-Text Inv. | SDXL + Null-Text Inv. | Meissonic (Ours) |
> | ----------------- | ----------------------- | --------------------- | ----------------- |
> | Time (s/10 pairs) | 1040 + 100              | 1850 + 120            | 108               |
> | GPU (GB)          | 13.4                    | 26.8                  | 5.9               |
>
> These results demonstrate the substantial potential for reduced processing time with Meissonic.
>
> Q2: Explain why Figure 7 seems to change the non-masked regions of the image.
> > Maybe it is due to the intrinsic characteristics of MIM framework, but the inpainting in Figure 7 seems to change the non-masked regions of the image. This deformation should be measured in quantitative metrics such as in BrushNet.
>
> A2: We apologize for any confusion. The original image input was not square, resulting in cropping and flipping during processing. However, when visualizing, only simple center-cropping was applied. We have fixed this issue in the updated manuscript in Figure 8 (line 503 - 507).

---

> ### Author Response · Authors · 2024-11-21
> **Official Response to Reviewer BVBS (2/3)**
>
> Q3: Present how the generated quality changes with varying inference steps.
> > It would be better to have plots depicting how the generated quality changes by changing the number of iterations in the inference, or at least a recommendation of the number of inference steps of using the pre-trained model.
>
> A3: We have provided recommended inference settings in our original manuscript, with `cfg = 9` and `steps = 48`. Additionally, **we include performance comparisons with different numbers of inference steps and Classifier Free Guidance (CFG) in Appendix E (Figure 14, 15, 16, 17, 18, 19) in our updated manuscript**. Please note that different prompts may require varying optimal inference steps. We observe that 30 steps are sufficient for synthesizing images in some cases, and 48 steps produce images that are more than satisfactory in most cases. Beyond 48 steps, further increasing the number of inference steps yields minimal improvement in image quality in most cases.
>
>
> Q4:  Justify the inclusion of extensive examples from page 20 onward.
> > I believe image demonstration from page 20 to the end is unnecessary and carry little information about the model. It is not bad to have a lot of demonstration images in a technical report, but there should be at least qualitative comparison between other models in order to have persuasive narrative. For example, generating random 5-16 images from the same text prompt without curation and show the diversity and the point of breaking (e.g., broken hands or limbs) can be visually compared with other large scale generative models such as SDXL-base, which is quoted multiple time throughout the manuscript.
>
> A4: We cannot supply the model checkpoints or an inference API for external verification. Thus, to strengthen our findings, we include a substantial number of inference samples to demonstrate Meissonic's capabilities. This section aims to highlight the model’s unique strengths rather than a direct comparison with SDXL.
>
> We follow your suggestions and compare Meissonic with SDXL by generating random 9 images from the same text prompt, **more details are provided in Appendix G (Figure 21, 22, 23)**. Our findings indicate that Meissonic produces more colorful and visually appealing images while adhering more closely to the prompts (e.g., "in a glass vase," "deep indigo sky").

---

> ### Author Response · Authors · 2024-11-21
> **Official Response to Reviewer BVBS (3/3)**
>
> Q5: Could you provide more details on the training and inference algorithms, particularly regarding the “micro controls” and the architectural specifics of the “feature compression layers”?
>
> > I would like to recommend adding details about training and inference algorithms. For example, the explanation of “micro controls” and architectural detail of “feature compression layers” is not described in appropriate level of details (what is chosen, how's the size of this controls, and how each component benefits the training, etc.). Moreover, even if the original idea is borrowed from previous papers, such information should be contained in the paper for completeness, e.g., how is 1D RoPE encoding used in this specific 2D MIM Transformer block?
>
> A5: The training pipeline integrates micro conditions such as the original image resolution, cropped image dimensions, and human performance scores. While previous works have utilized various micro conditions, none have incorporated human performance scores as part of the model's guidance. In our approach, these micro control tokens are embedded and concatenated with the pooled text embeddings to guide the generation process effectively.
>
> During training, the backbone learns to recognize the resolution, crop information and human performance scores of the images, understanding both the size and quality of its training data. This enables the model to adjust its learning preference, focusing more on high-quality data while placing less emphasis on lower-quality data. This strategy significantly enhances training efficiency based on data quality. **We have included detailed explaination in Section 2.2 Diverse Micro Conditions (line 212 - 217 in the updated manuscript) in our original manuscript.**
>
> The feature compression layers are designed with a series of convolutional and normalization layers. They reduce the input feature dimension from 4096 to an output dimension of 1024. This module helps maintain computational efficiency while preserving essential feature information. **We have included detailed explaination in Section 2.3 Feature Compression Layers in our original manuscript, and more explaination in line 284 - 287.**
>
> Similar to [1][2][3], 1D RoPE encoding is applied within this specific 2D MIM Transformer block. The 2D MIM tokens are categorized into two types: image tokens and text tokens. Although image tokens are inherently 2D, they can be reshaped into a 1D sequence, allowing for the concatenation of 1D image tokens and text tokens using 1D RoPE encoding. **We have included detailed explaination in line 242 - 245 in the updated manuscript.**
>
> We consider these to be common knowledge within the field, just as we do not introduce what diffusion models or MIM models are. And we plan to release the source code, which we believe will offer a clearer, more intuitive understanding of the micro controls and feature compression layers. So our initial manuscript does not provide sufficient detail in these areas.
>
> [1] Fit: Flexible vision transformer for diffusion model, arXiv:2402.12376
>
> [2] VILA: On Pre-training for Visual Language Models, arXiv:2312.07533
>
> [3] Lumina-Next: Making Lumina-T2X Stronger and Faster with Next-DiT, arXiv:2406.18583
>
>
> Q6: Include information on related applications like DreamBooth and ControlNet.
> > Does this model architecture supports DreamBooth-type fine-tuning for practical  application?
> > The success of U-Net-based architecture (e.g., SD, SDXL) partially relies on the strong controllability gained from ControlNet. However, such I do not want to require the authors to provide the experimental results or  detailed idea, but I wonder if they are planning to develop in this direction.
>
> A6: The Meissonic architecture supports DreamBooth and LoRA fine-tuning, and we will release related training codes.
>
> For spatial control, existing methods either append semantic tokens (e.g., GLIGEN [1]) or use a residual backbone copy. Given Meissonic’s MIM-based encoding of images into discrete tokens, our forthcoming spatial control pipeline, inspired by GLIGEN but uniquely implemented, has shown promising results, and development is ongoing.
>
> [1] GLIGEN: https://gligen.github.io/
>
>
>
> We trust this clarifies and addresses your concerns. We are dedicated to fully incorporating your feedback into the updated manuscript. If you have any further questions, please let us know.

---

> ### Author Response · Authors · 2024-11-23
> **Look forward to your post-rebuttal feedback!**
>
> Dear Reviewer BVBS,
>
> Thanks again for your positive remarks on the “production-ready quality” of our work. Since the deadline of discussion is approaching, we are happy to provide any additional clarification that you may need.
>
> In our previous response, we have carefully studied your comments and made detailed responses summarized below:
>
> 1. Provided further comparisons to demonstrate the zero-shot image editing ability of Meissonic in Appendix F.
>
> 2. Presented how the generated quality changes with varying inference steps in Appendix E.
>
> 3. Provided more comparisons with SDXL to demonstrate the image generation ability of Meissonic in Appendix G.
>
> 4. Refined more details on the training and inference algorithms and highlighted in blue.
>
> 5. Offered additional discussion on related applications like DreamBooth and ControlNet.
>
> We hope that the provided new experiments and the additional discussion have convinced you of the contributions of our submission.
>
> Please do not hesitate to contact us if there's additional clarification we can offer. Thanks!
>
> Thank you for your time!
>
> Best, Authors

---

> > ### Comment · Reviewer_BVBS · 2024-11-24
> > **I will raise my score as my concerns are resolved**
> >
> > Dear Authors,
> >
> > Thank you for the comprehensive and diligent responses to my review. I have carefully read the new manuscript and your responses. I am very much impressed on the amount of hard works behind this update you have made in this short timeframe. Well done, Authors! I will happily raise my scores since all of my concerns are resolved. I think this has become more an academic paper than before.
> >
> > A few final thoughts (not affecting scores):
> > - I know this is somewhat superfluous for a conference paper, but have you tried scaling up the sample resolution? SD1.5/SD2.1/SDXL are based on U-Net, so they are generally hard to extend from basic 512x512/768x768 scales without having special sampling strategy (e.g., MultiDiffusion/ScaleCrafter/DemoFusion, etc.). Stable Diffusion 3 are based on Transformer; I have not seen similar works on this specific design but I find this model capable of generating resolutions beyond 1024x1024 of which is trained. Since the proposed Meissonic is based on Transformer as well, is it capable of generating larger images than 1024x1024?
> > - We have little time left, so I am not requiring this, but having a depicted figure for Masking Strategy in Line 218-228, e.g., figures of sampled masks for different scales/time & accumulated sampled counts for each masks during 48 steps of inference, will greatly benefit the understanding of your masking algorithm.
> >
> > Thank you again for your effort, and I will try the model later.
> >
> > Best,
> > Reviewer BVBS

---

> > > ### Author Response · Authors · 2024-11-25
> > >
> > > Thank you so much for your thoughtful and encouraging comments. We deeply appreciate the time and effort you’ve dedicated to reviewing our manuscript and providing such constructive feedback. Your kind words about the hard work behind our revisions mean a great deal to us, especially as we strive to make this paper a valuable contribution to the academic community.
> > >
> > > **Q1: Is Meissonic capable of generating images larger than 1024x1024?**
> > > > I know this is somewhat superfluous for a conference paper, but have you tried scaling up the sample resolution? SD1.5/SD2.1/SDXL are based on U-Net, so they are generally hard to extend from basic 512x512/768x768 scales without having special sampling strategy (e.g., MultiDiffusion/ScaleCrafter/DemoFusion, etc.). Stable Diffusion 3 are based on Transformer; I have not seen similar works on this specific design but I find this model capable of generating resolutions beyond 1024x1024 of which is trained. Since the proposed Meissonic is based on Transformer as well, is it capable of generating larger images than 1024x1024?
> > >
> > >
> > > **A1:**
> > > Yes, **we provide additional examples of Meissonic generating images larger than 1024x1024 in Appendix N** in the updated manuscript. Meissonic supports generating high-quality images with resolutions ranging from 960x960 to 1280x1280. Beyond this resolution range, the quality of the generated images tends to degrade. Nevertheless, due to the use of Rotary Positional Encoding (RoPE), Meissonic supports sampling at arbitrary resolutions.
> > >
> > >
> > >
> > > **Q2: Could you depict a figure for the Masking Strategy in Lines 218-228?**
> > > > We have little time left, so I am not requiring this, but having a depicted figure for Masking Strategy in Line 218-228, e.g., figures of sampled masks for different scales/time & accumulated sampled counts for each masks during 48 steps of inference, will greatly benefit the understanding of your masking algorithm.
> > >
> > > **A2:**
> > > Depicting the masking strategy for 1024 (32x32) tokens at a 512x512 resolution or 4096 (64x64) tokens at a 1024x1024 resolution would be challenging to visualize effectively. Instead, we present an example using 16 (4x4) tokens for a 64x64 resolution. Below, we outline examples of the text-to-image synthesis process using masked tokens (`X`):
> > >
> > > #### 2-step text-to-image synthesis:
> > >
> > > **Init:**
> > > | X   | X   | X   | X   |
> > > | --- | --- | ---- | --- |
> > > | X   | X   | X   | X   |
> > > | X   | X   | X   | X   |
> > > | X   | X   | X   | X   |
> > >
> > > **Step 1:**
> > >
> > > | X   | X   | 4724 | X   |
> > > | --- | --- | ---- | --- |
> > > | X   | 754 | X    | 4062 |
> > > | X   | X   | X    | X   |
> > > | X   | X   | 4062 | 4062 |
> > >
> > > **Step 2: (Final output)**
> > >
> > > | 2374 | 6727 | 4724 | 4079 |
> > > | ---- | ---- | ---- | ---- |
> > > | 1111 | 754  | 458  | 4062 |
> > > | 2006 | 5097 | 7590 | 7803 |
> > > | 4434 | 2360 | 4062 | 4062 |
> > >
> > > #### 4-step text-to-image synthesis:
> > >
> > > **Init:**
> > >
> > > | X   | X   | X   | X   |
> > > | --- | --- | --- | --- |
> > > | X   | X   | X   | X   |
> > > | X   | X   | X   | X   |
> > > | X   | X   | X   | X   |
> > >
> > > **Step 1:**
> > >
> > > | X   | X   | X   | 4724 |
> > > | --- | --- | --- | ---- |
> > > | X   | X   | X   | X    |
> > > | X   | X   | X   | X    |
> > > | 5956 | X   | X   | X    |
> > >
> > > **Step 2:**
> > >
> > > | X   | X   | X   | 4724 |
> > > | --- | --- | --- | ---- |
> > > | 4422 | X   | 4902 | X    |
> > > | X   | X   | X    | X    |
> > > | 5956 | 6664 | X   | X    |
> > >
> > > **Step 3:**
> > >
> > > | X   | 4511 | 2824 | 4724 |
> > > | --- | ---- | ---- | ---- |
> > > | 4422 | X    | 4902 | 1529 |
> > > | X    | 8035 | X    | X    |
> > > | 5956 | 6664 | 4376 | X    |
> > >
> > > **Step 4: (Final output)**
> > >
> > > | 5951 | 4511 | 2824 | 4724 |
> > > | ---- | ---- | ---- | ---- |
> > > | 4422 | 5008 | 4902 | 1529 |
> > > | 3273 | 8035 | 6543 | 6113 |
> > > | 5956 | 6664 | 4376 | 836  |
> > >
> > > We will release our code for a better understanding of the masking algorithm. Figure 2 in MaskGIT[1] further describes this sampling algorithm, so including a similar figure in the manuscript might be redundant.
> > >
> > > Once again, we are grateful for your generous feedback and constructive suggestions.
> > >
> > > [1] MaskGIT: Masked Generative Image Transformer

---

> > > > ### Comment · Reviewer_BVBS · 2024-11-26
> > > >
> > > > Thank you for providing valuable information. All my concerns are resolved.

---

> > > > > ### Author Response · Authors · 2024-11-26
> > > > > **Thanks**
> > > > >
> > > > > Dear reviewer,
> > > > >
> > > > > Thanks again for raising the score and give response to us!
> > > > >
> > > > > Best Regards!

---

### Official Review · Reviewer_TngQ · 2024-11-04

**Soundness:** 3
**Presentation:** 4
**Contribution:** 3
**Rating:** 6
**Confidence:** 4

**Summary:**

- The paper presents Meissonic, a high-resolution, non-autoregressive masked image modeling (MIM) approach for text to image generation.
- It introduces architectural innovations to elevate MIM’s performance to rival diffusion models like SDXL.
- Its key advancements include a multi-modal transformer architecture, Rotary Position Encoding (RoPE), feature compression layers, and micro-conditions based on human preference scores.
- It enables high-fidelity image generation at 1024×1024 resolution on 8GB VRAM.
- The model is trained on high-quality datasets and achieves state-of-the-art results on various benchmarks.

**Strengths:**

- The integration of multi-modal transformers with single-modal refinements optimizes MIM for high-resolution synthesis and addresses the historical gap in resolution and quality compared to diffusion models.
- The model's efficiency is notable as it requires only 48 H100 GPU days which is far less than other high-resolution models like SDXL while achieving comparable performance.

**Weaknesses:**

- Using CLIP text encoder makes Meissonic potentially less adept at handling complex text prompts compared to models using larger language models like T5 which could impact its effectiveness in scenarios demanding high text comprehension. e.g. generating images with visual texts.
- The reliance on high-quality datasets may limit the model's generalization in open-domain applications.

**Questions:**

Since Meissonic uses a CLIP text encoder, which might limit its effectiveness in scenarios demanding high text comprehension (e.g. generating images with visual texts), could the authors provide examples comparing Meissonic's performance on complex prompts versus simpler prompts? Also, would there be noticeable performance degradation if the model were applied to open-domain data?

---

> ### Author Response · Authors · 2024-11-21
> **Official Response to Reviewer TngQ**
>
> Thank you for your insightful comments and questions.
>
> Q1: How do you balance the size of the text encoder and the ability to generate images with visual texts?
> > Using CLIP text encoder makes Meissonic potentially less adept at handling complex text prompts compared to models using larger language models like T5 which could impact its effectiveness in scenarios demanding high text comprehension. e.g. generating images with visual texts.
> >
> > Since Meissonic uses a CLIP text encoder, which might limit its effectiveness in scenarios demanding high text comprehension (e.g. generating images with visual texts), could the authors provide examples comparing Meissonic's performance on complex prompts versus simpler prompts?
>
> A1: We appreciate the reviewer's question regarding the text encoder selection. Our choice of CLIP (ViT-H-14) over T5-XXL was driven by careful consideration of both training and inference efficiency while maintaining strong text understanding capabilities. As shown in our comparative analysis (see table below), CLIP offers significant advantages in both aspects:
> | Text Encoder | Sequence Length | Embedding Size | 1M Captions Inference Time (min:sec) | 1M Captions Embedding Storage (GB) | VRAM Usage (MB) | Parameter Count (M) |
> |----------------------|------------------|----------------|---------------------------|--------------|-----------------|---------------------|
> | CLIP (ViT-H-14) | 77 Tokens | 1024 | 3:20 | 157 | 1,370 | 354 |
> | T5 (T5-xxl) | 120 Tokens | 4096 | 33:04 | 983 | 18,194 | 4,762 |
>
> The training efficiency heavily favors CLIP, as T5-XXL would require massive storage resources for caption embeddings (approximately 983GB per million captions). For large-scale training with 200M image-text pairs, this would demand nearly 200TB of storage, creating substantial I/O bottlenecks and significantly reducing training throughput. In contrast, CLIP's more compact embeddings (157GB per million captions) enable much more efficient data handling while maintaining robust text understanding.
> Regarding inference efficiency, CLIP demonstrates clear advantages with only 1.37GB VRAM usage compared to T5's 18GB requirement, while delivering 10x faster inference speed (3:20 vs 33:04). This efficiency doesn't come at the cost of performance - CLIP's proven capability in understanding visual-semantic relationships makes it particularly well-suited for image generation tasks. The CLIP text encoder's 354M parameters (versus T5-xxl's 4.7B) provide an excellent balance between computational efficiency and text interpretation capabilities, making it the optimal choice for our system's requirements.
>
>
>
> > e.g. generating images with visual texts.
>
> Additionally, we found that the capability for word rendering primarily stems from larger models like T5 XXL, so **we present Meissonic’s ability to synthesize letters in Figure 10** (as described in line 215 of the original manuscript). Our first goal is to develop an efficient text-to-image foundation model, which guided our choice of text encoder.
>
>
> Furthermore, **we provide performance comparisons for complex and simple prompts in Appendix D in the updated manuscript**. Our findings indicate that with 1–3 word prompts, the model tends to generate simpler images. With 7–10 word prompts, it produces images of sufficient quality, and for prompts up to 70 words, it incorporates more detailed descriptions, although image quality does not significantly improve. As a result, our current model effectively handles prompts of fewer than 77 tokens, generating visually appealing images with satisfactory quality.
>
>
>
> Q2: How about the model's generalization performance?
> > The reliance on high-quality datasets may limit the model's generalization in open-domain applications.
> > Also, would there be noticeable performance degradation if the model were applied to open-domain data?
>
> A2: In the first stage of training, we leverage approximately 200 million high-quality data samples to instill the model with a robust understanding of fundamental concepts (as detailed in Section 2.5, Stage 1, of the original manuscript). These datasets are carefully curated to ensure sufficient diversity across styles, subjects, and contexts, which prevents the model from becoming overly specialized and supports its generalization to open-domain applications.
>
> To demonstrate this, Figure 6 in the original manuscript highlights the model's ability to generate images in a wide range of styles, while Figures 29 to 53 showcase its response to varied open-domain prompts. These results consistently show that the model retains its performance across diverse and open-domain scenarios, with no significant degradation observed.
>
>
>
> We hope the above discussion resolves your concerns.

---

> ### Author Response · Authors · 2024-11-26
> **Please let us know whether all issues are addressed**
>
> Dear reviewer:
>
> We have updated the response and corresponding draft. Moreover, two reviewers have stated that their concerns are solved and one has improved his score. We want to know whether your concerns are solved and whether you can improve your score, since the deadline for discussion is near the end.
>
> If you have more questions, we will reply it as soon as possible.
>
> Best regards!

---

### Author Response · Authors · 2024-11-21
**General Response: Contributions and New Experiments**

We sincerely thank all the reviewers for their insightful and valuable comments! Overall, we are encouraged that they find that:
1. The proposed model in this manuscript can be regarded as a good image generative model with production-ready quality. (Reviewer TngQ, BVBS)
2. The proposed method is efficient enough, in terms of the training cost and inference cost. (all reviewers)
3. The paper is well-written. The technical parts are clear and easy to understand. (Reviewer TngQ, qRU5)
4. The proposed method introduces architectural innovations to elevate MIM’s performance to rival diffusion models like SDXL. (Reviewer TngQ)

And we also want to emphasize our contributions: With five aspects of architectural innovations, Meissonic is the first to elevate MIM to SDXL-level performance, verifying the feasibility and potential of MIM in high-quality image generation. This work sets the base model for future masked image modeling research and presents a foundation model that can run on consumer GPUs with SDXL-level quality.

We have revised the manuscript according to the reviewers' comments:
1. In Appendix D, we present performance comparisons for complex versus simple prompts.
2. In Appendix E, we present performance comparisons with different numbers of inference steps and Classifier Free Guidance (CFG).
3. In Appendix F, we present more comparisons for zero-shot image editing ability.
4. In Appendix G, we present more comparisons with SDXL for image generation ability.
5. We fix typo issues in Figure 7 (new as Figure 8) and include part of Figure 15 (new as Figure 3 and Figure 27) in the main manuscript. We supplement some architecture details and highlight them in blue for clarity (from line 242 - 245, line 284 - 287).

Next, we address each reviewer's detailed concerns point by point. We sincerely thank all reviewers for their recognition of our work and the valuable suggestions provided. Discussions are always welcome. Thank you!

---

### Meta-Review · Area_Chair_jdoP · 2024-12-18

**Metareview:**

The work introduced Messonic that improved MIM with enhanced transformer architecture, advanced positional encoding and adding masking rate as a condition. The effectiveness of proposed approach is demonstrated well and efficiency brought by the improvement is obvious. It consistently received positive feedback from reviewers with three borderline accept. No critical concerns from reviewers but there are some concerns about presentation and the choice of text encoders. Authors made significant effort for the rebuttal which addressed most of concerns after discussions. AC appreciate challenging the traditional backbone and finding sort of breakthrough to improve the foundation models, especially for high resolution generation which is known as non-trivial. After checking all the comments and discussions, a decision of accept is made and authors are suggested to incorporating the useful comments and newly added experiments in the revised version.

**Additional Comments On Reviewer Discussion:**

There is no critical concerns from reviewers but some concerns about presentation and the choice of text encoders are raised. Authors prepared detailed feedback in the rebuttal and two reviewers explicitly mentioned that all concerns are resolved. Since the review is unanimously positive and AC thinks this work is challenging foundation model design which should be encouraged and could impact more following works, the decision of accept is made.

---

### Decision · Program_Chairs · 2025-01-22

Accept (Poster)